# Boosting Targeted Adversarial Transferability: A Generative Approach Guided by Core Target Samples

## Abstract

Adversarial examples generated on one model can often be transferred to other unseen models, but achieving high targeted transferability remains challenging due to overfitting—especially under single-surrogate constraints. In this work, we propose BAT, a generative approach that Boosts targeted Adversarial Transferability by training the generator to align its outputs with a curated set of high-confidence *core target samples*. These samples—either selected from real data or synthesized from noise—serve as guidance across both output and feature spaces. To mitigate overfitting without requiring multiple surrogates, BAT employs an ensemble of frozen discriminators derived via pruning from a single pretrained surrogate model. BAT is applicable whether both the generator's training (source) and the evaluation images come from the target models' training domain or exhibit a domain shift; it remains effective even without real target-class images during training. Extensive experiments on ImageNet-1K show that BAT notably outperforms existing $\ell_\infty$-constrained targeted attacks. We also provide theoretical bounds that reveal how ensemble size influences transferability, aligning with observed empirical trends.

## 1 Introduction

Adversarial examples, imperceptible to humans, can readily deceive deep neural networks (DNNs) (Goodfellow et al., 2014; Moosavi-Dezfooli et al., 2017; Lin et al., 2019). Adversarial attacks are broadly classified into two categories based on attacker's knowledge: *white-box* (Szegedy et al., 2013; Carlini & Wagner, 2017; Moosavi-Dezfooli et al., 2016; Madry et al., 2017; Paniagua et al., 2023) and *black-box* (Chen et al., 2020; Reza et al., 2023; Guo et al., 2019; Dong et al., 2019; Wu et al., 2021) attacks. While white-box attacks presume complete knowledge of the target classifier, black-box attacks do not make such extreme assumptions. Black-box attacks further split into *query-based* (Ilyas et al., 2018; Maho et al., 2021; Rahmati et al., 2020; Reza et al., 2025) and *transferable* (Wang et al., 2024b; Inkawhich et al., 2020a; Wu et al., 2024; Zhu et al., 2024) attacks. Despite improvements in query-based attacks, excessive queries are still needed for success, driving interest in transferable attacks, where adversarial examples are generated using a surrogate model and then transferred to unknown target/victim models.

Depending on the objective, attacks can be either *untargeted* or *targeted*. The use of surrogate models has shown remarkable success in *transferability* for untargeted attacks lately (Zhu et al., 2023; Wang et al., 2024b; 2021; Wang & He, 2021; Chen et al., 2023b). However, their direct adaptations to the *targeted* setting often overfit and fail to learn the target class distribution (Liu et al., 2016). Recently, several innovative approaches have emerged to enhance targeted transferability. Targeted attacks are generally divided into iterative (Inkawhich et al., 2019; Li et al., 2020a) and generative (Naseer et al., 2021; Zhao et al., 2023; Fang et al., 2024) methods. *Iterative* (Zhao et al., 2021; Wei et al., 2023) methods that craft instance-specific perturbations; and *generative* (Wang et al., 2023; Gao et al., 2024) methods that train a generator to produce adversarial examples for arbitrary inputs. Generative methods, which explicitly encourage the generator to learn the target class feature distribution, have proven especially effective for targeted transfer.

Generative adversarial attacks are best described along two orthogonal axes. First, whether the generator's source distribution $\mathcal{P}$ matches the target models' training domain $\mathcal{Q}$ (no domain shift, $\mathcal{P} = \mathcal{Q}$) or differs from it (domain shift, $\mathcal{P} \neq \mathcal{Q}$); unless noted otherwise, evaluation images are also sampled from $\mathcal{P}$. Second, whether training uses *real* target-class images from $\mathcal{Q}$ as *references* in the

loss (*target-data-guided*) or not (*target-data-free*). Many attacks only address learning when domains match ($\mathcal{P} = \mathcal{Q}$) (Zhao et al., 2023; Gao et al., 2024; Sun et al., 2024). Some works also tackle learning when domains are shifted ($\mathcal{P} \neq \mathcal{Q}$), often in a target-data-guided manner that incorporates real target images as references (Naseer et al., 2021; Wang et al., 2023). These target-class references enable measuring distributional distance between generated adversarial examples and target images using a discriminator pretrained on $\mathcal{Q}$. *However, because all target samples are treated uniformly without considering fidelity, the resulting adversarial examples may exhibit lower target-class confidence, ultimately reducing transferability.*

Existing targeted transferable attacks, particularly under a single-surrogate constraint, often suffer from low transfer rates due to discrepancies between the surrogate's and the unknown targets' decision boundaries. To mitigate this, Naseer et al. (2021) replaced a single discriminator with an ensemble of pretrained surrogates, improving transfer by steering perturbations toward regions vulnerable across diverse boundaries. Zhao et al. (2023) instead derived two discriminators from a single surrogate (pretrained vs. fine-tuned) to maximize boundary discrepancy during generator training, but at the cost of extra discriminator training with the access to source sample. Despite empirical evidence, how ensemble *size* impacts targeted transfer remains theoretically underexplored.

Inspired by the effectiveness of model ensembles and motivated by the limitations of prior work (Zhao et al., 2023), we ask: *Can we train a generator to produce highly transferable, targeted adversarial examples using only discriminators derived from a single surrogate—with no additional model training?* To investigate this, we revisit the premise that discriminator diversity improves transferability. Fig. 1 shows that attention regions differ not only across distinct architectures pretrained on ImageNet-1K (Russakovsky et al., 2015) but also across *slightly pruned* variants of a single model (e.g., randomly removing just 2% of weights from ResNet50 (He et al., 2016)) when adversarial examples are crafted with I-FGSM (Kurakin et al., 2018). These observations suggest that a diverse

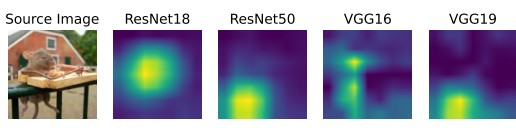

(a) Attention heatmaps on different pretrained classifiers.

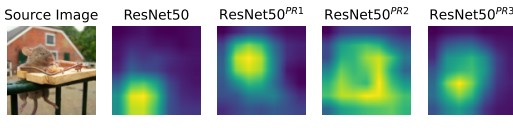

(b) Attention heatmaps on ResNet50 and its different pruned versions.

Figure 1: Attention heatmaps obtained leveraging Grad-CAM (Selvaraju et al., 2017) for adversarial images of a target class crafted on different models.

discriminator ensemble can be obtained from a single model via pruning, with no extra training or architectural changes. While self-ensembling has been explored in iterative attacks (Li et al., 2020b; Wang et al., 2024a), its role in guiding *generative* attacks remains underexplored. Additional related works are provided in Appendix B.

**Our approach: BAT.**  We propose **BAT**, a generative framework that trains a generator by *aligning both output and intermediate feature distributions* of generated adversarial examples with those of a small, carefully selected set of *core target samples*—which are consistently classified as the target class with high confidence across the discriminator ensemble. Under a single-surrogate constraint, BAT builds this ensemble by *pruning* the surrogate to obtain diverse discriminators (no extra training). To the best of our knowledge, BAT is among the first to leverage such a self-ensemble to guide a generative attack using confidence-aware core target samples, encouraging the generator to produce highly confident adversarial examples that generalize to unseen models. Based on the core target sample type, we introduce three variants: **BAT-BS** (Best Samples) selects the most confident real target-class images; **BAT-CS** (Crafted Samples) further increases their confidence via targeted perturbations; and **BAT-CN** (Crafted Noise) uses *no real target-domain images*, synthesizing target-class references from noise. Accordingly, when $\mathcal{P} \neq \mathcal{Q}$, BAT-BS and BAT-CS instantiate *target-data-guided* training, whereas BAT-CN instantiates *target-data-free* training. By combining (i) self-ensembling via pruning with (ii) output–feature alignment to high-confidence *core* targets, BAT achieves state-of-the-art targeted (SOTA) transfer for both $\mathcal{P} = \mathcal{Q}$ and $\mathcal{P} \neq \mathcal{Q}$, including cases without access to real target-domain images  The contributions are as follows:

- We propose BAT, a generative framework that significantly improves targeted adversarial transferability by aligning generated examples with a small set of high-confidence *core target samples* in both output and feature spaces.
- To mitigate overfitting to a single surrogate, BAT exploits an *ensemble of pruned discriminators* from one pretrained model, enhancing transferability without additional training.

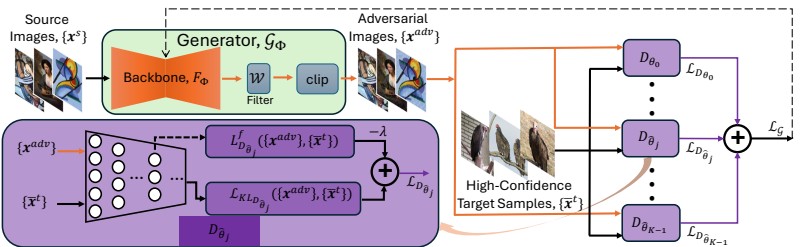

Figure 2: Schematic of BAT, comprising a generator $\mathcal{G}_\Phi$ and $K$ discriminators derived from a single surrogate model $D_{\theta_0}$. The generator is trained to craft adversarial examples for a given target class, with the goal of minimizing the difference between the distribution of the generated adversarial examples and that of the core (high-confidence) target samples.

- When multiple surrogates are available, BAT naturally leverages architectural diversity; pruning remains effective and stable even then.
- Extensive experiments on ImageNet-1K demonstrate that BAT outperforms state-of-the-art $\ell_\infty$-constrained targeted attacks, improving transfer success rates by 6–7% in BAT-CS.
- We theoretically derive lower and upper bounds on transferability, and present trade-off analyses showing how the number of discriminators affects performance.

## 2 PRELIMINARIES

Let $[L] := \{1, \ldots, L\}$ denote the label set and $\mathcal{Y} := \{\boldsymbol{p} \in [0,1]^L : \sum_{c=1}^L p_c = 1\}$ the probability simplex over $L$ classes. We consider an $L$-class classifier with parameters $\theta_j$, modeled as $D_{\theta_j} : \mathcal{X} \to \mathcal{Y}$. Here, $\mathcal{D}_s := \{D_{\theta_j}\}_{j=0}^{K-1} \subset \mathcal{D}$ denotes the set of $K$ classifiers accessible to an adversary and $\mathcal{D}$ the set of all possible classifiers for the same task. Each $D_{\theta_j}$ maps an input image $\boldsymbol{x} \in \mathcal{X} \subset [0,1]^{C \times H \times W}$ to a predicted distribution $\boldsymbol{y} = D_{\theta_j}(\boldsymbol{x}) \in \mathcal{Y}$ over $L$ classes, where $C, H, W$ denote the number of channels, height, and width of $\boldsymbol{x}$, respectively. The *top-1* prediction of $D_{\theta_j}$ is

$$\hat{D}_{\theta_j}(\boldsymbol{x}) = \arg\max_{c \in [L]} \left[ D_{\theta_j}(\boldsymbol{x}) \right]_c,$$

where $\left[ D_{\theta_j}(\boldsymbol{x}) \right]_c$ is the predicted probability of class $c$. Additionally, let $D_{\theta_j}^{(f)} : \mathcal{X} \to \mathbb{R}^{d_f}$ denote the feature extractor corresponding to the $f$-th intermediate layer of $D_{\theta_j}$; we write $\mathcal{F} := \mathbb{R}^{d_f}$ for this feature space.

Let $V : \mathcal{X} \to \mathcal{Y}$ be an *unknown victim* model and $y_t \in [L]$ a specified target class. A **targeted transferable attack** seeks an adversarial example $\boldsymbol{x}^{\mathrm{adv}} = \boldsymbol{x} + \boldsymbol{\delta}$ such that $\hat{V}(\boldsymbol{x}^{\mathrm{adv}}) = y_t$ while satisfying a perceptual constraint $\|\boldsymbol{\delta}\|_\infty \leq \epsilon$.

To encourage transferability to any $V \in \mathcal{D} \setminus \mathcal{D}_s$, we consider the constrained optimization problem

$$\boldsymbol{x}^{\mathrm{adv}} = \arg\min_{\boldsymbol{x}'} \mathbb{E}_{D_{\theta_i} \sim \mathcal{D}} \ell_{D_{\theta_i}}(\boldsymbol{x}', y_t) \quad \text{s.t.} \quad \|\boldsymbol{x}' - \boldsymbol{x}\|_\infty \leq \epsilon, \tag{1}$$

where $\ell_{D_\theta}(\cdot, y_t)$ is a targeted loss (e.g., KL loss Kullback & Leibler (1951)) that encourages $\boldsymbol{x}'$ to be classified as $y_t$ by $D_\theta$, and the $\ell_\infty$ constraint enforces imperceptibility. When an adversary only has access to a finite surrogate set $\mathcal{D}_s$, Eq. equation 1 can be approximated as

$$\boldsymbol{x}^{\mathrm{adv}} = \arg\min_{\boldsymbol{x}'} \frac{1}{|\mathcal{D}_s|} \sum_{D_{\theta_j} \in \mathcal{D}_s} \ell_{D_{\theta_j}}(\boldsymbol{x}', y_t) \quad \text{s.t.} \quad \|\boldsymbol{x}' - \boldsymbol{x}\|_\infty \leq \epsilon. \tag{2}$$

## 3 PROPOSED METHOD: BAT

A schematic of BAT is shown in Fig. 2. BAT trains a single generator $G_\Phi$ to craft $\ell_\infty$-bounded adversarial examples for a target class $y_t$ while the attacker has access to only *one* pretrained surrogate $D_{\theta_0}$, trained on the target domain $\mathcal{Q}$. To obtain the model diversity needed for enhanced transferability, we construct an ensemble $\mathcal{D}_s = \{D_{\theta_j}\}_{j=1}^K$ by *randomly pruning $D_{\theta_0}$*, and *use the pruned copies as discriminators*. All discriminators are *frozen* and require no additional training.

Let $\mathcal{P}$ denote the distribution of source images used to train $G_\Phi$; unless noted otherwise, evaluation images are also drawn from $\mathcal{P}$. Let $\mathcal{Q}$ denote domain of the surrogate and unknown victims models training dataset. We refer to *no domain shift* when $\mathcal{P} = \mathcal{Q}$ and *domain shift* when $\mathcal{P} \neq \mathcal{Q}$. Orthogonal to this axis, we distinguish whether training uses *real* target-class images from $\mathcal{Q}$ as *references* in the loss (*target-data-guided*) or uses no real target-class images (*target-data-free*).

Given a source set $S = {x_i^s}_i$ with $x_i^s \sim P$, we also apply standard data augmentation, following existing generative targeted-transfer attacks, to each source image and compute the loss over both original and augmented views. BAT mitigates overfitting to low-fidelity references by constructing a compact set of *core target samples* $\mathcal{T}^\star$ for the target class $y_t$ using confidence *consensus* across the pruned ensemble $\mathcal{D}_s$. In the target-data-guided setting, **BAT-BS** selects the top-$k$ *real* target images from $Q$ ranked by ensemble confidence for $y_t$, and **BAT-CS** further *crafts* higher-confidence references by perturbing these images toward $y_t$. In the target-data-free setting, **BAT-CN** *synthesizes* target-class references directly from Gaussian noise by ascending ensemble confidence toward $y_t$. Subsequent subsections detail the self-ensemble method, the construction of $\mathcal{T}^\star$ and the dual-space alignment losses that train $G_\Phi$ using the frozen discriminators in $\mathcal{D}_s$.

### 3.1 ENSEMBLE OF PRUNED DISCRIMINATORS

BAT derives an ensemble of $K$ discriminators leveraging pruning of the surrogate, in the constrained access to a single surrogate $D_{\theta_0}$ with parameters $\theta_0 \in \mathbb{R}^d$, where $d$ is the dimension of the parameter space. Then, pruned versions of $D_{\theta_0}$ are obtained through both $L_1$-norm unstructured pruning and random-unstructured pruning (Paszke et al., 2019). The $L_1$-norm unstructured pruning process is formalized as follows:

$$D_{\hat{\theta}_1} = D_{\theta_0 \odot P}, \text{ where } P^{(i)} = \begin{cases} 1, & \text{if } |\theta_0^{(i)}| > \gamma \\ 0, & \text{otherwise,} \end{cases}, \tag{3}$$

where $P \in \{0,1\}^d$ is a binary masking vector and $\odot$ denotes the Hadamard product. Besides, $\gamma$ is a threshold such that $\#\{i \in [d] \mid |\theta_0^{(i)}| \le \gamma\} = p_1 \cdot d$, where $p_1$ is the pruning ratio. Additional pruned models are obtained using random-unstructured pruning, which is expressed as follows:

$$D_{\hat{\theta}_j} = D_{\theta_0 \odot M_j} : M_j^{(i)} \sim \text{Bernoulli}(1 - p_r), \forall i \in [d], j > 1, \tag{4}$$

where $M_j \in \{0,1\}^d$ is another binary masking vector with each element $M_j^{(i)}$ being a Bernoulli random variable, effectively zeroing out the $i$-th parameter with probability $p_r$. Thus, by combining the original model $D_{\theta_0}$ with its pruned variants, an ensemble of $K$ discriminators is given by

$$\mathcal{D}_s = \{D_{\theta_0}\} \cup \{D_{\hat{\theta}_1}\} \cup \{D_{\hat{\theta}_j}\}_{j=2}^{K-1}. \tag{5}$$

While BAT employs these two simple methods for self-ensembling, structured pruning (Paszke et al., 2019) or techniques from (Li et al., 2020b), ensuring diverse discriminators, can also be employed.

### 3.2 CORE TARGET SAMPLES SELECTION

The key objective in BAT is to guide the generator to produce adversarial examples that lie in *high–confidence target regions* in both output and feature spaces under the discriminator ensemble $\mathcal{D}_s$. Thus, the choice of target-class references used during training is crucial for transferability. Let $\mathcal{T}$ denote the available target-class data (if any). Based on how these references are constructed, BAT has three variants: BAT-BS, BAT-CS, and BAT-CN. BAT-BS and BAT-CS assume access to $\mathcal{T}$, while BAT-CN operates without real target images.



(a)  (b)  (c)

Figure 3: (a) Target samples colored by ensemble confidence $\bar{p}(y_t \mid \boldsymbol{x})$ (brighter is higher). (b) Retain high-confidence samples and refine them by bounded targeted perturbations. (c) Resulting crafted references with increased ensemble confidence.

For each discriminator $D_{\theta_j} \in \mathcal{D}_s$, let $p_j(\boldsymbol{x}) := [D_{\theta_j}(\boldsymbol{x})]_{y_t}$ be the softmax confidence for the target label $y_t$. We define the *ensemble confidence* $\bar{p}(y_t \mid \boldsymbol{x}) = \frac{1}{|\mathcal{D}_s|} \sum_{D_{\theta_j} \in \mathcal{D}_s} p_j(\boldsymbol{x})$, and rank candidates according to $\bar{p}(y_t \mid \boldsymbol{x})$. To focus on high–confidence regions, **BAT-BS** selects a subset of benign target images by taking the top-$K$ elements of $\mathcal{T}$:

$$\mathcal{T}_{\text{BS}}^\star = \text{TopK}(\mathcal{T}; \bar{p}(y_t \mid \boldsymbol{x})).$$

Equivalently, $\mathcal{T}_{\text{BS}}^\star$ contains those $\boldsymbol{x} \in \mathcal{T}$ whose ensemble confidence exceeds that of non-selected.

**BAT-CS** further increases the ensemble confidence of each $\boldsymbol{x} \in \mathcal{T}_{\text{BS}}^\star$ by adding targeted perturbations (PGD-style) that maximize $\bar{p}(y_t \mid \boldsymbol{x})$, as shown in Fig. 3 and detailed in Algorithm 1. This produces a refined set $\mathcal{T}_{\text{CS}}^\star$ that lies deeper in the high–confidence target region. Conversely, **BAT-CN** synthesizes a high-confidence set $\mathcal{T}_{\text{CN}}^\star$ by optimizing the same objective starting from noise initializations $\boldsymbol{n} \sim \mathcal{N}(0, I)$ (with clipping to $[0,1]$), which does not require access to real target-domain images.

We refer to $\mathcal{T}_{\text{BS}}^\star$, $\mathcal{T}_{\text{CS}}^\star$, and $\mathcal{T}_{\text{CN}}^\star$ collectively as the *core target set* $\mathcal{T}^\star$, which is then used to drive output- and feature-space alignment in the subsequent loss.

### 3.3 Distributions Distance Measurement

To guide the generator in crafting transferable adversarial examples, BAT minimizes the discrepancy between the generated examples and the core target samples in both *output space* and *feature space*. This dual-space alignment is enforced across all discriminators in the ensemble $\mathcal{D}_s$.

**(i) Output Distribution Alignment.** We use *Kullback–Leibler (KL) divergence* to quantify the mismatch between the predicted class distributions of the generated adversarial examples and the core target samples. For a mini-batch of size $B$, the symmetric KL divergence on a discriminator $D_{\theta_j} \in \mathcal{D}_s$ is given by:

$$\mathcal{L}^{\text{KL}}_{D_{\theta_j}} = \frac{1}{B} \sum_{i=1}^{B} \left[ \text{KL}(D_{\theta_j}(\boldsymbol{x}_i^{adv}) \,\|\, D_{\theta_j}(\boldsymbol{x}_i^{t\star})) + \text{KL}(D_{\theta_j}(\boldsymbol{x}_i^{t\star}) \,\|\, D_{\theta_j}(\boldsymbol{x}_i^{adv})) \right] \tag{6}$$

where $\boldsymbol{x}_i^{adv}$ is a generated adversarial example and $\boldsymbol{x}_i^{t\star} \in \mathcal{T}^*$ is a core target sample. The symmetric formulation ensures stable optimization and mutual alignment between distributions.

**(ii) Feature Distribution Alignment.** To further constrain the generator to match the internal target-class representation, we measure the *cosine similarity* between the intermediate features of the generated and core samples:

$$\mathcal{L}^{f}_{D_{\theta_j}} = \frac{1}{B} \sum_{i=1}^{B} \cos \langle h_j^{(f)}(\boldsymbol{x}_i^{adv}), h_j^{(f)}(\boldsymbol{x}_i^{t\star}) \rangle, \tag{7}$$

where $h_j^{(f)}(\boldsymbol{x}) = D_{\theta_j}^f(\boldsymbol{x})/\|D_{\theta_j}^f(\boldsymbol{x})\|_2$, and $D_{\theta_j}^f(\boldsymbol{x})$ denotes the intermediate feature representation extracted from the $f^{th}$ layer of discriminator $D_{\theta_j}$.

These losses collectively ensure that generated examples resemble high-confidence target-class samples both at the output and representational levels, improving generalization to unseen models.

### 3.4 Generator Training

The goal of the generator training is to update the parameters $\Phi$ of $\mathcal{G}_\Phi$ so that it learns to generate an adversarial example $\boldsymbol{x}_i^{adv}$, for a source image $\boldsymbol{x}_i^s$, which is capable of mapping to the target class with high transferability satisfying the perturbation constraint $\|\boldsymbol{x}_i^{adv} - \boldsymbol{x}_i^s\|_\infty \leq \epsilon$. We use the same generator backbone, $F_\Phi$, as in (Zhao et al., 2023; Naseer et al., 2021; Wang et al., 2023). The architecture of the generator is provided in Appendix F. The output from the generator satisfying the perturbation constraint can be expressed as:

$$\boldsymbol{x}_i^{adv} = \mathcal{G}_\Phi(\boldsymbol{x}_i^s) = \text{clip}(\mathcal{W} * F_\Phi(\boldsymbol{x}_i^s)), \tag{8}$$

where $\mathcal{W}$ is a smoothing parameter with fixed weights to filter out the high-frequency components from the generated image, and $\text{clip}(\mathcal{W} * F_\Phi(\boldsymbol{x}_i^s)) = \min(\boldsymbol{x}_i^s + \epsilon, \max(\mathcal{W} * F_\Phi(\boldsymbol{x}_i^s), \boldsymbol{x}_i^s - \epsilon))$ keeps each pixel of $\boldsymbol{x}_i^{adv}$ within the perturbation budget $\epsilon$. The generator is optimized using the combined distribution alignment loss defined in Section 3.3. Specifically, the total loss is:

$$\mathcal{L}_\mathcal{G} = \frac{1}{|\mathcal{D}_s|} \sum_{D_{\theta_j} \in \mathcal{D}_s} \left[ \mathcal{L}^{\text{KL}}_{D_{\theta_j}} - \lambda \, \mathcal{L}^{f}_{D_{\theta_j}} \right], \tag{9}$$

where $\mathcal{L}_\mathcal{G}$ captures the distributions distance between the generated adversarial examples and high-confidence target samples, both in output and feature spaces, for all the discriminators $D_{\theta_j} \in \mathcal{D}_s$, while $\lambda$ controls the weight of the feature alignment term. The training procedure is outlined in Algorithm 2, provided in the Appendix.

## 4 Experiments

**Baselines and hyperparameter settings.** We compare **BAT** against state-of-the-art *transferable targeted* attacks: two iterative methods (Po-Trip (Li et al., 2020a) and SU (Wei et al., 2023)) and four generative methods (TTP (Naseer et al., 2021), M3D (Zhao et al., 2023), ESMA (Gao et al., 2024), and CGNC (Fang et al., 2024)). ESMA and CGNC train a single generator for multiple target classes, which typically reduces transfer; for fairness we also report **CGNC**$_{\text{FT}}$, obtained by fine-tuning the CGNC generator separately for each target class.

During BAT training, all discriminators $D_{\theta_j} \in \mathcal{D}_s$ are *frozen*; only the generator parameters $\Phi$ are updated. We use the backbone $F_\Phi$ and optimize with Adam (initial learning rate $2 \times 10^{-3}$, exponential decay each epoch; $\beta_1 = 0.5$, $\beta_2 = 0.999$) for $T = 20$ epochs with mini-batch size 16. Unless

Table 1: TSR(%) of various attacks on different target classifiers under $\mathcal{P}=\mathcal{Q}$. BAT variants, specifically BAT-CS and BAT-CN, outperform the SOTA methods by a large margin. '*' indicates the performance on the white-box surrogate model ($D_{\theta_0}$). For each target model, the best overall method is highlighted in bold, while the best baseline method is underlined. Values in parentheses indicate the improvement in TSR(%) over the best baseline.

| Surrogate | Attack | RN18 | RN50 | RN101 | DN121 | DN161 | VGG16$_{BN}$ | VGG19$_{BN}$ | MN-V2 | ViT-B | Average |
|---|---|---|---|---|---|---|---|---|---|---|---|
| RN50 | Po-Trip | 39.84 | 99.90* | 56.95 | 61.26 | 61.87 | 21.28 | 23.90 | 19.18 | 3.81 | 43.11 |
| | SU | 69.84 | 97.78* | 79.83 | 76.35 | 77.62 | 71.82 | 72.00 | 50.88 | 6.71 | 66.98 |
| | ESMA | 57.74 | 92.75* | 66.71 | 65.59 | 64.87 | 72.04 | 66.99 | 54.04 | 21.97 | 62.52 |
| | CGNC | 79.02 | 96.14* | 84.82 | 83.26 | 84.34 | 80.71 | 75.14 | 65.31 | 24.56 | 74.81 |
| | CGNC$_{FT}$ | 85.67 | 96.50* | 89.17 | 88.83 | 89.17 | 85.17 | 81.33 | 75.83 | 40.83 | 81.39 |
| | TTP | 78.06 | 94.96* | 80.16 | 74.39 | 72.11 | 80.93 | 70.79 | 62.92 | 22.22 | 70.73 |
| | M3D | 86.50 | 95.77* | 88.73 | 88.32 | 87.62 | 84.17 | 82.57 | 81.54 | 51.73 | 82.99 |
| | BAT-BS | 89.61 | 98.08* | 92.76 | 92.23 | 89.73 | 92.64 | 89.67 | 81.76 | 42.67 | 85.46$_{(+2.35)}$ |
| | BAT-CS | 93.78 | 98.78* | 95.22 | 94.16 | 93.31 | 94.45 | 94.04 | 86.60 | 50.45 | 88.98$_{(+5.87)}$ |
| | BAT-CN | 92.26 | 98.68* | 94.57 | 93.94 | 92.51 | 93.70 | 92.13 | 85.46 | 47.27 | 87.84$_{(+4.73)}$ |
| DN121 | Po-Trip | 23.43 | 25.36 | 23.67 | 99.96* | 54.14 | 10.64 | 13.36 | 13.18 | 2.75 | 29.61 |
| | SU | 50.02 | 58.08 | 47.47 | 98.50* | 78.72 | 49.46 | 53.43 | 31.05 | 5.08 | 52.42 |
| | ESMA | 62.29 | 66.60 | 54.97 | 94.67* | 77.80 | 66.15 | 60.22 | 46.14 | 20.70 | 61.06 |
| | CGNC | 62.14 | 73.82 | 63.14 | 93.90* | 74.20 | 65.78 | 74.23 | 56.44 | 24.60 | 65.36 |
| | CGNC$_{FT}$ | 74.45 | 84.72 | 72.61 | 94.48* | 85.19 | 80.28 | 81.49 | 70.14 | 34.66 | 75.34 |
| | TTP | 64.71 | 61.27 | 60.54 | 93.75* | 69.19 | 62.37 | 57.41 | 51.06 | 23.32 | 60.40 |
| | M3D | 82.79 | 85.48 | 80.34 | 96.86* | 88.17 | 80.96 | 79.28 | 75.16 | 48.77 | 79.76 |
| | BAT-BS | 88.80 | 86.05 | 83.79 | 98.75* | 88.97 | 83.53 | 82.38 | 76.49 | 42.02 | 81.20$_{(+1.44)}$ |
| | BAT-CS | 92.46 | 92.30 | 90.51 | 99.15* | 92.02 | 90.71 | 89.51 | 81.66 | 48.36 | 86.30$_{(+6.54)}$ |
| | BAT-CN | 92.11 | 91.82 | 89.79 | 99.14* | 93.90 | 91.18 | 88.38 | 79.21 | 48.45 | 86.00$_{(+6.24)}$ |

otherwise specified, we train on 12 randomly selected ImageNet-1K target classes using a pretrained ResNet-50 (He et al., 2016) surrogate and repeat with a pretrained DenseNet-121 (Huang et al., 2017). To build $\mathcal{D}_s$, we include the unpruned surrogate and its pruned variants using magnitude (L$_1$) unstructured pruning with ratio $p_1 = 0.6$ (60% weights pruned) and random unstructured pruning with probability $p_r = 0.02$ (2% per-weight pruning); features are taken from block-3 for both architectures (as in SU (Wei et al., 2023)). By default we use $|\mathcal{D}_s| = 5$, an $\ell_\infty$ perturbation budget $\epsilon = 16/255$, and loss weights $\lambda = 1.5$ for the output/feature alignment terms (see Eq. 9).

**Dataset.** To evaluate BAT under both *no domain shift* ($\mathcal{P}=\mathcal{Q}$) and *domain shift* ($\mathcal{P}\neq\mathcal{Q}$), following TTP (Naseer et al., 2021) we use **ImageNet-1K** (Russakovsky et al., 2015) and the **Painting** dataset (Saleh & Elgammal, 2015). All surrogate and victim models (and thus the discriminator ensemble) are trained on ImageNet-1K, which we take as the models' training domain $\mathcal{Q}$. In the no-shift setting we train the generator on ImageNet-1K ($\mathcal{P}=\mathcal{Q}$); in the shift setting we train on Painting ($\mathcal{P}\neq\mathcal{Q}$). For training, we sample 50,000 source images, and for evaluation, we consider 5,000 validation images from the corresponding domain. Additionally, when $\mathcal{P}\neq\mathcal{Q}$, we report results on 5,000 ImageNet images. Unless otherwise specified, we perform experiments under $\mathcal{P}=\mathcal{Q}$.

BAT uses three variants—BAT-BS, BAT-CS, and BAT-CN—distinguished by how core target samples are constructed. For each target class, **BAT-BS** ranks approximately 1,300 ImageNet-1K training images of that class by ensemble confidence and selects the top $k=300$. **BAT-CS** starts from these 300 and increases their target confidence using Algorithm 1. **BAT-CN** initializes 300 references from Gaussian noise and applies the same algorithm, using no real target-domain ($\mathcal{Q}$) images. For crafting, we use step size $\alpha_c=0.25$ for BAT-CS and $\alpha_c=1$ for BAT-CN with $T_c=25$ updates.

**Target models.** To assess the effectiveness of the adversarial examples produced by the trained generator, we evaluate transferability on unseen victim models pretrained on ImageNet-1K: VGG-16$_{BN}$, VGG-19$_{BN}$ (Simonyan & Zisserman, 2014), ResNet-18/50/101 (RN18/RN50/RN101) (He et al., 2016), DenseNet-121/161 (DN121/DN161) (Huang et al., 2017), MobileNetV2 (MNv2) (Sandler et al., 2018), and ViT-B (Dosovitskiy et al., 2020). Beyond standard classifiers, we also test against robustly trained models: adversarially trained Inception-v3 (Inc-v3$_{adv}$) (Kurakin et al., 2016), ensemble adversarially trained Inception-ResNet-v2 (IR-v2$_{ens}$) (Tramèr et al., 2017), and four robustness-oriented ResNet-50 variants—RN50$_{SIN}$ (Stylized-ImageNet), RN50$_{IN}$ (stylized + natural ImageNet) (Geirhos et al., 2018), RN50$_{fine}$ (fine-tuned RN50$_{IN}$ with an auxiliary set), and RN50$_{Aux}$ (AugMix) (Hendrycks et al., 2019). We additionally evaluate under input-processing defenses; detailed results are provided in the Appendix (Tab. 8).

**Evaluation metric.** We report the *transfer success rate* (TSR) for targeted attacks, i.e., the percentage of adversarial examples that cause an *unknown* victim to predict the intended target label. For a

Table 2: TSR(%) of various attacks on different target classifiers under $\mathcal{P}\neq\mathcal{Q}$ where the source images to train the generators are sampled from the Painting dataset. The performance is evaluated on the Painting test set. '*' indicates the performance on the white-box surrogate ($D_{\theta_0}$). For each target model, the best overall method is highlighted in bold, while the best baseline method is underlined. Values in parentheses indicate the improvement in TSR(%) over the best baseline.

| Surrogate | Attack | RN18 | RN50 | RN101 | DN121 | DN161 | VGG16$_{BN}$ | VGG19$_{BN}$ | MN-V2 | ViT-B | Average |
|---|---|---|---|---|---|---|---|---|---|---|---|
| RN50 | TTP | 76.41 | 93.07* | 74.29 | 79.48 | 75.83 | 78.09 | 65.02 | 56.54 | 37.07 | 70.64 |
| | CGNC | 83.09 | 97.48* | 81.61 | 80.98 | 82.79 | 86.24 | 82.56 | 71.5 | 46.01 | 79.14 |
| | CGNC$_{FT}$ | 91.43 | 98.56* | 94.75 | 91.69 | 89.75 | 91.35 | 87.16 | 78.29 | 58.82 | 86.87 |
| | BAT-BS | 92.65 | 98.16* | 94.40 | 93.15 | 92.66 | 92.78 | 87.01 | 83.84 | 61.17 | 88.42$_{(+1.55)}$ |
| | BAT-CS | 93.48 | **98.93*** | **96.00** | **96.27** | **95.41** | **95.44** | **93.68** | **90.10** | **73.58** | **92.54**$_{(+5.67)}$ |
| | BAT-CN | **93.73** | 98.88* | 95.80 | 95.82 | 94.82 | 94.73 | 93.52 | 88.69 | 69.94 | 91.77$_{(+4.90)}$ |
| DN121 | TTP | 65.89 | 64.85 | 61.94 | 94.56* | 76.61 | 64.04 | 53.55 | 46.72 | 27.76 | 61.77 |
| | CGNC | 82.80 | 82.58 | 77.73 | 98.26* | 89.90 | 83.13 | 78.88 | 63.83 | 49.21 | 78.48 |
| | CGNC$_{FT}$ | 88.71 | 90.20 | 85.66 | 98.46* | 92.68 | 90.41 | 86.55 | 76.13 | 56.01 | 84.98 |
| | BAT-BS | 88.82 | 90.67 | 86.24 | 98.45* | 90.20 | 89.10 | 87.11 | 77.10 | 59.57 | 85.25$_{(+0.27)}$ |
| | BAT-CS | **94.00** | **95.40** | **93.46** | **99.13*** | **95.63** | **94.51** | **93.30** | **82.42** | **70.17** | **90.89**$_{(+5.91)}$ |
| | BAT-CN | 92.36 | 93.69 | 91.50 | 99.02* | 94.75 | 91.56 | 90.05 | 78.41 | 69.97 | 89.03$_{(+4.05)}$ |

given victim $D_{\theta_k} \in \mathcal{D} \setminus \mathcal{D}_s$, a target-class set $\Upsilon$, and $N$ evaluation images per class, the TSR is

$$\text{TSR}(\%) = \frac{100}{N \cdot |\Upsilon|} \sum_{y_t \in \Upsilon} \sum_{i=1}^{N} \mathbb{1}\big(\hat{D}_{\theta_k}(\mathcal{G}_\Phi^{(y_t)}(\boldsymbol{x}_i)) = y_t\big), \tag{10}$$

where $\mathcal{G}_\Phi^{(y_t)}$ is the generator trained for target class $y_t$, $\boldsymbol{x}_i$ are evaluation inputs, and $\hat{D}_{\theta_k}$ denotes the top-1 prediction. For multiple victims, we also report the average TSR across the evaluation set.

**Performance under no domain shift.** Tab. 1 compares TSR across all methods with $\mathcal{P}=\mathcal{Q}$. Consistent with prior work, *generative* approaches substantially outperform *iterative* ones in targeted transfer. All three BAT variants attain the highest average TSR, which correlates with their ability to produce adversarial examples with *higher target-class confidence* (Tab. 10; Appendix C). In particular, the crafted-target variants (BAT-CS) yield the largest gains by explicitly concentrating training on higher-confidence regions. Remarkably, BAT-CN remains competitive despite using no real target-domain images, underscoring the strength of confidence-guided references synthesized from noise. BAT variants also retain their advantage under tighter perturbation budgets (Tab. 15; see Appendix C), indicating robustness to smaller $\ell_\infty$ constraints. To further demonstrate generalization, we also use ViT-B as the surrogate and observe consistent gains (Tab. 13, Appendix C). In addition, BAT remains effective under an $\ell_2$ threat model without retraining the generator (Tab. 14, Appx. C).

**Performance under domain shift.** For the $\mathcal{P}\neq\mathcal{Q}$ setting, we compare against methods *applicable* under domain shift—TTP (Naseer et al., 2021) and CGNC (Fang et al., 2024)—and exclude methods that require source images from the target domain (e.g., ESMA (Gao et al., 2024), M3D (Zhao et al., 2023)). Tab. 2 reports TSR on the *Painting* test set: BAT substantially improves transferability in this regime as well, with **BAT-CS** and **BAT-CN** achieving results comparable to their no-shift performance. Notably, while TTP is *target-data-guided* (uses real target-class images from $\mathcal{Q}$ as references), **BAT-CN** is *target-data-free* and still surpasses it without any real images. Additional results trained on Painting and evaluated on ImageNet-1K are provided in Appendix C, Tab. 11. To further demonstrate universality under domain shift, we also compare BAT-CS and M3D on the AnimeFace–ImageNet pair (Tab. 12; Appendix C).

**Applicability matrix.** Tab. 3 summarizes the generative methods we evaluate, organized by (i) domain match vs. shift ($\mathcal{P}=\mathcal{Q}$ vs. $\mathcal{P}\neq\mathcal{Q}$) and (ii) references used in the loss—*target-data-guided* if real $\mathcal{Q}$ target images are used, *target-data-free* otherwise. All methods use surrogates trained on $\mathcal{Q}$. A checkmark (✓) indicates the setting is demonstrated in prior work or directly applicable without modification; a cross (×) indicates it is unsupported. As shown, all three **BAT** variants apply to both the matched and shifted regimes: **BAT-BS** and **BAT-CS** are *target-data-guided* (like TTP), whereas **BAT-CN** is *target-data-free* (like CGNC). Across both regimes ($\mathcal{P}=\mathcal{Q}$ and $\mathcal{P}\neq\mathcal{Q}$), BAT consistently achieves higher targeted success rates than TTP/CGNC (see Tab. 1 and Tab. 2).

Table 3: **Applicability matrix.** Which settings each generative method supports: $\mathcal{P} = \mathcal{Q}$ vs. $\mathcal{P} \neq \mathcal{Q}$, and target-data-guided vs. target-data-free losses.

| Method | Domain match/shift | | References in loss | |
|---|---|---|---|---|
| | $\mathcal{P}=\mathcal{Q}$ | $\mathcal{P}\neq\mathcal{Q}$ | guided | free |
| ESMA | ✓ | × | × | ✓ |
| TTP | ✓ | ✓ | ✓ | × |
| CGNC | ✓ | ✓ | × | ✓ |
| M3D | ✓ | × | × | ✓ |
| BAT-BS | ✓ | ✓ | ✓ | × |
| BAT-CS | ✓ | ✓ | ✓ | × |
| BAT-CN | ✓ | ✓ | × | ✓ |

**Performance against robust models.** Tab. 4 compares the TSR, considering ResNet50 as surrogate, against six robust models that are evaluated at two perturbation thresholds: $\epsilon = \frac{16}{255}$ and $\frac{32}{255}$. As expected, TSR increases with $\epsilon$. BAT variants perform better against the mentioned robust models than the baseline attacks. TSR considering DenseNet121 as the surrogate against these models, along with experiments demonstrating the BAT variants' effectiveness against input processing defenses and the robustness of BAT due to the variability introduced by random pruning, are discussed in Appendix C.

**Impact of discriminators from different surrogates.** We analyze how using discriminators from various pretrained surrogates affects TSR. Tab. 5 demonstrates that BAT-CS, which employs an ensemble of discriminators derived from a single ResNet50 model through pruning, achieves a higher average TSR than $TTP_{ens}$ (Naseer et al., 2021), which uses five distinct pretrained ResNet models. The performance of BAT-CS improves when its discriminators are replaced with pretrained ResNet models similar to $TTP_{ens}$ (third row). Furthermore, using discriminators from three model families—ResNet, DenseNet, and VGG—slightly boosts TSR compared to using only ResNet (rows four and five). When deriving five discriminators from two model families, *i.e.*, ResNet50 with two pruned versions of it and DenseNet121 with a pruned version of it, BAT-CS achieves similar TSR to that with diverse pretrained models from single or multiple model families (last row). These results suggest that BAT-CS can further boost TSR by leveraging discriminator ensembles from diverse surrogate models, when available, and pruned versions of these models, indicating the effectiveness of pruning.

Table 4: TSR(%) comparison among the generative methods, considering RN50 as surrogate, under $\mathcal{P}=\mathcal{Q}$, against classifiers with robust training mechanism on ImageNet.

| Surrogate | $\epsilon$ | Attack | Inc-v3$_{adv}$ | IR-v2$_{ens}$ | RN50$_{SIN}$ | RN50$_{IN}$ | RN50$_{fine}$ | RN50$_{Aux}$ |
|---|---|---|---|---|---|---|---|---|
| RN50 | $\frac{16}{255}$ | ESMA | 1.10 | 1.07 | 28.03 | 74.73 | 78.10 | 54.94 |
| | | TTP | 6.25 | 6.05 | 26.68 | 80.97 | 79.91 | 69.51 |
| | | M3D | 7.25 | 8.21 | 45.69 | 88.60 | 91.73 | 80.54 |
| | | CGNC$_{PT}$ | 7.33 | 9.26 | 34.98 | 91.23 | 93.48 | 78.05 |
| | | BAT-BS | 10.22 | 12.68 | 52.25 | 93.23 | 92.29 | 85.22 |
| | | BAT-CS | **10.33** | **12.94** | 57.28 | **95.66** | 94.63 | 87.34 |
| | | BAT-CN | 9.26 | 12.44 | 57.11 | 95.33 | **95.33** | **87.76** |
| | $\frac{32}{255}$ | ESMA | 10.93 | 15.02 | 43.38 | 78.75 | 79.07 | 63.66 |
| | | TTP | 23.61 | 25.92 | 37.48 | 81.28 | 80.27 | 73.86 |
| | | M3D | 21.57 | 39.00 | 61.33 | 92.38 | 93.43 | 88.89 |
| | | CGNC$_{PT}$ | 33.19 | 44.61 | 62.95 | 94.25 | 93.85 | 90.97 |
| | | BAT-BS | 38.16 | 47.50 | 64.33 | 95.48 | 94.40 | 90.03 |
| | | BAT-CS | **41.60** | **51.53** | **72.28** | 96.46 | 94.31 | **92.64** |
| | | BAT-CN | 39.36 | 50.53 | 71.74 | **97.24** | **95.54** | 92.03 |

Table 5: TSR(%) comparison of BAT-CS and $TTP_{ens}$ using different combinations of the five discriminators derived from one or more surrogates. Symbols: '†' indicates generator training leveraging pretrained ResNet{18, 34, 50, 101, 152}, '‡' indicates leveraging ResNet{18, 50}, DN121 and VGG{16, 16$_{BN}$}, and '◇' indicates leveraging RN50, two pruned versions of RN50, DN121 and one pruned DN121 as discriminators. '*' marks white-box surrogate performance.

| Attack | RN18 | RN50 | RN101 | DN121 | DN161 | VGG16$_{BN}$ | VGG19$_{BN}$ | MN-V2 | ViT-B | Average |
|---|---|---|---|---|---|---|---|---|---|---|
| BAT-CS | 93.78 | 98.78* | 95.22 | 94.16 | 93.31 | 94.45 | 94.04 | 86.60 | 50.45 | 88.98 |
| $TTP_{ens}$† | 96.15* | 96.36* | 97.12* | 92.25 | 91.90 | 88.91 | 89.72 | 88.41 | 48.32 | 87.68 |
| BAT-CS† | 98.50* | 98.28* | 98.44* | 97.29 | 96.71 | 96.38 | 95.64 | 93.47 | 59.80 | 92.72 |
| $TTP_{ens}$‡ | 95.41* | 95.45* | 91.76 | 95.46* | 90.06 | 94.33* | 90.52 | 88.90 | 49.03 | 87.88 |
| BAT-CS‡ | 98.45* | 97.81* | 96.14 | 98.22* | 96.42 | 98.22* | 96.06 | 94.22 | 61.61 | 93.02 |
| BAT-CS◇ | 95.98 | 98.62* | 96.55 | 98.67* | 96.23 | 96.77 | 95.81 | 91.63 | 58.53 | 92.09 |

**Ablation study.** Tab. 6 presents the step-by-step progression of the BAT framework, beginning with a baseline using a single discriminator and all available target class samples ($\sim$1300). Increasing the number of discriminators to 5 via pruning improves TSR from 71.12% to 75.85%. Replacing all samples with a curated set of 300 high-confidence target samples also yields a boost (78.35%) even with a single discriminator. Combining both—core target samples and pruned ensemble—raises TSR to 85.46%. Finally, BAT-CS and BAT-CN—both employing five discriminators and confidently crafted core samples—further elevate the TSR to 88.98% and 87.84%, respectively. These results highlight the individual and combined benefits of discriminator diversity and confidence-aware target selection.

Table 6: Ablation study on BAT variants showing the impact of discriminator size ($|\mathcal{D}_s|$) and core target sample selection on TSR (%).

| Method Variant | $|\mathcal{D}_s|$ | Target Sample Selection | TSR (%) |
|---|---|---|---|
| BAT (baseline) | 1 | All ($\sim$1300) | 71.12 |
| BAT-BS | 5 | All ($\sim$1300) | 75.85 |
| BAT-BS | 1 | Core (best 300) | 78.35 |
| BAT-BS | 5 | Core (best 300) | 85.46 |
| BAT-CS | 5 | Confident Core (from best 300) | 88.98 |
| BAT-CN | 5 | Crafted Core (from noise) | 87.84 |

**Impact of $|\mathcal{D}_s|$ and $\lambda$.** Fig. 4a demonstrates the impact of the number of discriminators $|\mathcal{D}_s|$ on TSR. As shown, the TSR increases with $|\mathcal{D}_s|$ and quickly begins to saturate as $|\mathcal{D}_s|$ increases. However, this improvement comes at the cost of increased training time. Thus, a tradeoff exists between TSR and training time. For a comprehensive analysis of this trade-off, including a comparison of training times across all methods, please refer to Section D. Moreover, we investigate the impact of $\lambda$ in Eq. 9 on TSR. From Fig. 4b, the inclusion of cosine similarity between the adversarial and core

target samples in the feature space in the loss function enhances TSR than that without ($\lambda = 0$). The maximum TSR is obtained when $\lambda = 1.5$. We use ResNet50 as the surrogate to depict these figures. Details on the discussion of multi-seed stability, and the influence of target sample size are provided in Appendix C.

**Choice of pruning parameters and pruning-induced diversity.** Our goal is to preserve the accuracy of pruned models while inducing sufficient diversity in their decision boundaries. We construct the discriminator ensemble using one $L_1$-pruned variant ($p_1 = 0.6$) and several randomly pruned variants, and quantify pruning-induced diversity via *decision disagreement*: the fraction of images on which a randomly pruned model's top-1 prediction differs from that of the unpruned ResNet50.

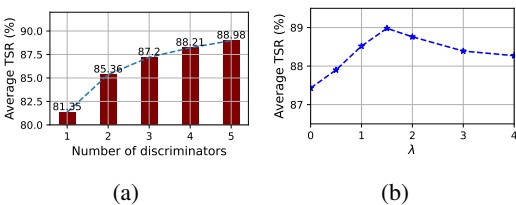

(a) (b)

Figure 4: (a) TSR of BAT-CS for different numbers of discriminators, (b) TSR of BAT-CS for different values of $\lambda$.

Tab. 7 reports both disagreement and TSR for BAT-BS as we vary the random pruning ratio $p_r$. Moving from $p_r = 0$ (no random pruning) to $p_r = 0.01$ and $0.02$ increases disagreement from $0\%$ to $15.06\%$ and $23.16\%$, and TSR from $78.35\%$ to $83.06\%$ and $85.46\%$, indicating that moderate additional diversity is beneficial for targeted transfer. However, further increasing $p_r$ to $0.05$ and $0.10$ pushes disagreement above $48\%$ and $74\%$, while TSR drops to $78.59\%$ and $72.62\%$, showing that over-pruned subnetworks lose discriminative power and provide noisy gradients despite their high disagreement. For reference, when we compare ResNet50 to ten standard architectures (`RN18`, `RN101`, `vgg16`, `VGG19`, `VGG`$_{BN}$, `VGG19`$_{BN}$, `DN121`, `DN169`, `MN`$_{v2}$, `ViT-B`), the average inter-architecture *decision disagreement* is $22.35\%$, very close to the disagreement at $p_r = 0.02$ ($23.16\%$). This suggests that moderate pruning induces a level of diversity comparable to that obtained from heterogeneous architectures.

Table 7: Impact of random pruning ratio $p_r$ on ensemble diversity and TSR for BAT-BS with ResNet50 as surrogate. "Disagreement" is the fraction of validation images for which a randomly pruned model's top-1 prediction differs from the unpruned ResNet50.

| $p_r$ | 0.0 | 0.01 | 0.02 | 0.05 | 0.10 |
|---|---|---|---|---|---|
| Decision disagreement (%) | 0.0 | 15.06 | 23.16 | 48.12 | 74.16 |
| TSR (%) | 78.35 | 83.06 | 85.46 | 78.59 | 72.62 |

This analysis makes the pruning–diversity trade-off explicit: there is a *sweet spot* where random pruning introduces enough diversity to help the ensemble without significantly harming accuracy. The chosen setting $p_r = 0.02$ lies in this regime and yields the best TSR, with a disagreement level comparable to both the $60\%$ $L_1$-pruned model (disagreement $\approx 25\%$) and the average inter-architecture disagreement ($22.35\%$). Overall, these results empirically support the use of pruning to induce useful ensemble diversity and clarify the main *failure mode*: excessive pruning, where weakened discriminators provide noisy gradients and fail to guide the generator toward meaningful targeted adversarial examples.

## 5 THEORETICAL ANALYSIS

As in Eq. 1, ideally an adversary aims to generate adversarial examples that minimize the expected loss across all possible classifiers in $\mathcal{D}$, ensuring high transferability. Additionally, it has been observed that model ensemble offers greater robustness against adversarial attacks Pang et al. (2019). Based on these observations, our theoretical analysis considers an extreme case: a virtual victim model $\bar{V} \in \mathcal{D}$, which is the ensemble average of all possible models in $\mathcal{D}$, *i.e.* $\ell_{\bar{V}}(\boldsymbol{x}, y_t) = \mathbb{E}_{D_{\theta_i} \sim \mathcal{D}} \left[ \ell_{D_{\theta_i}}(\boldsymbol{x}, y_t) \right]$. Intuitively, adversarial examples capable of deceiving this virtual model can deceive any unknown classifier with higher probability.

### 5.1 LOWER BOUND OF TRANSFERABILITY

In this part, we theoretically demonstrate the impact of the number of accessible models on the lower-bound of transferability, which is inspired by Yang et al. (2021).

**Theorem 1.** *Consider,* $\exists \bar{V} \in \mathcal{D}$, *a virtual victim model, such that* $\nabla_{\boldsymbol{x}} \ell_{\bar{V}}(\boldsymbol{x}, y_t) = \mathbb{E}_{D_{\theta_i} \sim \mathcal{D}}\big[\nabla_{\boldsymbol{x}} \ell_{D_{\theta_i}}(\boldsymbol{x}, y_t)\big]$. *Additionally, assume that the similarity of the gradient of* $\forall D_{\theta_i} \in \mathcal{D}$ *with the gradient of* $\bar{V}$ *is captured by* $\mathbb{E}_{D_{\theta_i} \sim \mathcal{D}}\big[\|\nabla_{\boldsymbol{x}} \ell_{D_{\theta_i}}(\boldsymbol{x}, y_t) - \nabla_{\boldsymbol{x}} \ell_{\bar{V}}(\boldsymbol{x}, y_t)\|_2^2\big] \leq \sigma^2$, *and* $\|\nabla_{\boldsymbol{x}} \ell_{D_{\theta_i}}(\boldsymbol{x}, y_t)\|_2 \leq B$. *Assume the loss function of a set of randomly picked accessible models* $D_{\theta_j} \in \mathcal{D}_s \subset \mathcal{D}$ *and the target model* $\bar{V}$ *are* $\beta$-*smooth, and* $\forall D_{\theta_j} \in \mathcal{D}_s$ *are* $(\alpha_j, D_{\theta_j})$-*effective on the generated samples with a perturbation constraint* $\|\boldsymbol{\delta}\|_2 \leq \epsilon'$. *Under these conditions, the probability of transferability can be lower bounded by:*

$$\Pr(T_r(\mathcal{D}_s, \bar{V}, \boldsymbol{x}^{adv}, y_t) = 1) \geq$$

$$1 - A - \frac{\epsilon'(1 + A) + c_{\mathcal{D}_s}(1 - A)}{c_v + \epsilon'} - \frac{\epsilon'}{c_v + \epsilon'} \sqrt{2\Big(1 - \frac{\|\nabla_{\boldsymbol{x}} \ell_{\bar{V}}(\boldsymbol{x}, y_t)\|_2 - \frac{\sigma}{\sqrt{|\mathcal{D}_s|}}}{B}\Big)},$$

*where* $A = \sum_{i=0}^{|\mathcal{D}_s|} \alpha_j$, $\quad c_v := \min_{\boldsymbol{x} \in \mathcal{X}} \frac{\min_{y \in [L] - \{y_t\}} \ell_{\bar{V}}(\boldsymbol{x}^{adv}, y) - \ell_{\bar{V}}(\boldsymbol{x}, y_t) - \frac{\beta}{2}\epsilon'^2}{\|\nabla_{\boldsymbol{x}} \ell_{\bar{V}}(\boldsymbol{x}, y_t)\|_2}$, *and*

$$c_{\mathcal{D}_s} := \max_{\boldsymbol{x} \in \mathcal{X}} \frac{\big(\min_{y \in [L] - \{y_t\}} \frac{1}{|\mathcal{D}_s|} \sum_{D_{\theta_j} \in \mathcal{D}_s} \ell_{D_{\theta_j}}(\boldsymbol{x}^{adv}, y) - \frac{1}{|\mathcal{D}_s|} \sum_{D_{\theta_j} \in \mathcal{D}_s} \ell_{D_{\theta_j}}(\boldsymbol{x}, y_t) + \frac{\beta}{2}\epsilon'^2\big)}{\|\frac{1}{|\mathcal{D}_s|} \sum_{D_{\theta_j} \in \mathcal{D}_s} \nabla_{\boldsymbol{x}} \ell_{D_{\theta_j}}(\boldsymbol{x}, y_t)\|_2}.$$

*Here* $c_{\mathcal{D}_s}$ *is the average risk of the models in* $\mathcal{D}_s$ *and* $c_v$ *is the risk of the virtual victim model* $\bar{V}$.

The definitions of transferability ($T_r(.)$) and ($\alpha_j, D_{\theta_j}$)-effective attack are deferred to Appendix G. In theorem 1, the value of $A$ is sufficiently small as it is measured on the accessible models. Additionally, $c_v$ is also sufficiently small as it is scaled by $\|\nabla_{\boldsymbol{x}} \ell_{\bar{V}}\|_2$ Yang et al. (2021). Thus, $\Pr(T_r(\mathcal{D}_s, \bar{V}, \boldsymbol{x}^{adv}, y_t) = 1)$ takes the form $\xi - \zeta \sqrt{\kappa + \frac{\sigma}{B\sqrt{|\mathcal{D}_s|}}}$, where $\zeta$ and $\kappa$ are the positive constants, and $\xi$ depends on $|\mathcal{D}_s|$. In $\xi$, $A$ can be approximated as a constant for a limited $|\mathcal{D}_s|$; and $c_{\mathcal{D}_s}$, representing the average risk across $\forall D_{\theta_j} \in \mathcal{D}_s$, can also be treated as a constant. Hence, the term that mainly captures the impact of $|\mathcal{D}_s|$ on transferability is $\sigma / \sqrt{|\mathcal{D}_s|}$. According to this, the lower bound of transferability is positively correlated with the number of accessible models when $|\mathcal{D}_s|$ is small, and the rate of increase in transferability decays quickly and saturates as $|\mathcal{D}_s|$ grows, a similar trend as observed in Fig. 4a in the Appendix. However, with a sufficiently large number of models, as $\sigma / \sqrt{|\mathcal{D}_s|}$ approaches zero, the term $A = \sum_{i=0}^{|\mathcal{D}_s|} \alpha_j$ becomes dominant. This indicates that an optimal number of accessible models, $|\mathcal{D}_s|$, exists beyond which the lower bound of transferability first increases positively with $|\mathcal{D}_s|$ but then decreases once this threshold is exceeded. Nevertheless, if we redefine transferability simply as: $T_r(\mathcal{D}_s, \bar{V}, \boldsymbol{x}^{adv}, y_t) = (\hat{\bar{V}}(\boldsymbol{x}^{adv}) = y_t)$ that only focuses on if the crafted $\boldsymbol{x}^{adv}$ exploiting $\mathcal{D}_s$ successfully deceives the target model $\bar{V}$ (without the constraint of deceiving $\forall D_{\theta_j} \in \mathcal{D}_s$), $\xi$ can be approximated as independent of $|\mathcal{D}_s|$. Under this condition, transferability exhibits a purely positive correlation with $|\mathcal{D}_s|$. *We note that theoretical analysis is meant to offer guidance on how diversity impacts transferability, not a strict implementation blueprint.* In our theoretical analysis, we adopt the L2 norm primarily for its analytical convenience. The geometry of the L2 ball allows for smoother derivation of bounds, enabling gradient-alignment and smoothness-based arguments, which are more challenging to formulate under the $L_\infty$ constraint. Importantly, the two norms are related. For any input of dimension $d$, an $L_\infty$-bounded perturbation also satisfies an L2 bound: $\|\delta\|_2 \leq \sqrt{d} \cdot \|\delta\|_\infty$. This relationship ensures that our theoretical insights under the L2 setting can be interpreted or extended to the $L_\infty$ regime by substituting the corresponding bound. The detailed proof of theorem 1 along with upper bound of transferability are deferred to Appendix G.

# 6 CONCLUSION

In this work, we propose BAT, a generative framework that improves targeted adversarial transferability under single-surrogate constraints. BAT guides the generator using *core target samples*—derived from natural images, refined, or synthesized from noise—and aligns adversarial examples with these samples in both output and feature spaces using an ensemble of pruned discriminators. The framework can also incorporate diverse model architectures when available, further enhancing transferability. This confidence-aware alignment strategy enables BAT to produce highly transferable adversarial examples that generalize well across unseen models. Experimental results show consistent gains under *no domain shift* ($\mathcal{P} = \mathcal{Q}$) and *domain shift* ($\mathcal{P} \neq \mathcal{Q}$). Complementary theory provides lower/upper bounds on targeted transferability and explains how the ensemble size trades off with performance.

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

# Appendix

In this supplementary material, we provide additional details and analyses that support and extend the main findings of the paper:

- **Section A** describes the step-by-step procedures for (i) crafting high-confidence core target samples across the discriminator ensemble and (ii) generator training.

- **Section B** reviews additional related works on adversarial attacks and ensemble-based strategies.

- **Section C** presents extended experimental results, including evaluations on both standard and robust models under $\mathcal{P}=\mathcal{Q}$ and $\mathcal{P}\neq\mathcal{Q}$, along with ablations.

- **Section D** provides a trade-off analysis between transferability and training cost as a function of discriminator count.

- **Section E** explores the applicability of BAT to vision-language models.

- **Section F** demonstrates the generator architecture.

- **Section G** contains formal proofs for the theoretical results, including the transferability bounds discussed in Section 5.

- **Section H** discusses the limitations of proposed BAT and its broader impacts.

- **Section I** includes visualizations of adversarial examples and perturbations.

- **Section J** provides additional attention heatmaps across models and their pruned variants.

Code for reproducing BAT is included as supplementary material and will be released publicly.

## A    ALGORITHMS

In this section, we provide the detailed procedures for (i) generating confident core target samples across the discriminator ensemble and (ii) training the generator using the proposed BAT objective.

---

**Algorithm 1:** Crafted target sample

1 **Inputs:** Samples set $\tilde{\mathcal{T}}$, discriminators $\mathcal{D}_s$, target class $y_t$, iteration number $T_c$, learning rate $\alpha_c$.
2 **Output:** More confident target samples set $\hat{\mathcal{T}}$.
3 $\boldsymbol{\delta}_0 = \mathbf{0}, \quad \hat{\mathcal{T}} = \{\}$
4 **foreach** $\boldsymbol{x}_i^t \in \tilde{\mathcal{T}}$ **do**
5      **for** $m = 0 : T_c - 1$ **do**
6          $\boldsymbol{x}_i = \boldsymbol{x}_i^t + \boldsymbol{\delta}_m$
7          Loss: $L_{\mathcal{D}_s}(\boldsymbol{x}_i, y_t) = \sum_{D_{\theta_j} \in \mathcal{D}_s} \text{CE}(D_{\theta_j}(\boldsymbol{x}_i), y_t)$     /* CE: cross-entropy loss */
8          Obtain the gradient: $\nabla_{\boldsymbol{\delta}} L_{\mathcal{D}_s}(\boldsymbol{x}_i, y_t)$
9          Update $\boldsymbol{\delta}_m$: $\boldsymbol{\delta}_{m+1} = \boldsymbol{\delta}_m - \alpha_c * \nabla_{\boldsymbol{\delta}} L_{\mathcal{D}_s}(\boldsymbol{x}_i, y_t)$
10          Clip: $\boldsymbol{\delta}_{m+1} = \min(\max(\boldsymbol{x}_i^t + \boldsymbol{\delta}_{m+1}, 0), 1) - \boldsymbol{x}_i^t$
11      $\hat{\mathcal{T}} = \hat{\mathcal{T}} \cup \{\hat{\boldsymbol{x}}_i^t\}$, where $\hat{\boldsymbol{x}}_i^t = \boldsymbol{x}_i^t + \boldsymbol{\delta}_{T_c}$

---

**Algorithm 2:** Training Generator

1 **Inputs:** Source dataset $\mathcal{S}$, available surrogate model $\mathcal{D}_{\theta_0}$, target class $y_t$, iteration number $T$.
2 **Output:** Trained generator $\mathcal{G}_\Phi$.
3 Obtain ensemble of surrogate models $\mathcal{D}_s$ using Eq. 5.
4 $\mathcal{T}^* \leftarrow$ Core target samples exploiting $\mathcal{D}_s$.
5 **for** $\kappa = 0 : T - 1$ **do**
6      **foreach** *mini-batch* $\{\boldsymbol{x}_i^s\}_{i=1}^B, \boldsymbol{x}_i^s \sim \mathcal{S}$ **do**
7          Sample $B$ target samples: $\{\boldsymbol{x}_i^{t\star}\}_{i=1}^B, \boldsymbol{x}_i^{t\star} \sim \mathcal{T}^*$
8          Generate adv. examples using Eq. 8: $\{\boldsymbol{x}_i^{adv}\}_i, \ \forall \boldsymbol{x}_i^s \in \{\boldsymbol{x}_i^s\}_{i=1}^B$
9          Calculate loss $\mathcal{L}_\mathcal{G}$ using Eq. 9
10          Update parameters of $\mathcal{G}_\Phi$: $\Phi \leftarrow \min \mathcal{L}_\mathcal{G}$

---

# B RELATED WORK

**Untargeted transferable attacks.** Untargeted adversarial attacks primarily utilize I-FGSM (Kurakin et al., 2018), an iterative method, which iteratively adds perturbations in the direction of the gradient w.r.t. input to craft adversarial examples. To escape local minima and enhance transferability, MI-FGSM (Dong et al., 2018) introduces momentum-based optimization. Further improvements in transferability have been achieved with more advanced momentum-based attacks such as NI-FGSM (Lin et al., 2019), VMI-FGSM (Wang & He, 2021), GRA (Zhu et al., 2023) and so on. Additionally, several works employ input transformation techniques to mitigate the over-fitting problem on surrogate models. For instance, Diverse input method (DIM) (Xie et al., 2019) randomly resizes and adds padding to input samples; Time invariant method (TIM) (Dong et al., 2019) adopts a Gaussian kernel to smooth the gradient before updating the perturbation; Scale invariant method (SIM) (Lin et al., 2019) uses multiple scaled versions of the input to calculate the gradient; Admix (Wang et al., 2021) extends SIM by incorporating small portions of images from other categories; Block shuffle and rotation (BSR) (Wang et al., 2024b) divides the input image into blocks and calculates the gradient from a set of images obtained by randomly shuffling and rotating these blocks. Additionally, some works enhance adversarial attacks by augmenting images with multiple transformations predicted by a neural network. Automatic Model Augmentation (AutoMA) (Yuan et al., 2021) adopts a Proximal Policy Optimization algorithm to find a strong policy. The Transformation-enhanced Transfer Attack (ATTA) (Wu et al., 2021) trains an adversarial transformation network to capture the most harmful distortions. Learning to Transform (L2T) (Zhu et al., 2024) identifies the optimal combination of transformations to increase adversarial transferability.

**Targeted transferable attacks.** The untargeted attacks can be modified to craft targeted adversarial examples; however, they show limited transferability. Consequently, a number of recent works are dedicated to developing new methods to generate targeted adversarial examples. To enhance the targeted transferability, (Inkawhich et al., 2019) optimizes the loss in feature space to improve the feature similarity between source images and target images. Po-Trip (Li et al., 2020a) introduces Poincare loss and Triplet loss, with the former designed to alleviate noise curing and the latter to push the adversarial image from the source class to the target class. Moreover, (Zhao et al., 2021) identifies that using simple logit loss, rather than cross-entropy loss, enhances targeted transferability. SU (Wei et al., 2023) improves targeted transferability by incorporating feature similarity loss between the source image and different local region within the source image. Additionally, auxiliary neural networks are trained to learn the intermediate feature distribution of the target class considering features from single or multiple layers in (Inkawhich et al., 2020a;b). SASD-WS (Wu et al., 2024) enhances the generalization capability of the surrogate model by fine-tuning it, assuming full access to the surrogate model's training dataset.

Generative approaches have demonstrated leading targeted transferability. TTP (Naseer et al., 2021) trains a generator to craft adversarial examples to align the output distribution of the source and target domain obtained from the surrogate model. TTAA (Wang et al., 2023) improves over TTP by additionally training a feature discriminator to capture and align the feature distribution of the source and target images, M3D (Zhao et al., 2023) trains the generator by leveraging two discriminators, both derived from a single surrogate model, to simultaneously maximize the discrepancy between their decision boundaries during generator training to improve transferability to unknown models. Furthermore, ESMA (Gao et al., 2024) and CGNC (Fang et al., 2024) train generators to generate adversarial examples for multiple target classes. However, these methods often exhibit limited transferability across models. To address this, CGNC enhances transferability by fine-tuning the pretrained generator specifically for each target class.

**Ensemble-based transferable attacks.** The transferability of adversarial examples can be enhanced by leveraging an ensemble of surrogates (Liu et al., 2016). The iterative attack in (Liu et al., 2016) improves transferability by accumulating losses, while (Dong et al., 2019) incorporates both logits and losses of the ensemble. (Cai et al., 2022) further refines this by taking a weighted average of ensemble losses, where the weights are optimized through queries to the target model. Recognizing the variance among ensemble models, (Xiong et al., 2022) proposed a stochastic variance-reduced ensemble (SVRE) attack for better generalization, whereas (Chen et al., 2023a) adaptively ensembles model outputs via the adaptive gradient modulation (AGM) strategy. Additionally, (Chen et al., 2023b) introduced an iterative attack targeting common weak regions across the ensemble. While surrogate ensembles significantly boost attack success rates, their effectiveness extends beyond classification tasks (Chen et al., 2023a; Huang et al., 2023). Beyond standard ensembles, self-ensembling strategies

Table 8: TSR(%) comparison of the proposed BAT variants with the baselines, under $\mathcal{P}=\mathcal{Q}$, against the target model VGG19$_{BN}$ with different input processing-based defenses, including a set of image smoothing techniques (Gaussian, Median, and Average), JPEG compression with different quality factors (Q=70, Q=80, Q=90), and various data augmentation methods: Resize and Crop (R&C), Horizontal Flip (HF), and Rotation by 30°. $D_{\theta_0}$ represents the surrogate model used to train the generator. The best overall method is highlighted in bold, while the best baseline method is underlined. Values in parentheses indicate the improvement by BATs in TSR(%) over the best baseline.

| $D_{\theta_0}$ | Attack | Without | Smoothing | | | JPEG compression | | | Data Augmentation Methods | | |
|---|---|---|---|---|---|---|---|---|---|---|---|
| | | Defense | Gaussian | Median | Average | Q=70 | Q=80 | Q=90 | R&C | HF | Rotate(30⁰) |
| RN50 | ESMA | 67.16 | 48.39 | 55.32 | 36.41 | 40.43 | 48.02 | 56.42 | 15.38 | 33.80 | 11.77 |
| | TTP | 71.10 | 62.86 | 67.64 | 53.65 | 58.39 | 61.58 | 64.78 | 16.05 | 40.02 | 12.92 |
| | CGNC$_{FT}$ | 81.36 | _77.59_ | _80.17_ | _69.13_ | _69.24_ | _73.21_ | _77.24_ | _18.02_ | 42.53 | _15.48_ |
| | M3D | _83.38_ | 67.09 | 72.58 | 56.71 | 62.93 | 68.71 | 75.13 | 17.94 | _42.55_ | 11.31 |
| | BAT-BS | 89.71$_{(+6.33)}$ | 79.75$_{(+2.16)}$ | 84.16$_{(+3.99)}$ | 72.42$_{(+3.29)}$ | 78.48$_{(+9.24)}$ | 81.63$_{(+8.42)}$ | 85.27$_{(+8.03)}$ | 19.13$_{(+1.11)}$ | 43.71$_{(+1.16)}$ | 16.86$_{(+1.38)}$ |
| | BAT-CS | **93.97**$_{(+10.59)}$ | **84.98**$_{(+7.39)}$ | **87.22**$_{(+7.05)}$ | **77.43**$_{(+8.30)}$ | **84.00**$_{(+14.76)}$ | **86.60**$_{(+13.39)}$ | **89.71**$_{(+12.47)}$ | **23.50**$_{(+5.48)}$ | 52.35$_{(+9.80)}$ | **20.53**$_{(+5.05)}$ |
| | BAT-CN | 92.13$_{(+8.75)}$ | 83.07$_{(+5.48)}$ | 85.68$_{(+5.51)}$ | 75.88$_{(+6.75)}$ | 81.04$_{(+11.80)}$ | 84.28$_{(+11.07)}$ | 87.84$_{(+10.60)}$ | 21.37$_{(+3.35)}$ | **52.78**$_{(+10.23)}$ | 18.57$_{(+3.09)}$ |
| DN121 | ESMA | 61.23 | 50.98 | 59.65 | 42.40 | 39.90 | 45.60 | 50.95 | 11.47 | 30.73 | 11.57 |
| | TTP | 62.57 | 53.69 | 57.59 | 48.73 | 50.00 | 52.55 | 55.75 | 11.71 | 33.44 | 11.74 |
| | CGNC$_{FT}$ | _81.54_ | _71.27_ | _75.56_ | _65.08_ | _67.67_ | _70.28_ | _75.63_ | _17.44_ | _44.78_ | _14.68_ |
| | M3D | 79.24 | 63.14 | 70.71 | 54.40 | 57.66 | 63.33 | 70.03 | 16.24 | 41.66 | 11.33 |
| | BAT-BS | 82.66$_{(+1.12)}$ | 72.96$_{(+1.69)}$ | 76.33$_{(+0.77)}$ | 67.53$_{(+2.45)}$ | 68.93$_{(+1.26)}$ | 71.53$_{(+1.25)}$ | 76.11$_{(+0.48)}$ | 17.45$_{(+0.01)}$ | 46.63$_{(+1.85)}$ | 14.68$_{(0.00)}$ |
| | BAT-CS | **89.62**$_{(+8.08)}$ | **82.15**$_{(+10.88)}$ | **84.74**$_{(+9.18)}$ | **76.96**$_{(+11.88)}$ | **80.69**$_{(+13.02)}$ | **82.70**$_{(+12.42)}$ | **85.49**$_{(+9.86)}$ | **25.68**$_{(+8.24)}$ | **48.72**$_{(+3.94)}$ | **19.98**$_{(+5.30)}$ |
| | BAT-CN | 88.45$_{(+6.91)}$ | 79.87$_{(+8.60)}$ | 82.47$_{(+6.91)}$ | 74.63$_{(+9.55)}$ | 78.91$_{(+11.24)}$ | 81.23$_{(+10.95)}$ | 84.46$_{(+8.83)}$ | 22.57$_{(+5.13)}$ | 46.84$_{(+2.06)}$ | 18.73$_{(+4.05)}$ |

Table 9: TSR(%) comparison of the proposed BAT variants with the SOTA generative methods, under $\mathcal{P}=\mathcal{Q}$, considering DenseNet121 as surrogate model, against classifiers with robust training mechanism on ImageNet.

| Surrogate | $\epsilon$ | Attack | Inc-v3$_{adv}$ | IR-v2$_{ens}$ | RN50$_{SIN}$ | RN50$_{IN}$ | RN50$_{fine}$ | RN50$_{Aux}$ |
|---|---|---|---|---|---|---|---|---|
| DN121 | $\frac{16}{255}$ | ESMA | 1.30 | 1.38 | 18.37 | 58.19 | 61.50 | 47.76 |
| | | TTP | 4.69 | 5.98 | 13.85 | 53.05 | 56.64 | 49.92 |
| | | M3D | 5.37 | 6.80 | 38.28 | 77.73 | 83.02 | 71.41 |
| | | CGNC$_{FT}$ | 7.33 | 8.68 | 18.95 | 73.62 | 79.77 | 63.75 |
| | | BAT-BS | 7.47 | 11.44 | 38.03 | 81.52 | 81.70 | 73.72 |
| | | BAT-CS | **9.96** | **13.88** | **44.66** | 85.41 | 84.15 | 80.52 |
| | | BAT-CN | 7.28 | 13.69 | 41.80 | **86.88** | **85.53** | **81.25** |
| | $\frac{32}{255}$ | ESMA | 12.85 | 19.41 | 31.58 | 69.35 | 72.19 | 61.34 |
| | | TTP | 19.59 | 23.71 | 25.05 | 59.22 | 57.92 | 51.49 |
| | | M3D | 27.46 | 35.13 | 54.13 | 84.26 | 84.19 | 81.60 |
| | | CGNC$_{FT}$ | 28.69 | 38.18 | 42.68 | 85.82 | 83.95 | 83.29 |
| | | BAT-BS | 30.27 | 43.49 | 50.66 | 85.30 | 82.69 | 76.73 |
| | | BAT-CS | **36.84** | **50.08** | **58.48** | 89.82 | **86.73** | 84.14 |
| | | BAT-CN | 29.42 | 51.62 | 57.77 | **90.15** | 84.88 | **85.01** |

such as dropout and skip connections have been explored in (Li et al., 2020b). Furthermore, the generative attack TTP (Naseer et al., 2021) demonstrates that replacing a single surrogate with an ensemble can substantially improve attack performance.

## C  ADDITIONAL EXPERIMENTAL RESULTS

In this section, we present a comprehensive set of additional experiments to further analyze and validate the effectiveness of BAT. We evaluate the robustness of BAT variants against various input-processing defenses and adversarially trained target models, using both ResNet50 and DenseNet121 as surrogates. We also investigate the impact of reduced perturbation budgets on targeted transferability.

Beyond robustness, we showcase BAT's ability to generate highly confident adversarial examples, thereby improving transferability. We further analyze the stability of BAT under different pruned ensembles and explore the effect of key design choices, including the pruning ratio, the number of core target samples used during training, the number of discriminators ($|\mathcal{D}_s|$), and the parameter $\lambda$. These analyses offer deeper insights into the generalization, scalability, and robustness of the BAT framework across varying conditions.

**Robustness against input-processing defense.**  We evaluate the performance of the proposed BAT variants against a target model employing various input-processing-based defenses. These defenses include smoothing techniques (Ding et al., 2019) such as Gaussian, Median, and Average filters; the JPEG compression (Dziugaite et al., 2016) algorithm; and several data augmentation techniques. For JPEG compression, we explore different quality factors (Q = 70, 80, and 90), where a higher Q value corresponds to less compression. The data augmentation techniques include Resize and Crop

Table 10: Average prediction probability of the target class for the generated adversarial examples from 2,000 ImageNet validation images across various target classifiers under $\mathcal{P}=\mathcal{Q}$. '*' indicates the performance on the white-box surrogate model ($D_{\theta_0}$). The BAT variants, specifically BAT-CS and BAT-CN, generate more confident adversarial examples by learning to generate samples targeting the high-confidence region across discriminators. For each target model, the best overall method is highlighted in bold, while the best baseline method is underlined. Values in parentheses indicate the improvement in prediction probability over the best baseline.

| $D_{\theta_0}$ | Attack | RN18 | RN50 | RN101 | DN121 | DN161 | VGG16$_{BN}$ | VGG19$_{BN}$ | MN-V2 | ViT-B | Average |
|---|---|---|---|---|---|---|---|---|---|---|---|
| RN50 | ESMA | 0.459 | 0.884* | 0.595 | 0.541 | 0.560 | 0.611 | 0.583 | 0.419 | 0.150 | 0.533 |
| | TTP | 0.580 | 0.795* | 0.660 | 0.557 | 0.577 | 0.620 | 0.523 | 0.443 | 0.149 | 0.545 |
| | CGNC | 0.667 | 0.901* | 0.779 | 0.725 | 0.767 | 0.697 | 0.639 | 0.513 | 0.179 | 0.652 |
| | CGNC$_{FT}$ | 0.769 | 0.930* | 0.863 | 0.817 | 0.802 | 0.793 | 0.747 | 0.630 | 0.243 | 0.733 |
| | M3D | 0.728 | 0.899* | 0.797 | 0.770 | 0.791 | 0.703 | 0.696 | 0.657 | **0.346** | 0.710 |
| | BAT-BS | 0.794 | 0.934* | 0.846 | 0.792 | 0.811 | 0.819 | 0.778 | 0.673 | 0.283 | 0.748$_{(+0.015)}$ |
| | BAT-CS | **0.859** | **0.962*** | **0.897** | **0.856** | **0.854** | **0.867** | **0.867** | **0.742** | 0.335 | **0.804**$_{(+0.072)}$ |
| | BAT-CN | 0.840 | 0.961* | 0.888 | 0.853 | 0.847 | 0.853 | 0.844 | 0.723 | 0.319 | 0.792$_{(+0.059)}$ |
| DN121 | ESMA | 0.506 | 0.566 | 0.477 | 0.883* | 0.689 | 0.557 | 0.502 | 0.362 | 0.135 | 0.520 |
| | TTP | 0.488 | 0.486 | 0.495 | 0.790* | 0.533 | 0.499 | 0.459 | 0.353 | 0.149 | 0.472 |
| | CGNC | 0.601 | 0.632 | 0.561 | 0.953* | 0.749 | 0.608 | 0.601 | 0.421 | 0.165 | 0.588 |
| | CGNC$_{FT}$ | 0.724 | 0.743 | 0.722 | 0.954* | 0.775 | 0.727 | 0.728 | 0.565 | 0.230 | 0.685 |
| | M3D | 0.694 | 0.729 | 0.704 | 0.923* | 0.803 | 0.673 | 0.666 | 0.603 | **0.319** | 0.679 |
| | BAT-BS | 0.740 | 0.735 | 0.729 | 0.948* | 0.798 | 0.745 | 0.736 | 0.593 | 0.270 | 0.699$_{(+0.014)}$ |
| | BAT-CS | **0.840** | **0.843** | **0.830** | 0.971* | 0.847 | **0.814** | **0.797** | **0.682** | 0.312 | **0.771**$_{(+0.085)}$ |
| | BAT-CN | 0.829 | 0.834 | 0.821 | **0.973*** | **0.874** | **0.814** | 0.783 | 0.648 | 0.312 | 0.765$_{(+0.080)}$ |

(R&C), which resizes each input image from $3 \times 224 \times 224$ to $3 \times 256 \times 256$, then crops it back to $3 \times 224 \times 224$, Horizontal Flip (HF), and a $30°$ rotation of the input images to the target model.

To assess performance, we generate adversarial examples form the trained generators under $\mathcal{P}=\mathcal{Q}$ using 2,000 randomly selected ImageNet validation images. We then compare the transferability of the generated adversarial examples—generated by the proposed BAT variants and baseline methods—to the unknown target model VGG19$_{BN}$, employing aforementioned defenses. As shown in Tab. 8, all attacks exhibit a reduced transfer success rate (TSR) when input-processing defenses are applied to VGG19$_{BN}$, compared to the scenario without such defenses. This decrease in TSR can be attributed to the information loss caused by the defenses. Among the input-processing defenses, R&C and rotation are particularly effective, as they remove more information from the input, which can also result in a loss of normal accuracy. Despite these challenges, our proposed BAT variants, specifically BAT-CS and BAT-CN, outperform all baselines by a significant margin.

**Performance against robust models.** In the main text in Tab. 4, we compare TSR of the generative methods, considering ResNet50 as the surrogate model, against six robust-trained models. Here, we extend the evaluation by analyzing the TSR of the generators trained with different methods considering DenseNet121 (DN121) as the model accessible to the adversary. The results are demonstrated in Tab. 9. From these results, a similar trend has been observed, and our proposed BAT variants continue to demonstrate better performance over baseline attacks.

**Confidence of adversarial examples.** We examine the prediction probability for the target class of the generated adversarial examples from 2,000 ImageNet validation images across the surrogate model and various unknown target models. As shown in Tab. 10, adversarial examples generated by the proposed BAT variants achieve significantly higher average confidence on the target class across various target models compared to baseline methods. Specifically, as BAT-CS and BAT-CN train generators to minimize the distribution distance between the generated adversarial examples and the core target samples across discriminators (discussed in Section 3.2), the generators are capable of generating adversarial examples that are more confidently classified towards the target class. Hence, the generated adversarial examples using the proposed BAT variants demonstrate higher transferability to the unknown target models.

**More analysis under domain shift.** In Tab. 2 (main text), we report results for the $\mathcal{P}\neq\mathcal{Q}$ setting where the generator is trained on *Painting* ($\mathcal{P}$) while both the accessible surrogate and the target models are trained on *ImageNet-1K* ($\mathcal{Q}$); evaluation there uses *Painting* test images.

Table 11: TSR(%) of various attacks on different target classifiers under $\mathcal{P} \neq \mathcal{Q}$ where source images for training the generators are sampled from the Painting dataset, and target models are pretrained on ImageNet-1K. The BAT variants, specifically BAT-BS and BAT-CS outperform the baselines applicable for domain shift by a notable margin, as evaluated on the 5,000 images from the ImageNet validation set. This demonstrates that, despite being trained on the Painting dataset, the generator can effectively craft adversarial examples of the images in the domain of target class training dataset. '*' denotes the performance on the white-box surrogate model ($D_{\theta_0}$). For each target model, the best overall method is highlighted in bold, while the best baseline method is underlined. Values in parentheses indicate the improvement in TSR(%) over the best baseline.

| $D_{\theta_0}$ | Attack | RN18 | RN50 | RN101 | DN121 | DN161 | VGG16$_{BN}$ | VGG19$_{BN}$ | MN-V2 | ViT-B | Avg. |
|---|---|---|---|---|---|---|---|---|---|---|---|
| RN50 | TTP | 62.27 | 87.88* | 62.91 | 68.21 | 63.09 | 65.97 | 57.39 | 47.26 | 16.02 | 59.00 |
| | CGNC | 79.29 | 96.30* | 85.30 | 83.63 | 84.74 | 81.20 | 75.62 | 65.50 | 24.80 | 75.15 |
| | CGNC$_{FT}$ | 86.70 | 97.82* | 91.85 | 90.56 | 90.83 | 88.31 | 84.31 | 76.44 | 35.32 | 82.46 |
| | BAT-BS | 87.53 | 98.10* | 91.81 | 91.15 | 89.70 | 88.44 | 85.74 | 76.50 | 39.01 | 83.11$_{(+0.65)}$ |
| | BAT-CS | 89.64 | 98.27* | 91.85 | 93.32 | 91.96 | 90.93 | 89.58 | 83.07 | 43.37 | 85.78$_{(+3.32)}$ |
| | BAT-CN | 90.69 | 98.17* | 91.43 | 93.00 | 90.91 | 91.05 | 89.92 | 80.92 | 42.8 | 85.43$_{(+2.97)}$ |
| DN121 | TTP | 51.98 | 51.67 | 47.84 | 89.83* | 63.15 | 53.38 | 45.99 | 39.10 | 12.24 | 50.58 |
| | CGNC | 66.21 | 78.29 | 67.08 | 91.82* | 71.53 | 64.32 | 62.03 | 48.78 | 20.94 | 63.44 |
| | CGNC$_{FT}$ | 85.02 | 85.19 | 80.28 | 98.73* | 91.49 | 84.72 | 81.48 | 70.14 | 34.66 | 79.08 |
| | BAT-BS | 87.29 | 84.54 | 82.29 | 98.34* | 87.47 | 82.04 | 80.91 | 73.99 | 40.52 | 79.82$_{(+0.74)}$ |
| | BAT-CS | 88.57 | 88.81 | 85.46 | 98.73* | 92.06 | 88.24 | 87.19 | 74.34 | 45.41 | 83.20$_{(+4.12)}$ |
| | BAT-CN | 88.80 | 89.03 | 84.80 | 98.61* | 92.09 | 87.33 | 85.88 | 74.45 | 44.34 | 82.81$_{(+3.73)}$ |

Table 12: TSR (%) on ImageNet-1K under the $\mathcal{P} \neq \mathcal{Q}$, using ResNet-50 as the surrogate (marked with *), where the source images are sampled from AnimeFace dataset, and the target models are pretrained on ImageNet-1K. BAT-CS consistently outperforms CGNC-FT under domain-shift across both CNN and transformer victims while preserving strong white-box performance on the surrogate.

| Attack | RN18 | RN50* | RN101 | DN121 | DN161 | VGG16-BN | VGG19-BN | MN-V2 | ViT-B |
|---|---|---|---|---|---|---|---|---|---|
| CGNC-FT | 88.06 | 97.97* | 92.51 | 90.93 | 90.68 | 89.01 | 85.07 | 81.30 | 54.73 |
| BAT-CS | 92.21 | 98.92* | 94.77 | 94.17 | 94.57 | 93.11 | 92.17 | 88.91 | 64.09 |

Here, we extend this analysis by keeping the same generators trained on *Painting* ($\mathcal{P}$) but evaluating on *ImageNet-1K* validation images (5,000 from $\mathcal{Q}$). Tab. 11 compares BAT variants with baselines under this protocol.

The results show that BAT remains competitive under this shift of evaluation seeds from $\mathcal{P} \rightarrow \mathcal{Q}$: targeted success rates decrease only modestly relative to the *Painting*-seed evaluation, yet **BAT-CS** and **BAT-CN** continue to rank among the top performers. This indicates that BAT-trained generators—guided by the frozen, $\mathcal{Q}$-trained discriminator ensemble—generalize beyond the source training domain, producing adversarial examples that transfer to images drawn from the models' training domain $\mathcal{Q}$.

In addition to the ImageNet–Painting setup for the domain shift analysis, we additionally evaluate BAT under a second, substantially different domain shift using the AnimeFace dataset (Branwen et al., 2019). Using ResNet-50 as the surrogate, in Tab. 12, we compare BAT-CS with the strong generative baseline CGNC-FT and report TSR on ImageNet-1K validation images. These results, together with the ImageNet–Painting experiments, provide additional evidence that BAT remains effective across distinct domain-shift settings.

Table 13: TSR (%) on ImageNet-1K using ViT-B/16 as the surrogate. Asterisk (*) marks the surrogate model. BAT-CS consistently improves over M3D on both CNN and transformer victims, while preserving strong white-box performance on ViT-B.

| Attack | VGG19-BN | RN50 | DN121 | MN-V2 | ViT-B* | Swin-B | DeiT-B |
|---|---|---|---|---|---|---|---|
| M3D | 46.62 | 63.52 | 65.55 | 52.67 | 97.43* | 57.15 | 76.42 |
| BAT-CS | **50.12** | **66.91** | **68.20** | **56.76** | 97.83* | **59.77** | **78.39** |

**ViT surrogate: cross-family transferability.** To further assess cross-architecture transferability and verify that BAT is not tied to convolutional surrogates, we also consider a transformer surrogate.

Table 14: TSR (%) on ImageNet-1K under an $\ell_2$ constraint using generators trained under the $\ell_\infty$ threat model (no retraining). All methods use ResNet-50 as the surrogate. BAT-CS consistently outperforms M3D on most victims and achieves a higher average TSR under the $\ell_2$ constraint.

| Attack | RN18 | RN50 | RN101 | DN121 | DN161 | VGG16-BN | VGG19-BN | MN-V2 | ViT-B/16 | Avg. |
|--------|------|------|-------|-------|-------|----------|----------|-------|----------|------|
| M3D | 82.19 | 95.48 | 85.28 | 82.85 | 80.67 | 79.05 | 80.10 | 78.72 | 50.12 | 79.38 |
| BAT-CS | **91.46** | **97.74** | **92.03** | **90.59** | **91.24** | **93.00** | **91.81** | **86.81** | 48.93 | **87.07** |

Table 15: TSR(%) of various attacks on different target classifiers under $\mathcal{P}=\mathcal{Q}$ for varying perturbation budgets $\epsilon$ with ResNet50 as surrogate.

| $\epsilon$ | Attack | RN18 | RN50 | RN101 | DN121 | DN161 | VGG16$_{BN}$ | VGG19$_{BN}$ | MN-V2 | ViT-B | Average |
|-----|--------|------|------|-------|-------|-------|----------|----------|-------|-------|---------|
| $\frac{12}{255}$ | TTP | 65.92 | 91.65 | 69.95 | 71.12 | 63.38 | 66.7 | 66.48 | 52.77 | 12.31 | 62.25 |
| | M3D | 76.49 | 91.98 | 79.72 | 73.37 | 73.41 | 79.15 | 76.61 | 70.69 | 30.96 | 72.49 |
| | CGNC$_{FT}$ | 67.01 | 91.19 | 76.37 | 72.09 | 72.19 | 75.36 | 70.28 | 59.89 | 27.33 | 67.97 |
| | BAT-BS | 84.06 | 96.82 | 87.22 | 87.88 | 84.62 | 86.71 | 84.94 | 73.47 | 25.18 | 78.99 |
| | BAT-CS | **88.18** | **97.67** | **91.25** | **90.23** | **89.42** | **89.81** | **88.71** | **79.08** | **31.13** | **82.83** |
| | BAT-CN | 87.64 | 97.58 | 90.45 | 89.25 | 87.97 | 89.48 | 87.99 | 78.18 | 28.38 | 81.88 |
| $\frac{8}{255}$ | TTP | 30.43 | 69.92 | 36.36 | 43.4 | 37.48 | 33.89 | 37.52 | 20.85 | 2.68 | 34.73 |
| | M3D | 37.24 | 68.3 | 42.32 | 40.36 | 39.28 | 41.21 | 38.24 | 33.09 | 7.06 | 38.57 |
| | CGNC$_{FT}$ | 17.83 | 47.39 | 21.93 | 23.55 | 26.47 | 29.18 | 26.38 | 12.27 | 2.90 | 23.10 |
| | BAT-BS | 53.95 | 85.6 | 59.29 | 62.55 | 58.83 | 57.9 | 59.08 | 38.68 | 6.27 | 53.57 |
| | BAT-CS | **58.34** | **88.23** | **65.46** | **66.7** | **64.44** | **62.97** | **63.93** | **44.88** | **7.37** | **58.04** |
| | BAT-CN | 58.87 | 88.09 | 63.79 | 65.34 | 63.58 | 62.59 | 63.12 | 43.93 | 6.79 | 57.34 |

Specifically, we use ViT-B/16 as the surrogate and evaluate targeted transferability to both CNN and transformer victims on ImageNet-1K. The training and evaluation protocols are identical to those used for CNN surrogates in the main text.

As shown in Tab. 13, BAT-CS improves over M3D on all CNN victims and also on the transformer victims Swin-B and DeiT-B, while maintaining very high white-box TSR on the ViT-B surrogate. The overall average TSR increases from 65.62% for M3D to 68.28% for BAT-CS (and from 60.32% to 63.36% if the surrogate is excluded), indicating that the pruning-based ensemble and core-target training mechanism generalize across CNN and ViT families rather than being specific to a particular architecture type.

$\ell_2$ **evaluation without retraining** To verify that the observed trends are not specific to the $\ell_\infty$ threat model, we also evaluate BAT-CS and M3D under an $\ell_2$ constraint without retraining the generators. Concretely, we reuse the generators trained with $\epsilon_\infty = 16/255$ and, at test time, rescale the perturbation $\boldsymbol{\delta}$ for each source image $\boldsymbol{x}$ to have $\|\boldsymbol{\delta}\|_2 = \epsilon_2$ and form $\boldsymbol{x}_{\mathrm{adv}} = \boldsymbol{x} + \epsilon_2 \boldsymbol{\delta}/\|\boldsymbol{\delta}\|_2$ (with $\epsilon_2 = 16$). Tab. 14 reports the TSR (%) on ImageNet-1K for the same set of victim models as in the main text. BAT-CS maintains a clear advantage over M3D under the $\ell_2$ constraint, increasing the average TSR from 79.38% to 87.07%, which supports our claim that the proposed alignment mechanism yields perturbations that remain effective across norms.

**Impact of reduced perturbation budget.** While the default perturbation budget is set to $\epsilon = 16/255$ to evaluate the performance of BAT variants, we further examine the effect of lower budgets, considering $\epsilon = 12/255$ and $\epsilon = 8/255$. Using ResNet50 as the surrogate model, we observe from Tab. 1 and Tab. 15 that TSR declines as the perturbation budget decreases. However, BAT methods consistently achieve significantly higher TSR than the generative baselines, even under reduced perturbation, demonstrating their strong generalization capability.

Table 16: TSR (%) on ImageNet-1K under 5 random seeds, reporting mean ± standard deviation across seeds for M3D and BAT-BS. Both methods use ResNet-50 as the surrogate and are evaluated in the no-domain-shift setting with the same seed set. BAT-BS consistently improves over M3D on most victim models while exhibiting comparable variability across seeds.

| Attack | RN18 | RN50 | RN101 | DN121 | DN161 | VGG16$_{BN}$ | VGG19$_{BN}$ | MN-V2 | ViT-B/16 |
|--------|------|------|-------|-------|-------|----------|----------|-------|----------|
| M3D | 86.52 ± 1.75 | 95.79 ± 0.22 | 88.63 ± 0.92 | 88.67 ± 1.43 | 89.03 ± 0.99 | 85.32 ± 1.22 | 84.15 ± 0.95 | 81.54 ± 1.59 | 49.69 ± 1.38 |
| BAT-BS | 89.49 ± 2.18 | 98.50 ± 0.25 | 94.61 ± 1.15 | 93.35 ± 0.71 | 91.81 ± 1.42 | 91.39 ± 1.86 | 89.09 ± 1.03 | 88.10 ± 3.64 | 40.61 ± 1.60 |

**Multi-seed stability.** We examine the robustness of BAT to randomness by comparing BAT-BS to the strongest generative baseline, M3D, under the same seed conditions. For both methods, we run 5 independent experiments with different random seeds (affecting pruning and generator initialization for BAT-BS and generator initialization for M3D), using ResNet-50 as the surrogate and the no-domain-shift ImageNet-1K setting. Tab. 16 reports the mean and standard deviation of TSR (%) across seeds for each victim model. The standard deviations are small for both methods (typically $\leq 2\%$–$3\%$), indicating that the reported gains are not due to a single lucky run. Across almost all CNN victims and MobileNet-V2, BAT-BS achieves consistently higher mean TSR than M3D while exhibiting comparable variability, and only underperforms M3D on ViT-B/16. Overall, these results show that, under the given pruning ratio, BAT-BS provides a stable improvement over M3D across random seeds, supporting the claim that the benefits of the pruning-based ensemble and core-target training are robust to randomness.

**Impact of target sample's size.** We investigate the effect of the number of target samples used to train the generator by the BAT-BS method on the TSR. For this analysis, we consider the *no domain shift* scenario and employ ResNet50 as a surrogate model, pretrained on the ImageNet dataset. To train the generator for a specific target class, we begin by sorting approximately 1,300 target samples based on their average confidence scores across the discriminators. Starting with the top 100 most confident samples, we gradually increase the number of samples to assess the TSR at different levels.

As shown in Fig. 5, we obtain maximum TSR at around 85.5% using 300 target samples. However, as the number of target samples increases beyond 300, the TSR gradually declines. Based

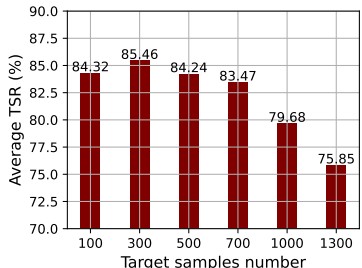

Figure 5: TSR(%) variation, under $\mathcal{P}=\mathcal{Q}$, of the adversarial examples generated from the trained generator using the BAT-BS method with different number of target samples to guide the generator training, leveraging ResNet50 as a surrogate.

on these observations, we select 300 target samples for training all proposed BAT variants in our experiments to ensure better performance.

## D   TRAINING TIME AND TRADEOFF ANALYSIS

In this section, we examine the time required to train the generator using our proposed BAT method, which utilizes multiple discriminators (five in the default setting). We also analyze the tradeoff between the TSR and training time complexity.

In a single iteration, let the time complexity of a single discriminator and the generator be $\mathcal{O}(D_\theta)$ and $\mathcal{O}(\mathcal{G}_\Phi)$, respectively. If $v$ is the total number of iterations per epoch and $T$ denotes the number of epochs for generator training, the total complexity for the BAT method with a single discriminator is $\mathcal{O}(vT(\mathcal{G}_\Phi + D_\theta)) = vT\mathcal{O}(\mathcal{G}_\Phi) + vT\mathcal{O}(D_\theta)$. BAT uses an ensemble of discriminators derived from a single surrogate model, so all discriminators have the same architecture and time complexity. Thus, with $K = |\mathcal{D}_s|$ discriminators, the total complexity becomes $\mathcal{O}(vT(\mathcal{G}_\Phi + KD_\theta)) = vT\mathcal{O}(\mathcal{G}_\Phi) + vTK\mathcal{O}(D_\theta)$.

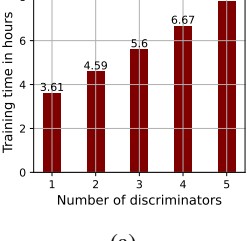

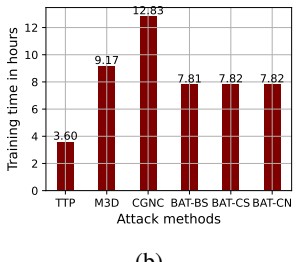

(a)

(b)

Figure 6: (a) Training time per target class (in hours) required to train a generator for proposed BAT-BS with varying number of discriminators; (b) Training time per target class (in hours) required to train a generator for different generative methods.

This linear increase in training time with the number of discriminators suggests higher computational costs with added discriminators. Empirically, we measure the training time per target class for BAT-BS (a BAT variant). Fig. 6a illustrates that the training time per target class increases approximately

linearly with the number of discriminators, ranging from 3.61 hours with one discriminator to 7.81 hours with five discriminators. This trend indicates that adding more discriminators incurs higher computational costs. The training time of the other variants of BAT is very similar: crafting 300 core target samples for BAT-CS/CN with 25 PGD steps takes only $\sim$2 minutes per target class, which is negligible compared to the multi-hour generator training ($\sim$8 hours). Core target samples are used as a pool from which mini-batches are drawn, so increasing the core pool size does not change the per-iteration cost in our regime.

Fig. 5a in the main text and Theorem 1 illustrate that the TSR is positively correlated with $|\mathcal{D}_s|$. However, as discussed, higher $|\mathcal{D}_s|$ increases training complexity. Hence, there is a tradeoff between TSR and training time. Nevertheless, according to Fig. 5a and Theorem 1, the TSR improvement rate decreases and eventually saturates as $|\mathcal{D}_s|$ grows. This suggests that, beyond a certain point, adding discriminators yields marginal gains in transferability while continuing to increase training time.

Additionally, in Fig. 6b, we compare the training time per target class across different methods. All the experiments are conducted on four NVIDIA Quadro RTX 6000, each with 24 GB of memory. The TTP method, which uses only one discriminator, requires the least amount of time (3.60 hours). M3D, despite using two discriminators, takes 9.17 hours, which is more than the time incurred by BAT-BS with five discriminators (7.81 hours). This is because M3D focuses on maximizing the discrepancy between discriminators during the generator's training process, increasing the time requirement. Both BAT-CS and BAT-CN take a few additional minutes to craft adversarial examples as compared to BAT-BS.

CGNC, despite utilizing only one discriminator, requires 12.83 hours per class. This high training time is due to CGNC's use of a much larger ImageNet training set (around 1.3 million images over 10 epochs) compared to the 50,000-image subset used by TTP, M3D, and BAT-BS (which are trained over 20 epochs). Furthermore, CGNC's generator architecture is more complex, comprising components like a Vision-Language Feature Purifier, a Feature Fusion Encoder, and a Cross-Attention-based Decoder, whereas TTP, M3D, and BAT-BS use simpler architectures with down-sampling, residual, and up-sampling blocks. The added complexity of CGNC's architecture further contributes to its longer training time.

At inference time, only the trained generator is used: a targeted adversarial example is produced with a single forward pass through $G_\Phi$ followed by a projection step. The discriminator ensemble and core-target construction are not involved during inference. Consequently, inference latency is independent of $|\mathcal{D}_s|$ and the core-set size, and is substantially lower than iterative optimization-based targeted attacks that require many forward–backward steps per image.

## E  ATTACK ON BLIP

We conduct attacks on the Vision-Language Pretraining BLIP (Li et al., 2022) model, which generates image captions, to demonstrate the effectiveness of our method in targeting Vision-Language models. Using BAT-CS, we created adversarial examples from a number of images and compared the captions generated by BLIP for these adversarial images with those generated for the original images.

Fig. 7 showcases the captions produced by BLIP for adversarial examples, where the target classes are set as "vulture" and "crayfish". When the target class is "vulture," the generated captions predominantly refer to birds, while for "crayfish," the captions often describe crabs. These results indicate the potential of our approach to craft adversarial examples capable of misleading Vision-Language models, underscoring its broader applicability.

## F  GENERATOR ARCHITECTURE

For completeness, we describe the generator $G_\Phi$ used in all BAT variants. We follow the ResNet-style image-to-image architecture commonly adopted in prior generative targeted-transfer attacks (e.g., TTP and M3D), and keep all architectural hyperparameters fixed across methods to ensure a fair comparison.

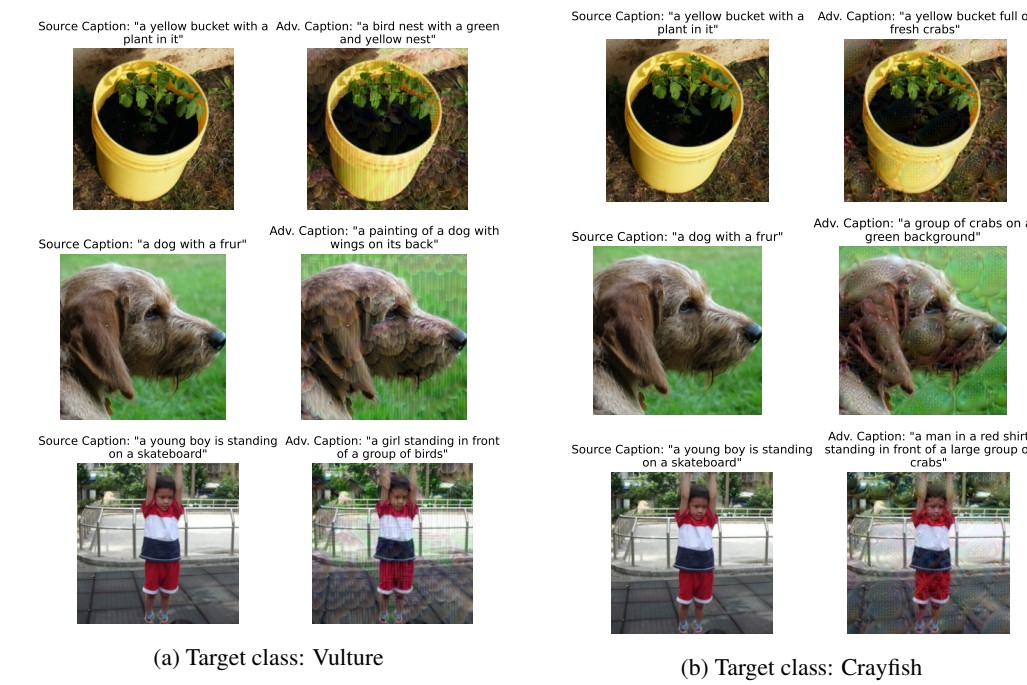

(a) Target class: Vulture

(b) Target class: Crayfish

Figure 7: Attack on image-to-caption generator Vision-Language Pre-training BLIP (Li et al., 2022). The adversarial images of the target class "**Vulture**" and "**Crayfish**" are generated from the source images using a generator trained with the proposed BAT-CS method exploiting ResNet50 as a surrogate. The generated adversarial examples are capable of successfully fooling BLIP as the generated captions are related to target classes.

**Overall structure.** The generator maps a source image $\boldsymbol{x} \in \mathbb{R}^{3 \times H \times W}$ to an adversarial image $G_\Phi(\boldsymbol{x}) \in \mathbb{R}^{3 \times H \times W}$ of the same size. It has an encoder–residual–decoder structure:

$$\boldsymbol{x} \xrightarrow{\text{encoder}} \boldsymbol{h} \xrightarrow{\text{residual blocks}} \boldsymbol{h}' \xrightarrow{\text{decoder}} G_\Phi(\boldsymbol{x}).$$

All convolutions use reflection padding, instance normalization, and ReLU activations unless otherwise specified.

**Encoder (down-sampling).** The encoder consists of three convolutional blocks:

- **Block E1:** ReflectionPad2d(3) $\rightarrow$ $7 \times 7$ Conv (3$\rightarrow$64, stride 1) $\rightarrow$ BatchNorm $\rightarrow$ ReLU.
- **Block E2:** $3 \times 3$ Conv (64$\rightarrow$128, stride 2) $\rightarrow$ BatchNorm $\rightarrow$ ReLU.
- **Block E3:** $3 \times 3$ Conv (128$\rightarrow$256, stride 2) $\rightarrow$ BatchNorm $\rightarrow$ ReLU.

This part reduces the spatial resolution by a factor of 4 and increases the channel dimension to 256.

**Residual bottleneck.** The bottleneck contains 6 residual blocks at 256 channels. Each block has the form $\boldsymbol{z} \mapsto \boldsymbol{z} + \mathcal{R}(\boldsymbol{z})$, where $\mathcal{R}$ is given by:

- ReflectionPad2d(1) $\rightarrow$ $3 \times 3$ Conv (256$\rightarrow$256) $\rightarrow$ BatchNorm $\rightarrow$ ReLU $\rightarrow$ Dropout(0.5) $\rightarrow$ ReflectionPad2d(1) $\rightarrow$ $3 \times 3$ Conv (256$\rightarrow$256) $\rightarrow$ BatchNorm.

The skip connection adds the input of the block to its output, as in standard ResNet designs.

**Decoder (up-sampling).** The decoder upsamples back to the input resolution using transposed convolutions:

- **Block D1:** ConvTranspose2d (256$\rightarrow$128, kernel 3, stride 2, padding 1, output_padding 1) $\rightarrow$ BatchNorm $\rightarrow$ ReLU.

- **Block D2:** ConvTranspose2d (128→64, kernel 3, stride 2, padding 1, output_padding 1) → BatchNorm → ReLU.

**Output head.** The final output layer is

- ReflectionPad2d(3) → $7 \times 7$ Conv (64→3),

followed by a $\tanh$ nonlinearity. We map the raw output to the image range via

$$\tilde{\boldsymbol{x}} = \frac{\tanh(G_\phi(\boldsymbol{x})) + 1}{2},$$

and then project $\tilde{\boldsymbol{x}}$ onto the $\ell_\infty$-ball around the source image to enforce $\|\tilde{\boldsymbol{x}} - \boldsymbol{x}\|_\infty \leq \epsilon$ as described in Sec. 3.

**Discussion.** This generator is significantly lighter than more complex conditional architectures (e.g., those based on cross-attention), fully convolutional, and resolution-agnostic. By reusing the same architecture as prior generative targeted-transfer methods, we isolate the effect of the proposed discriminator self-ensemble and core-target training scheme, rather than attributing gains to a stronger generator backbone.

# G    PROOF OF THEOREMS

**Definition 1.** $((\alpha_j, D_{\theta_j})$-Effective Attack). *For any input $\boldsymbol{x}$ with ground truth label $y$ and target label $y_t$, an attack is $(\alpha_j, D_{\theta_j})$-effective, if the crafted adversarial example $\boldsymbol{x}^{adv} = \boldsymbol{x} + \boldsymbol{\delta}$ satisfies $\Pr(\hat{D}_{\theta_j}(\boldsymbol{x}^{adv}) = y_t) \geq 1 - \alpha_j$, where $\hat{D}_{\theta_j}$ is the top-1 predicted label by the model $D_{\theta_j}$.*

Here, the $(\alpha_j, D_{\theta_j})$-Effective Attack captures the effectiveness of crafted adversarial examples to fool the model $D_{\theta_j}$ with a certain probability $(1 - \alpha_j)$. Note that a smaller $\alpha_j$ means the attack can better mislead $D_{\theta_j}$. If $D_{\theta_j}$ is among the accessible models used to train the generator to generate adversarial examples, $\alpha_j$ should be close to zero.

**Definition 2.** *(Transferability) Given a set of accessible models $\mathcal{D}_s = \{D_{\theta_j}\}_{j=0}^{K-1}$ and an unknown victim model $V$, the transferability of a generated adversarial example $\boldsymbol{x}^{adv} = \boldsymbol{x} + \boldsymbol{\delta}$, exploiting $\mathcal{D}_s$, to the target victim model $V$ is defined as: $T_r(\mathcal{D}_s, V, \boldsymbol{x}^{adv}, y_t) = \mathbb{1}\big((\wedge_{D_{\theta_j} \in \mathcal{D}_s}(\hat{D}_{\theta_j}(\boldsymbol{x}^{adv}) = y_t)) \wedge (\hat{V}(\boldsymbol{x}^{adv}) = y_t)\big)$, where $\mathbb{1}(.)$ denotes the indicator function and the operator $\wedge$ is a logical-and. Besides, $Tr(.) = 1$ indicates that along with the accessible models in $\mathcal{D}_s$, the crafted $\boldsymbol{x}^{adv}$ successfully deceives the target model $V$.*

In this definition of transferability, we are not concerned with whether the source image $\boldsymbol{x}$ is correctly classified by the accessible model $D_{\theta_j} \in \mathcal{D}_s$ or by the target model $V$ since $\boldsymbol{x}$ can be sampled from a different domain than the domain of the samples used to train the accessible models and the victim model, *e.g.*, the *domain shift* scenario.

## G.1    PROOF OF LOWER-BOUND OF TRANSFERABILITY

**Lemma 1.** *Let the vectors $\boldsymbol{x}, \boldsymbol{y}, \boldsymbol{\delta} \in \mathbb{R}^d$, where $\|\boldsymbol{x}\|_2 = \|\boldsymbol{y}\|_2 = 1$ and $\|\boldsymbol{\delta}\|_2 \leq \epsilon'$. For a real number $c$, if $\boldsymbol{\delta} \cdot \boldsymbol{y} > c + \epsilon'\sqrt{2 - 2m}$, then $\boldsymbol{\delta} \cdot \boldsymbol{x} > c$, where $m = \cos\langle\boldsymbol{x}, \boldsymbol{y}\rangle = \frac{\boldsymbol{x} \cdot \boldsymbol{y}}{\|\boldsymbol{x}\|_2 \|\boldsymbol{y}\|_2}$.*

*Proof.* From Cauchy-Schwarz inequality, $|\boldsymbol{\delta} \cdot (\boldsymbol{x} - \boldsymbol{y})| \leq \|\boldsymbol{\delta}\|_2 \|\boldsymbol{x} - \boldsymbol{y}\|_2 \leq \epsilon'\sqrt{\|\boldsymbol{x}\|_2 + \|\boldsymbol{y}\|_2 - 2\cos\langle\boldsymbol{x}, \boldsymbol{y}\rangle}$.

Thus, $\boldsymbol{\delta} \cdot \boldsymbol{x} = \boldsymbol{\delta} \cdot \boldsymbol{y} + \boldsymbol{\delta} \cdot (\boldsymbol{x} - \boldsymbol{y}) \geq \boldsymbol{\delta} \cdot \boldsymbol{y} - \epsilon'\sqrt{2 - 2m} > c.$ $\square$

**Lemma 2.** *For arbitrary events $A$ and $B$, we have $\Pr(A \cap B) \geq 1 - \Pr(\overline{A}) - \Pr(\overline{B})$, where $\Pr$ denotes the probability of an event.*

*Proof.* For events $A$ and $B$, we have $\Pr(A \cup B) + \Pr(\overline{A \cup B}) = 1$. As $\Pr(A) + \Pr(B) \geq \Pr(A \cup B)$ and $\Pr(\overline{A \cup B}) = \Pr(\overline{A} \cap \overline{B})$, we have $\Pr(\overline{A} \cap \overline{B}) \geq 1 - \Pr(A) - \Pr(B)$. Therefor, $\Pr(A \cap B) \geq 1 - \Pr(\overline{A}) - \Pr(\overline{B})$. $\qquad\square$

**Lemma 3.** *For two random variable a A and B, and constants a and b, we have:* $\Pr((A > a) \cup (B > b)) \geq \Pr(A + B > a + b)$.

*Proof.* Consider the event $\{A + B > a + b\}$. If $A + B > a + b$, then it must be true that at lest one of $A > a$ or $B > b$ must hold. This implies: $\{A + B > a + b\} \subseteq \{A > a\} \cup \{B > b\}$. Using the fact that the probability of a set is at least as large as the probability of any subset, we have: $\Pr((A > a) \cup (B > b)) \geq \Pr(A + B > a + b)$. $\qquad\square$

**Lemma 4.** *Given a random variable $\boldsymbol{z}$ and an arbitrary vector $\boldsymbol{b}$ such that $\boldsymbol{z}, \boldsymbol{b} \in \mathbb{R}^d$, $\|\boldsymbol{z}\|_2 \leq B$, the cosine similarity between $\boldsymbol{z}$ and $\boldsymbol{b}$ can be lower bounded by:*

$$\mathbb{E}\big[\cos \langle \boldsymbol{z}, \boldsymbol{b} \rangle\big] \geq \frac{\|\boldsymbol{b}\|_2 - \mathbb{E}[\|\boldsymbol{z} - \boldsymbol{b}\|_2]}{B}.$$

*Proof.*

$$\cos \langle \boldsymbol{z}, \boldsymbol{b} \rangle = \frac{\boldsymbol{z} \cdot \boldsymbol{b}}{\|\boldsymbol{z}\|_2 \|\boldsymbol{b}\|_2} \geq \frac{(\boldsymbol{b} + \boldsymbol{z} - \boldsymbol{b}) \cdot \boldsymbol{b}}{B\|\boldsymbol{b}\|_2}$$
$$= \frac{\|\boldsymbol{b}\|_2^2 + (\boldsymbol{z} - \boldsymbol{b}) \cdot \boldsymbol{b}}{B\|\boldsymbol{b}\|_2}$$
$$\geq \frac{\|\boldsymbol{b}\|_2^2 - \|\boldsymbol{z} - \boldsymbol{b}\|_2 \|\boldsymbol{b}\|_2}{B\|\boldsymbol{b}\|_2}$$
$$= \frac{\|\boldsymbol{b}\|_2 - \|\boldsymbol{z} - \boldsymbol{b}\|_2}{B}.$$

Thus,

$$\mathbb{E}\big[\cos \langle \boldsymbol{z}, \boldsymbol{b} \rangle\big] \geq \frac{\|\boldsymbol{b}\|_2 - \mathbb{E}\big[\|\boldsymbol{z} - \boldsymbol{b}\|_2\big]}{B}.$$

$\square$

**Theorem 1.** *Consider, $\exists \bar{V} \in \mathcal{D}$, a virtual victim model, such that $\nabla_{\boldsymbol{x}} \ell_{\bar{V}}(\boldsymbol{x}, y_t) = \mathbb{E}_{D_{\theta_i} \sim \mathcal{D}}\big[\nabla_{\boldsymbol{x}} \ell_{D_{\theta_i}}(\boldsymbol{x}, y_t)\big]$. Additionally, assume that the similarity of the gradient of $\forall D_{\theta_i} \in \mathcal{D}$ with the gradient of $\bar{V}$ is captured by $\mathbb{E}_{D_{\theta_i} \sim \mathcal{D}}\big[\|\nabla_{\boldsymbol{x}} \ell_{D_{\theta_i}}(\boldsymbol{x}, y_t) - \nabla_{\boldsymbol{x}} \ell_{\bar{V}}(\boldsymbol{x}, y_t)\|_2^2\big] \leq \sigma^2$, and $\|\nabla_{\boldsymbol{x}} \ell_{D_{\theta_i}}(\boldsymbol{x}, y_t)\|_2 \leq B$. Assume the loss function of a set of randomly picked accessible models $D_{\theta_j} \in \mathcal{D}_s \subset \mathcal{D}$ and the target model $\bar{V}$ are $\beta$-smooth, and $\forall D_{\theta_j} \in \mathcal{D}_s$ are $(\alpha_j, D_{\theta_j})$-effective on the generated samples with a perturbation constraint $\|\boldsymbol{\delta}\|_2 \leq \epsilon'$. Under these conditions, the transferability can be lower bounded by:*

$$\Pr(T_r(\mathcal{D}_s, \bar{V}, \boldsymbol{x}^{adv}, y_t) = 1) \geq 1 - A - \frac{\epsilon'(1 + A) + c_{\mathcal{D}_s}(1 - A)}{c_v + \epsilon'}$$
$$- \frac{\epsilon'}{c_v + \epsilon'} \sqrt{2\Big(1 - \frac{\|\nabla_{\boldsymbol{x}} \ell_{\bar{V}}(\boldsymbol{x}, y_t)\|_2 - \frac{\sigma}{\sqrt{|\mathcal{D}_s|}}}{B}\Big)},$$

*where $A = \sum_{i=0}^{|\mathcal{D}_s|} \alpha_j$,*

$$c_{\mathcal{D}_s} := \max_{\boldsymbol{x} \in \mathcal{X}} \frac{\Big(\min_{y \in [L] - \{y_t\}} \frac{1}{|\mathcal{D}_s|} \sum_{D_{\theta_j} \in \mathcal{D}_s} \ell_{D_{\theta_j}}(\boldsymbol{x}^{adv}, y) - \frac{1}{|\mathcal{D}_s|} \sum_{D_{\theta_j} \in \mathcal{D}_s} \ell_{D_{\theta_j}}(\boldsymbol{x}, y_t) + \frac{\beta}{2}\epsilon'^2\Big)}{\|\frac{1}{|\mathcal{D}_s|} \sum_{D_{\theta_j} \in \mathcal{D}_s} \nabla_{\boldsymbol{x}} \ell_{D_{\theta_j}}(\boldsymbol{x}, y_t)\|_2},$$

$$c_v := \min_{\boldsymbol{x} \in \mathcal{X}} \frac{\min_{y \in [L] - \{y_t\}} \ell_{\bar{V}}(\boldsymbol{x}^{adv}, y) - \ell_{\bar{V}}(\boldsymbol{x}, y_t) - \frac{\beta}{2}\epsilon'^2}{\|\nabla_{\boldsymbol{x}} \ell_{\bar{V}}(\boldsymbol{x}, y_t)\|_2}.$$

*Here $c_{\mathcal{D}_s}$ is the average risk of the models in $\mathcal{D}_s$ and $c_v$ is the risk of the virtual victim model $\bar{V}$.*

*Proof.* This proof builds upon the derivation in (Yang et al., 2021) with a primary focus on demonstrating the impact of an ensemble of accessible models on adversarial transferability. According to the definition of transferability, for a given input $\boldsymbol{x}$, the generated adversarial example $\boldsymbol{x}^{adv} = \boldsymbol{x} + \boldsymbol{\delta}$ must be misclassified as the target class $y_t$ by both surrogate models $D_{\theta_j} \in \mathcal{D}_s$ and the target model $\bar{V}$. Hence, we have

$$\Pr(T_r(\mathcal{D}_s, \bar{V}, \boldsymbol{x}^{adv}, y_t) = 1) = \Pr\left(\left(\bigwedge_{D_{\theta_j} \in \mathcal{D}_s}(\hat{D}_{\theta_j}(\boldsymbol{x}^{adv}) = y_t)\right) \wedge (\hat{\bar{V}}(\boldsymbol{x}^{adv}) = y_t)\right)$$

$$\overset{(a)}{\geq} 1 - \sum_{D_{\theta_j} \in \mathcal{D}_s} \Pr(\hat{D}_{\theta_j}(\boldsymbol{x}^{adv}) \neq y_t) - \Pr(\hat{\bar{V}}(\boldsymbol{x}^{adv}) \neq y_t) \overset{(b)}{\geq} 1 - \sum_{i=0}^{|\mathcal{D}_s|-1} \alpha_j - \Pr(\hat{\bar{V}}(\boldsymbol{x}^{adv}) \neq y_t),$$

$$(11)$$

where inequality $(a)$ follows Lemma 2 and the $(b)$ is obtained by utilizing Definition 1.

For a given input $\boldsymbol{x}^{adv}$, a model $D_{\theta_j}$ will predict the label for which the loss $\ell_{D_{\theta_j}}$ is minimum. Thus, $\hat{D}_{\theta_j}(\boldsymbol{x}^{adv}) \neq y_t \Leftrightarrow \ell_{D_{\theta_j}}(\boldsymbol{x}^{adv}, y_t) > \min_{y \in \mathcal{C}} \ell_{D_{\theta_j}}(\boldsymbol{x}^{adv}, y)$. Similarly, $\hat{\bar{V}}(\boldsymbol{x}^{adv}) \neq y_t \Leftrightarrow \ell_{\bar{V}}(\boldsymbol{x}^{adv}, y_t) > \min_{y \in \mathcal{C}} \ell_{\bar{V}}(\boldsymbol{x}^{adv}, y)$, where $\mathcal{C} = [L] - \{y_t\}$ is the set of all classes except the target one.

As the loss function $\ell_{D_{\theta_j}}, \forall D_{\theta_j} \in \mathcal{D}_s$ are $\beta$-smooth, we have:

$$|\ell_{D_{\theta_j}}(\boldsymbol{x}^{adv}, y_t) - \ell_{D_{\theta_j}}(\boldsymbol{x}, y_t) - \langle \boldsymbol{\delta}, \nabla_{\boldsymbol{x}} \ell_{D_{\theta_j}}(\boldsymbol{x}, y_t)\rangle| \leq \frac{\beta}{2}\|\boldsymbol{\delta}\|_2^2 \leq \frac{\beta}{2}\epsilon'^2; \ \ \forall D_{\theta_j} \in \mathcal{D}, \qquad (12)$$

$$\Rightarrow \ell_{D_{\theta_j}}(\boldsymbol{x}, y_t) + \boldsymbol{\delta} \cdot \nabla_{\boldsymbol{x}} \ell_{D_{\theta_j}}(\boldsymbol{x}, y_t) - \frac{\beta}{2}\epsilon'^2 \leq \ell_{D_{\theta_j}}(\boldsymbol{x}^{adv}, y_t) \leq \ell_{D_{\theta_j}}(\boldsymbol{x}, y_t) + \boldsymbol{\delta} \cdot \nabla_{\boldsymbol{x}} \ell_{D_{\theta_j}}(\boldsymbol{x}, y_t) + \frac{\beta}{2}\epsilon'^2,$$

$$(13)$$

where $\boldsymbol{x}^{adv} = \boldsymbol{x} + \boldsymbol{\delta}$ and $\|\boldsymbol{\delta}\|_2 \leq \epsilon'$. Similarly, for the victim model $\bar{V}$, we have

$$\ell_{\bar{V}}(\boldsymbol{x}, y_t) + \boldsymbol{\delta} \cdot \nabla_{\boldsymbol{x}} \ell_{\bar{V}}(\boldsymbol{x}, y_t) - \frac{\beta}{2}\epsilon'^2 \leq \ell_{\bar{V}}(\boldsymbol{x}^{adv}, y_t) \leq \ell_{\bar{V}}(\boldsymbol{x}, y_t) + \boldsymbol{\delta} \cdot \nabla_{\boldsymbol{x}} \ell_{\bar{V}}(\boldsymbol{x}, y_t) + \frac{\beta}{2}\epsilon'^2.$$

Now,

$$\sum_{D_{\theta_j} \in \mathcal{D}_s} \Pr\left(\hat{D}_{\theta_j}(\boldsymbol{x}^{adv}) \neq y_t\right) = \sum_{D_{\theta_j} \in \mathcal{D}_s} \Pr\left(\ell_{D_{\theta_j}}(\boldsymbol{x}^{adv}, y_t) > \min_{y \in \mathcal{C}} \ell_{D_{\theta_j}}(\boldsymbol{x}^{adv}, y)\right)$$

$$\overset{(a)}{\geq} \Pr\left(\bigcup_{D_{\theta_j} \in \mathcal{D}_s}\left(\ell_{D_{\theta_j}}(\boldsymbol{x}^{adv}, y_t) > \min_{y \in \mathcal{C}} \ell_{D_{\theta_j}}(\boldsymbol{x}^{adv}, y)\right)\right)$$

$$\overset{(b)}{\geq} \Pr\left(\sum_{D_{\theta_j} \in \mathcal{D}_s} \ell_{D_{\theta_j}}(\boldsymbol{x}^{adv}, y_t) > \sum_{D_{\theta_j} \in \mathcal{D}_s} \min_{y \in \mathcal{C}} \ell_{D_{\theta_j}}(\boldsymbol{x}^{adv}, y)\right)$$

$$\geq \Pr\left(\sum_{D_{\theta_j} \in \mathcal{D}_s} \ell_{D_{\theta_j}}(\boldsymbol{x}^{adv}, y_t) > \min_{y \in \mathcal{C}} \sum_{D_{\theta_j} \in \mathcal{D}_s} \ell_{D_{\theta_j}}(\boldsymbol{x}^{adv}, y)\right)$$

$$\overset{(c)}{\geq} \Pr\left(\frac{1}{|\mathcal{D}_s|} \sum_{D_{\theta_j} \in \mathcal{D}_s}(\ell_{D_{\theta_j}}(\boldsymbol{x}, y_t) + \boldsymbol{\delta} \cdot \nabla_{\boldsymbol{x}} \ell_{D_{\theta_j}}(\boldsymbol{x}, y_t) - \frac{\beta}{2}\epsilon'^2) > \min_{y \in \mathcal{C}} \frac{1}{|\mathcal{D}_s|} \sum_{D_{\theta_j} \in \mathcal{D}_s} \ell_{D_{\theta_j}}(\boldsymbol{x}^{adv}, y)\right)$$

$$= \Pr\left(\boldsymbol{\delta} \cdot \frac{\frac{1}{|\mathcal{D}_s|} \sum_{D_{\theta_j} \in \mathcal{D}_s} \nabla_{\boldsymbol{x}} \ell_{D_{\theta_j}}(\boldsymbol{x}, y_t)}{\|\frac{1}{|\mathcal{D}_s|} \sum_{D_{\theta_j} \in \mathcal{D}_s} \nabla_{\boldsymbol{x}} \ell_{D_{\theta_j}}(\boldsymbol{x}, y_t)\|_2} > f(\boldsymbol{x})\right) \qquad (14)$$

where the inequality $(a)$ due to the fact that $P(A) + P(B) \geq P(A \cup B)$, $(b)$ and $(c)$ is obtained using Lemma 3 and Eq. 13. Moreover, $f(\boldsymbol{x})$ is defined as follows:

$$f(\boldsymbol{x}) = \frac{\left(\min_{y \in \mathcal{C}} \frac{1}{|\mathcal{D}_s|} \sum_{D_{\theta_j} \in \mathcal{D}_s} \ell_{D_{\theta_j}}(\boldsymbol{x}^{adv}, y) - \frac{1}{|\mathcal{D}_s|} \sum_{D_{\theta_j} \in \mathcal{D}_s} \ell_{D_{\theta_j}}(\boldsymbol{x}, y_t) + \frac{\beta}{2}\epsilon'^2\right)}{\|\frac{1}{|\mathcal{D}_s|} \sum_{D_{\theta_j} \in \mathcal{D}_s} \nabla_{\boldsymbol{x}} \ell_{D_{\theta_j}}(\boldsymbol{x}, y_t)\|_2}.$$

From Definition 1, we have $\sum_{D_{\theta_j} \in \mathcal{D}_s} \Pr\left(\hat{D}_{\theta_j}(\boldsymbol{x}^{adv}) \neq y_t\right) \leq \sum_{j=0}^{|\mathcal{D}_s|} \alpha_j$. Thus, utilizing Eq. 14 we have,

$$\Pr\left(\delta \cdot \frac{\frac{1}{|\mathcal{D}_s|} \sum_{D_{\theta_j} \in \mathcal{D}_s} \nabla_{\boldsymbol{x}} \ell_{D_{\theta_j}}(\boldsymbol{x}, y_t)}{\|\frac{1}{|\mathcal{D}_s|} \sum_{D_{\theta_j} \in \mathcal{D}_s} \nabla_{\boldsymbol{x}} \ell_{D_{\theta_j}}(\boldsymbol{x}, y_t)\|_2} > f(\boldsymbol{x})\right) \leq A, \quad (15)$$

where $A := \sum_{j=0}^{|\mathcal{D}_s|} \alpha_j$.

Similarly,

$$\Pr\left(\hat{\bar{V}}(\boldsymbol{x}^{adv}) \neq y_t\right) = \Pr\left(\ell_{\bar{V}}(\boldsymbol{x}^{adv}, y_t) > \min_{y \in \mathcal{C}} \ell_{\bar{V}}(\boldsymbol{x}^{adv}, y)\right)$$

$$\overset{(a)}{\leq} \Pr\left(\ell_{\bar{V}}(\boldsymbol{x}, y_t) + \boldsymbol{\delta} \cdot \nabla_{\boldsymbol{x}} \ell_{\bar{V}}(\boldsymbol{x}, y_t) + \frac{\beta}{2}\epsilon'^2 > \min_{y \in \mathcal{C}} \ell_{\bar{V}}(\boldsymbol{x}^{adv}, y)\right)$$

$$= \Pr\left(\boldsymbol{\delta} \cdot \frac{\nabla_{\boldsymbol{x}} \ell_{\bar{V}}(\boldsymbol{x}, y_t)}{\|\nabla_{\boldsymbol{x}} \ell_{\bar{V}}(\boldsymbol{x}, y_t)\|_2} > g(\boldsymbol{x})\right), \quad (16)$$

where inequality $(a)$ is obtained from Eq. 13, and

$$g(\boldsymbol{x}) = \frac{\min_{y \in \mathcal{C}} \ell_{\bar{V}}(\boldsymbol{x}^{adv}, y) - \ell_{\bar{V}}(\boldsymbol{x}, y_t) - \frac{\beta}{2}\epsilon'^2}{\|\nabla_{\boldsymbol{x}} \ell_{\bar{V}}(\boldsymbol{x}, y_t)\|_2}.$$

Thus, according to Lemma 1 and having $\|\boldsymbol{\delta}\|_2 \leq \epsilon'$, $\boldsymbol{\delta} \cdot \frac{\frac{1}{|\mathcal{D}_s|} \sum_{D_{\theta_j} \in \mathcal{D}_s} \nabla_{\boldsymbol{x}} \ell_{D_{\theta_j}}(\boldsymbol{x}, y_t)}{\|\frac{1}{|\mathcal{D}_s|} \sum_{D_{\theta_j} \in \mathcal{D}_s} \nabla_{\boldsymbol{x}} \ell_{D_{\theta_j}}(\boldsymbol{x}, y_t)\|_2} > f(\boldsymbol{x})$ if

$$\boldsymbol{\delta} \cdot \frac{\nabla_{\boldsymbol{x}} \ell_{\bar{V}}(\boldsymbol{x}, y_t)}{\|\nabla_{\boldsymbol{x}} \ell_{\bar{V}}(\boldsymbol{x}, y_t)\|_2} > f(\boldsymbol{x}) + \epsilon' \sqrt{2 - 2S(\mathcal{D}_s, \bar{V})},$$

where $S(\mathcal{D}_s, \bar{V})$ measures the cosine similarity between $\frac{\frac{1}{|\mathcal{D}_s|} \sum_{D_{\theta_j} \in \mathcal{D}_s} \nabla_{\boldsymbol{x}} \ell_{D_{\theta_j}}(\boldsymbol{x}, y_t)}{\|\frac{1}{|\mathcal{D}_s|} \sum_{D_{\theta_j} \in \mathcal{D}_s} \nabla_{\boldsymbol{x}} \ell_{D_{\theta_j}}(\boldsymbol{x}, y_t)\|_2}$ and $\frac{\nabla_{\boldsymbol{x}} \ell_{\bar{V}}(\boldsymbol{x}, y_t)}{\|\nabla_{\boldsymbol{x}} \ell_{\bar{V}}(\boldsymbol{x}, y_t)\|_2}$. Thus, we get

$$\Pr\left(\boldsymbol{\delta} \cdot \frac{\nabla_{\boldsymbol{x}} \ell_{\bar{V}}(\boldsymbol{x}, y_t)}{\|\nabla_{\boldsymbol{x}} \ell_{\bar{V}}(\boldsymbol{x}, y_t)\|_2} > f(\boldsymbol{x}) + \epsilon' \sqrt{2 - 2S(\mathcal{D}_s, \bar{V})}\right)$$

$$\leq \Pr\left(\boldsymbol{\delta} \cdot \frac{\frac{1}{|\mathcal{D}|} \sum_{D_{\theta_j} \in \mathcal{D}} \nabla_{\boldsymbol{x}} \ell_{D_{\theta_j}}(\boldsymbol{x}, y_t)}{\|\frac{1}{|\mathcal{D}|} \sum_{D_{\theta_j} \in \mathcal{D}} \nabla_{\boldsymbol{x}} \ell_{D_{\theta_j}}(\boldsymbol{x}, y_t)\|_2} > f(\boldsymbol{x})\right) \leq A, \quad (17)$$

where the last inequality using Eq. 15. Given,

$$c_{\mathcal{D}_s} = \max_{\boldsymbol{x} \in \mathcal{X}} \frac{\begin{aligned}&\left(\min_{y \in \mathcal{C}} \frac{1}{|\mathcal{D}_s|} \sum_{D_{\theta_j} \in \mathcal{D}_s} \ell_{D_{\theta_j}}(\boldsymbol{x}^{adv}, y)\right.\\&\left.- \frac{1}{|\mathcal{D}_s|} \sum_{D_{\theta_j} \in \mathcal{D}_s} \ell_{D_{\theta_j}}(\boldsymbol{x}, y_t) + \frac{\beta}{2}\epsilon'^2\right)\end{aligned}}{\|\frac{1}{|\mathcal{D}_s|} \sum_{D_{\theta_j} \in \mathcal{D}_s} \nabla_{\boldsymbol{x}} \ell_{D_{\theta_j}}(\boldsymbol{x}, y_t)\|_2}.$$

Since $c_{\mathcal{D}_s} \geq f(x)$, EQ. 17 can be expressed as,

$$\Pr\left(\boldsymbol{\delta} \cdot \frac{\nabla_{\boldsymbol{x}} \ell_{\bar{V}}(\boldsymbol{x}, y_t)}{\|\nabla_{\boldsymbol{x}} \ell_{\bar{V}}(\boldsymbol{x}, y_t)\|_2} - \epsilon' \sqrt{2 - 2S(\mathcal{D}_s, \bar{V})} > c_{\mathcal{D}_s}\right) \leq A$$

Now, the maximum value of $\boldsymbol{\delta} \cdot \frac{\nabla_{\boldsymbol{x}} \ell_{\bar{V}}(\boldsymbol{x}, y_t)}{\|\nabla_{\boldsymbol{x}} \ell_{\bar{V}}(\boldsymbol{x}, y_t)\|_2} - \epsilon' \sqrt{2 - 2S(\mathcal{D}_s, \bar{V})}$ is $\epsilon'$. Therefore, the expectation can be bounded:

$$\mathbb{E}\left[\boldsymbol{\delta} \cdot \frac{\nabla_{\boldsymbol{x}} \ell_{\bar{V}}(\boldsymbol{x}, y_t)}{\|\nabla_{\boldsymbol{x}} \ell_{\bar{V}}(\boldsymbol{x}, y_t)\|_2} - \epsilon' \sqrt{2 - 2S(\mathcal{D}_s, \bar{V})}\right] \leq \epsilon' A + c_{\mathcal{D}_s}(1 - A)$$

Hence,

$$\mathbb{E}\left[\boldsymbol{\delta} \cdot \frac{\nabla_{\boldsymbol{x}} \ell_{\bar{V}}(\boldsymbol{x}, y_t)}{\|\nabla_{\boldsymbol{x}} \ell_{\bar{V}}(\boldsymbol{x}, y_t)\|_2}\right] \leq \mathbb{E}\left[\epsilon' \sqrt{2 - 2S(\mathcal{D}_s, \bar{V})}\right] + \epsilon' A + c_{\mathcal{D}_s}(1 - A)$$

$$\leq \epsilon' \sqrt{2 - 2\mathbb{E}\left[S(\mathcal{D}_s, \bar{V})\right]} + \epsilon' A + c_{\mathcal{D}_s}(1 - A)$$

Moreover, given

$$c_v = \min_{\boldsymbol{x} \in \mathcal{X}} \frac{\min_{y \in \mathcal{C}} \ell_{\bar{V}}(\boldsymbol{x}^{adv}, y) - \ell_{\bar{V}}(\boldsymbol{x}, y_t) - \frac{\beta}{2}\epsilon'^2}{\|\nabla_{\boldsymbol{x}} \ell_{\bar{V}}(\boldsymbol{x}, y_t)\|_2}.$$

Since $c_v \le g(x)$, applying Markov's inequality, we get

$$\Pr\left(\boldsymbol{\delta} \cdot \frac{\nabla_{\boldsymbol{x}} \ell_{\bar{V}}(\boldsymbol{x}, y_t)}{\|\nabla_{\boldsymbol{x}} \ell_{\bar{V}}(\boldsymbol{x}, y_t)\|_2} > g(\boldsymbol{x})\right)$$

$$\le \Pr\left(\boldsymbol{\delta} \cdot \frac{\nabla_{\boldsymbol{x}} \ell_{\bar{V}}(\boldsymbol{x}, y_t)}{\|\nabla_{\boldsymbol{x}} \ell_{\bar{V}}(\boldsymbol{x}, y_t)\|_2} > c_v\right)$$

$$\le \frac{\epsilon' \sqrt{2 - 2\mathbb{E}[S(\mathcal{D}_s, \bar{V})]} + \epsilon' A + c_{\mathcal{D}_s}(1 - A)}{c_v}$$

$$\le \frac{\epsilon'(1 + A) + c_{\mathcal{D}_s}(1 - A) + \epsilon' \sqrt{2 - 2\mathbb{E}[S(\mathcal{D}_s, \bar{V})]}}{c_v + \epsilon'}. \tag{18}$$

Given

$$\mathbb{E}_{D_{\theta_i} \sim \mathcal{D}}\left[\|\nabla_{\boldsymbol{x}} \ell_{D_{\theta_i}}(\boldsymbol{x}, y_t) - \nabla_{\boldsymbol{x}} \ell_{\bar{V}}(\boldsymbol{x}, y_t)\|_2^2\right] \le \sigma^2.$$

Since $\mathbb{E}_{D_{\theta_i} \sim \mathcal{D}}\left[\nabla_{\boldsymbol{x}} \ell_{D_{\theta_i}}(\boldsymbol{x}, y_t)\right] = \nabla_{\boldsymbol{x}} \ell_{\bar{V}}(\boldsymbol{x}, y_t)$, we have,

$$\mathbb{E}\left[\left\|\frac{1}{|\mathcal{D}_s|} \sum_{D_{\theta_j} \in \mathcal{D}_s} \nabla_{\boldsymbol{x}} \ell_{D_{\theta_j}}(\boldsymbol{x}, y_t) - \nabla_{\boldsymbol{x}} \ell_{\bar{V}}(\boldsymbol{x}, y_t)\right\|_2^2\right] \le \frac{\sigma^2}{|\mathcal{D}_s|}. \tag{19}$$

Given $\|\nabla_{\boldsymbol{x}} \ell_{D_{\theta_i}}(\boldsymbol{x}, y_t)\|_2 \le B, \forall D_{\theta_i} \in \mathcal{D}$. Thus, we have $\|\frac{1}{|\mathcal{D}_s|} \sum_{D_{\theta_j} \in \mathcal{D}_s} \nabla_{\boldsymbol{x}} \ell_{D_{\theta_j}}(\boldsymbol{x}, y_t)\|_2 \le B$. Therefore, using Lemma 4, we have the cosine similarity between $\frac{1}{|\mathcal{D}_s|} \sum_{D_{\theta_j} \in \mathcal{D}_s} \nabla_{\boldsymbol{x}} \ell_{D_{\theta_j}}(\boldsymbol{x}, y_t)$ and $\nabla_{\boldsymbol{x}} \ell_{\bar{V}}(\boldsymbol{x}, y_t)$:

$$S(\mathcal{D}_s, \bar{V}) \ge \frac{\|\nabla_{\boldsymbol{x}} \ell_{\bar{V}}(\boldsymbol{x}, y_t)\|_2 - \frac{\sigma}{\sqrt{|\mathcal{D}_s|}}}{B} \tag{20}$$

Combining Eq. 11, Eq. 16, Eq. 18 and Eq. 20, we have the desired upper bound:

$$\Pr(T_r(\mathcal{D}_s, \bar{V}, \boldsymbol{x}^{adv}, y_t) = 1) \ge 1 - A - \frac{\epsilon'(1 + A) + c_{\mathcal{D}_s}(1 - A)}{c_v + \epsilon'}$$

$$- \frac{\epsilon'}{c_v + \epsilon'} \sqrt{2\left(1 - \frac{\|\nabla_{\boldsymbol{x}} \ell_{\bar{V}}(\boldsymbol{x}, y_t)\|_2 - \frac{\sigma}{\sqrt{|\mathcal{D}_s|}}}{B}\right)}. \tag{21}$$

$\square$

## G.2 PROOF OF UPPER-BOUND OF TRANSFERABILITY

**Lemma 5.** *Suppose two unit vectors $\boldsymbol{x}$ and $\boldsymbol{y}$ satisfy $\boldsymbol{x} \cdot \boldsymbol{y} = S$, then for any $\boldsymbol{\delta}$, we have $\min(\boldsymbol{\delta} \cdot \boldsymbol{x}, \boldsymbol{\delta} \cdot \boldsymbol{y}) \le \|\boldsymbol{\delta}\|_2 \sqrt{(1 + S)/2}$.*

*Proof.* Denote $\alpha$ is the angle between $\boldsymbol{x}$ and $\boldsymbol{y}$ and then $S = \cos\langle \boldsymbol{x}, \boldsymbol{y} \rangle = \cos\alpha$. If $\alpha_x, \alpha_y$ are the angles between $\boldsymbol{\delta}$ and $\boldsymbol{x}$ and between $\boldsymbol{\delta}$ and $\boldsymbol{y}$, respectively, then we have $\max(\alpha_x, \alpha_y) \ge \frac{\alpha}{2} = \frac{\cos^{-1} S}{2}$. Since $\cos\alpha/2 = \sqrt{\frac{S+1}{2}}$, we have $\min(\boldsymbol{\delta} \cdot \boldsymbol{x}, \boldsymbol{\delta} \cdot \boldsymbol{y}) \le \|\boldsymbol{\delta}\|_2 \sqrt{(1 + S)/2}$. $\square$

**Lemma 6.** *For a set of $N$ random variables $\{\boldsymbol{x}_i\}_{i=1}^N$ with a same mean $\boldsymbol{b} = \mathbb{E}[\boldsymbol{x}_i], \forall i \in [N]$, if $\boldsymbol{y} = \sum_{i=1}^N \boldsymbol{x}_i, C \le \|\boldsymbol{x}_i\| \le B$ and $\lambda^2 \le \mathbb{E}[\|\boldsymbol{x}_i - \boldsymbol{b}\|^2]$, we have $\mathbb{E}[\cos\langle \boldsymbol{y}, \boldsymbol{b} \rangle] \le \frac{B^2 + \|\boldsymbol{b}\|^2 - \frac{\lambda^2}{N}}{2C\|\boldsymbol{b}\|}$.*

*Proof.* Given

$$\lambda^2 \leq \mathbb{E}\big[\|\boldsymbol{x}_i - \boldsymbol{b}\|^2\big].$$

If $y = \sum_{i=1}^N \boldsymbol{x}_i$, then, $\frac{\lambda^2}{N} \leq \mathbb{E}\big[\|\boldsymbol{y} - \boldsymbol{b}\|^2\big]$. Therefore,

$$\mathbb{E}\big[\|\boldsymbol{y}\|^2 + \|\boldsymbol{b}\|^2 - 2\|\boldsymbol{y}\|\|\boldsymbol{b}\| \cos\langle \boldsymbol{y}, \boldsymbol{b}\rangle\big] \geq \frac{\lambda^2}{N}$$

$$\implies B^2 + \|\boldsymbol{b}\|^2 - 2C\|\boldsymbol{b}\|\mathbb{E}[\cos\langle \boldsymbol{y}, \boldsymbol{b}\rangle] \geq \frac{\lambda^2}{N} \tag{22}$$

Hence,

$$\mathbb{E}[\cos\langle \boldsymbol{y}, \boldsymbol{b}\rangle] \leq \frac{B^2 + \|\boldsymbol{b}\|^2 - \frac{\lambda^2}{N}}{2C\|\boldsymbol{b}\|}.$$

$\square$

**Theorem 2.** *Consider,* $\exists \bar{V} \in \mathcal{D}$, *a virtual victim model, such that* $\nabla_{\boldsymbol{x}}\ell_{\bar{V}}(\boldsymbol{x}, y) = \mathbb{E}_{D_{\theta_i} \sim \mathcal{D}}\big[\nabla_{\boldsymbol{x}}\ell_{D_{\theta_i}}(\boldsymbol{x}, y)\big]$. *Additionally, assume that the similarity of the gradient of* $\forall D_{\theta_i} \in \mathcal{D}$ *with the gradient of* $\bar{V}$ *is captured by* $\lambda^2 \leq \mathbb{E}_{D_{\theta_i} \sim \mathcal{D}}\big[\|\nabla_{\boldsymbol{x}}\ell_{D_{\theta_i}}(\boldsymbol{x}, y_t) - \nabla_{\boldsymbol{x}}\ell_{\bar{V}}(\boldsymbol{x}, y_t)\|_2^2\big]$, *and* $C \leq \|\nabla_{\boldsymbol{x}}\ell_{D_{\theta_i}}(\boldsymbol{x}, y_t)\|_2 \leq B$. *Assume the loss function of a set of accessible models* $D_{\theta_j} \in \mathcal{D}_s \subset \mathcal{D}$ *and the target model* $\bar{V}$ *are* $\beta$-*smooth, and the accessible models* $D_{\theta_j} \in \mathcal{D}_s$ *are* $(\alpha_j, D_{\theta_j})$-*effective on the generated samples with a perturbation constraint* $\|\boldsymbol{\delta}\|_2 \leq \epsilon'$. *Under these conditions, the transferability can be upper bounded by:*

$$\Pr(T_r(\mathcal{D}_s, \bar{V}, \boldsymbol{x}^{adv}, y_t) = 1) \leq \frac{\xi + \epsilon'B\sqrt{\frac{1+\mathbb{E}[S(\mathcal{D}_s, \bar{V})]}{2}}}{\frac{1}{|\mathcal{D}_s|}\sum_{D_{\theta_j} \in \mathcal{D}_s}\ell_{D_{\theta_j}}(\boldsymbol{x}, y_t) - \epsilon'B - \beta\epsilon'^2} + \frac{\xi + \epsilon'B\sqrt{\frac{1+\mathbb{E}[S(\mathcal{D}_s, \bar{V})]}{2}}}{\ell_{\bar{V}}(\boldsymbol{x}, y_t) - \epsilon'B - \beta\epsilon'^2},$$

*where* $\xi = \mathbb{E}_{D_{\theta_i} \sim \mathcal{D}}\big[\ell_{D_{\theta_i}}(\boldsymbol{x}, y)\big]$, $S(\mathcal{D}_s, \bar{V})$ *is the cosine similarity between* $\frac{1}{|\mathcal{D}_s|}\sum_{j=1}^{|\mathcal{D}_s|}\nabla_{\boldsymbol{x}}\ell_{D_{\theta_j}}(\boldsymbol{x}, y)$ *and* $\nabla_{\boldsymbol{x}}\ell_{\bar{V}}(\boldsymbol{x}, y)$, *and*

$$\mathbb{E}[S(\mathcal{D}_s, \bar{V})] \leq \frac{B^2 + \|\nabla_{\boldsymbol{x}}\ell_{\bar{V}}(\boldsymbol{x}, y)\|^2 - \frac{\lambda^2}{|\mathcal{D}_s|}}{2C\|\nabla_{\boldsymbol{x}}\ell_{\bar{V}}(\boldsymbol{x}, y)\|}.$$

*Here* $\mathbb{E}[S(\mathcal{D}_s, \bar{V})]$ *captures the expected similarity between* $\frac{1}{|\mathcal{D}_s|}\sum_{j=1}^{|\mathcal{D}_s|}\nabla_{\boldsymbol{x}}\ell_{D_{\theta_j}}(\boldsymbol{x}, y)$ *and* $\nabla_{\boldsymbol{x}}\ell_{\bar{V}}(\boldsymbol{x}, y)$. $\mathbb{E}[S(\mathcal{D}_s, \bar{V})]$ *is positively correlated with* $|\mathcal{D}_s|$. *This implies the upper bound of the transferability is also positively correlated with* $|\mathcal{D}_s|$.

*Proof.* Let $\boldsymbol{x}^{adv} = \boldsymbol{x} + \boldsymbol{\delta}$ be an adversarial example of the image $\boldsymbol{x}$ with $y$ and $y_t$ as the true label and the target label, respectively. Since $D_{\theta_j}, \forall D_{\theta_j} \in \mathcal{D}_s$ minimizes the loss $\ell_{D_{\theta_j}}$, we have

$$\hat{D}_{\theta_j}(\boldsymbol{x}^{adv}) = y_t \implies \min_{c \in \mathcal{C}}\ell_{D_{\theta_j}}(\boldsymbol{x}^{adv}, c) > \ell_{D_{\theta_j}}(\boldsymbol{x}^{adv}, y_t),$$

where $\mathcal{C} = [L] - \{y_t\}$. Hence

$$\Pr(\hat{D}_{\theta_j}(\boldsymbol{x}^{adv}) = y_t) \leq \Pr(\ell_{D_{\theta_j}}(\boldsymbol{x}^{adv}, y) > \ell_{D_{\theta_j}}(\boldsymbol{x}^{adv}, y_t)). \tag{23}$$

Similarly, for $\bar{V}$, we have

$$\hat{\bar{V}}(\boldsymbol{x}^{adv}) = y_t \implies \min_{c \in \mathcal{C}}\ell_{\bar{V}}(\boldsymbol{x}^{adv}, c) > \ell_{\bar{V}}(\boldsymbol{x}^{adv}, y_t),$$

and that implies

$$\Pr(\hat{\bar{V}}(\boldsymbol{x}^{adv}) = y_t) \leq \Pr(\ell_{\bar{V}}(\boldsymbol{x}^{adv}, y) > \ell_{\bar{V}}(\boldsymbol{x}^{adv}, y_t)). \tag{24}$$

Since $D_{\theta_j}, \forall \in \mathcal{D}_s$ is $\beta$-smooth, we have:

$$\ell_{D_{\theta_j}}(\boldsymbol{x}, y) + \boldsymbol{\delta} \cdot \nabla_{\boldsymbol{x}}\ell_{D_{\theta_j}}(\boldsymbol{x}, y) + \frac{\beta}{2}\|\boldsymbol{\delta}\|_2^2 \geq \ell_{D_{\theta_j}}(\boldsymbol{x}^{adv}, y).$$

Thus,

$$\boldsymbol{\delta} \cdot \nabla_{\boldsymbol{x}} \ell_{D_{\theta_j}}(\boldsymbol{x}, y) \geq \ell_{D_{\theta_j}}(\boldsymbol{x}^{adv}, y) - \ell_{D_{\theta_j}}(\boldsymbol{x}, y) - \frac{\beta}{2} \|\boldsymbol{\delta}\|_2^2$$

$$\geq \ell_{D_{\theta_j}}(\boldsymbol{x}^{adv}, y_t) - \ell_{D_{\theta_j}}(\boldsymbol{x}, y) - \frac{\beta}{2} \|\boldsymbol{\delta}\|_2^2 := c_{D_{\theta_j}}. \tag{25}$$

Likewise for $\bar{V}$,

$$\boldsymbol{\delta} \cdot \nabla_{\boldsymbol{x}} \ell_{\bar{V}}(\boldsymbol{x}, y) \geq \ell_{\bar{V}}(\boldsymbol{x}^{adv}, y_t) - \ell_{\bar{V}}(\boldsymbol{x}, y) - \frac{\beta}{2} \|\boldsymbol{\delta}\|_2^2 := c_{\bar{V}}. \tag{26}$$

Hence, from Eq. 25, we have

$$\Pr\left(\ell_{D_{\theta_j}}(\boldsymbol{x}^{adv}, y) > \ell_{D_{\theta_j}}(\boldsymbol{x}^{adv}, y_t)\right) \leq \Pr\left(\boldsymbol{\delta} \cdot \nabla_{\boldsymbol{x}} \ell_{D_{\theta_j}}(\boldsymbol{x}, y) \geq c_{D_{\theta_j}}\right). \tag{27}$$

Similarly, from Eq. 26, we have

$$\Pr\left(\ell_{\bar{V}}(\boldsymbol{x}^{adv}, y) > \ell_{\bar{V}}(\boldsymbol{x}^{adv}, y_t)\right) \leq \Pr\left(\boldsymbol{\delta} \cdot \nabla_{\boldsymbol{x}} \ell_{\bar{V}}(\boldsymbol{x}, y) \geq c_{\bar{V}}\right) \tag{28}$$

Hence,

$$\Pr(T_r(\mathcal{D}_s, \bar{V}, \boldsymbol{x}^{adv}, y_t) = 1)$$

$$= \Pr\left(\left(\bigwedge_{D_{\theta_j} \in \mathcal{D}_s} \hat{D}_{\theta_j}(\boldsymbol{x}^{adv}) = y_t\right) \wedge \hat{\bar{V}}(\boldsymbol{x}^{adv}) = y_t\right)$$

$$\overset{(a)}{\leq} \Pr\left(\bigwedge_{D_{\theta_j} \in \mathcal{D}_s} \left(\ell_{D_{\theta_j}}(\boldsymbol{x}^{adv}, y) > \ell_{D_{\theta_j}}(\boldsymbol{x}^{adv}, y_t)\right) \wedge \left(\ell_{\bar{V}}(\boldsymbol{x}^{adv}, y) > \ell_{\bar{V}}(\boldsymbol{x}^{adv}, y_t)\right)\right)$$

$$\overset{(b)}{\leq} \Pr\left(\left(\frac{1}{|\mathcal{D}_s|} \sum_{D_{\theta_j} \in \mathcal{D}_s} \boldsymbol{\delta} \cdot \nabla_{\boldsymbol{x}} \ell_{D_{\theta_j}}(\boldsymbol{x}, y) \geq \frac{1}{|\mathcal{D}_s|} \sum_{D_{\theta_j} \in \mathcal{D}_s} c_{D_{\theta_j}}\right) \wedge \left(\boldsymbol{\delta} \cdot \nabla_{\boldsymbol{x}} \ell_{\bar{V}}(\boldsymbol{x}, y) \geq c_{\bar{V}}\right)\right)$$

$$\overset{(c)}{\leq} \Pr\left(\left(\frac{1}{|\mathcal{D}_s|} \sum_{D_{\theta_j} \in \mathcal{D}_s} c_{D_{\theta_j}} \leq \epsilon' \sqrt{\frac{1 + S(\mathcal{D}_s, \bar{V})}{2}} \left\|\frac{1}{|\mathcal{D}_s|} \sum_{D_{\theta_j} \in \mathcal{D}_s} \nabla_{\boldsymbol{x}} \ell_{D_{\theta_j}}(\boldsymbol{x}, y)\right\|_2\right)\right.$$

$$\left.\bigcup\left(c_{\bar{V}} \leq \epsilon' \sqrt{\frac{1 + S(\mathcal{D}_s, \bar{V})}{2}} \left\|\nabla_{\boldsymbol{x}} \ell_{\bar{V}}(\boldsymbol{x}, y)\right\|_2\right)\right)$$

$$\leq \Pr\left(\frac{1}{|\mathcal{D}_s|} \sum_{D_{\theta_j} \in \mathcal{D}_s} c_{D_{\theta_j}} \leq \epsilon' \sqrt{\frac{1 + S(\mathcal{D}_s, \bar{V})}{2}} \left\|\frac{1}{|\mathcal{D}_s|} \sum_{D_{\theta_j} \in \mathcal{D}_s} \nabla_{\boldsymbol{x}} \ell_{D_{\theta_j}}(\boldsymbol{x}, y)\right\|_2\right)$$

$$+ \Pr\left(c_{\bar{V}} \leq \epsilon' \sqrt{\frac{1 + S(\mathcal{D}_s, \bar{V})}{2}} \left\|\nabla_{\boldsymbol{x}} \ell_{\bar{V}}(\boldsymbol{x}, y)\right\|_2\right), \tag{29}$$

where $S(\mathcal{D}_s, \bar{V})$ is the cosine similarity between $\frac{1}{|\mathcal{D}_s|} \sum_{i=1}^{|\mathcal{D}_s|} \nabla_{\boldsymbol{x}} \ell_{D_{\theta_j}}(\boldsymbol{x}, y)$ and $\nabla_{\boldsymbol{x}} \ell_{\bar{V}}(\boldsymbol{x}, y)$. Inequality $(a)$ is using Eq. 23 and Eq. 24, inequality $(b)$ is due to the fact that $\Pr((A > a) \cap (B > b)) \leq \Pr((A + B) > (a + b))$ and using Eq. 27 and Eq. 28. The inequality $(c)$ is a result of Lemma 5: either

$$\boldsymbol{\delta} \cdot \frac{\frac{1}{|\mathcal{D}_s|} \sum_{D_{\theta_j} \in \mathcal{D}_s} \nabla_{\boldsymbol{x}} \ell_{D_{\theta_j}}(\boldsymbol{x}, y)}{\|\frac{1}{|\mathcal{D}_s|} \sum_{D_{\theta_j} \in \mathcal{D}_s} \nabla_{\boldsymbol{x}} \ell_{D_{\theta_j}}(\boldsymbol{x}, y)\|} \leq \|\boldsymbol{\delta}\|_2 \sqrt{\frac{1 + S(\mathcal{D}_s, \bar{V})}{2}}$$

or

$$\boldsymbol{\delta} \cdot \frac{\nabla_{\boldsymbol{x}} \ell_{\bar{V}}(\boldsymbol{x}, y)}{\|\nabla_{\boldsymbol{x}} \ell_{\bar{V}}(\boldsymbol{x}, y)\|} \leq \|\boldsymbol{\delta}\|_2 \sqrt{\frac{1 + S(\mathcal{D}_s, \bar{V})}{2}}.$$

We observe that by $\beta$-smoothness condition of the loss function,

$$c_{D_{\theta_j}} = \ell_{D_{\theta_j}}(\boldsymbol{x}^{adv}, y_t) - \ell_{D_{\theta_j}}(\boldsymbol{x}, y) - \frac{\beta}{2} \|\boldsymbol{\delta}\|_2^2 \geq \ell_{D_{\theta_j}}(\boldsymbol{x}, y_t) + \boldsymbol{\delta} \cdot \nabla_{\boldsymbol{x}} \ell_{D_{\theta_j}}(\boldsymbol{x}, y_t) - \ell_{D_{\theta_j}}(\boldsymbol{x}, y) - \beta \|\boldsymbol{\delta}\|_2^2$$

Thus,

$$\Pr\Big(\frac{1}{|\mathcal{D}_s|}\sum_{D_{\theta_j}\in\mathcal{D}_s}c_{D_{\theta_j}}\le\epsilon'\sqrt{\frac{1+S(\mathcal{D}_s,\bar{V})}{2}}\Big\|\frac{1}{|\mathcal{D}_s|}\sum_{D_{\theta_j}\in\mathcal{D}_s}\nabla_{\boldsymbol{x}}\ell_{D_{\theta_j}}(\boldsymbol{x},y)\Big\|_2\Big)$$

$$\le\Pr\Big(\frac{1}{|\mathcal{D}_s|}\sum_{D_{\theta_j}\in\mathcal{D}_s}\big(\ell_{D_{\theta_j}}(\boldsymbol{x},y_t)+\boldsymbol{\delta}\cdot\nabla_{\boldsymbol{x}}\ell_{D_{\theta_j}}(\boldsymbol{x},y_t)-\ell_{D_{\theta_j}}(\boldsymbol{x},y)-\beta\|\boldsymbol{\delta}\|_2^2\big)$$

$$\le\epsilon'\sqrt{\frac{1+S(\mathcal{D}_s,\bar{V})}{2}}\Big\|\frac{1}{|\mathcal{D}_s|}\sum_{D_{\theta_j}\in\mathcal{D}_s}\nabla_{\boldsymbol{x}}\ell_{D_{\theta_j}}(\boldsymbol{x},y)\Big\|_2\Big)$$

$$\le\Pr\Big(\frac{1}{|\mathcal{D}_s|}\sum_{D_{\theta_j}\in\mathcal{D}_s}\big(\ell_{D_{\theta_j}}(\boldsymbol{x},y_t)-\|\boldsymbol{\delta}\|_2\|\nabla_{\boldsymbol{x}}\ell_{D_{\theta_j}}(\boldsymbol{x},y_t)\|_2-\ell_{D_{\theta_j}}(\boldsymbol{x},y)-\beta\|\boldsymbol{\delta}\|_2^2\big)\le\epsilon'B\sqrt{\frac{1+S(\mathcal{D}_s,\bar{V})}{2}}\Big)$$

$$=\Pr\Big(\frac{1}{|\mathcal{D}_s|}\sum_{D_{\theta_j}\in\mathcal{D}_s}\ell_{D_{\theta_j}}(\boldsymbol{x},y)+\epsilon'B\sqrt{\frac{1+S(\mathcal{D}_s,\bar{V})}{2}}\ge\frac{1}{|\mathcal{D}_s|}\sum_{D_{\theta_j}\in\mathcal{D}_s}\ell_{D_{\theta_j}}(\boldsymbol{x},y_t)-\epsilon'B-\beta\epsilon'^2\Big)$$

$$\le\frac{\mathbb{E}\Big[\frac{1}{|\mathcal{D}_s|}\sum_{D_{\theta_j}\in\mathcal{D}_s}\ell_{D_{\theta_j}}(\boldsymbol{x},y)+\epsilon'B\sqrt{\frac{1+S(\mathcal{D}_s,\bar{V})}{2}}\Big]}{\frac{1}{|\mathcal{D}_s|}\sum_{D_{\theta_j}\in\mathcal{D}_s}\ell_{D_{\theta_j}}(\boldsymbol{x},y_t)-\epsilon'B-\beta\epsilon'^2}$$

$$\le\frac{\xi+\epsilon'B\sqrt{\frac{1+\mathbb{E}[S(\mathcal{D}_s,\bar{V})]}{2}}}{\frac{1}{|\mathcal{D}_s|}\sum_{D_{\theta_j}\in\mathcal{D}_s}\ell_{D_{\theta_j}}(\boldsymbol{x},y_t)-\epsilon'B-\beta\epsilon'^2}, \tag{30}$$

where $\xi=\mathbb{E}_{D_{\theta_i}\sim\mathcal{D}}\big[\ell_{D_{\theta_i}}(\boldsymbol{x},y)\big]$. Similarly for $\bar{V}$,

$$\Pr\Big(c_{\bar{V}}\le\epsilon'\sqrt{\frac{1+S(\mathcal{D}_s,\bar{V})}{2}}\Big\|\boldsymbol{\delta}\cdot\nabla_{\boldsymbol{x}}\ell_{\bar{V}}(\boldsymbol{x},y)\Big\|_2\Big)\le\frac{\xi+\epsilon'B\sqrt{\frac{1+\mathbb{E}[S(\mathcal{D}_s,\bar{V})]}{2}}}{\ell_V(\boldsymbol{x},y_t)-\epsilon'B-\beta\epsilon'^2}. \tag{31}$$

Hence,

$$\Pr(T_r(\mathcal{D}_s,\bar{V},\boldsymbol{x}^{adv},y_t)=1)\le\frac{\xi+\epsilon'B\sqrt{\frac{1+\mathbb{E}[S(\mathcal{D}_s,\bar{V})]}{2}}}{\frac{1}{|\mathcal{D}_s|}\sum_{D_{\theta_j}\in\mathcal{D}_s}\ell_{D_{\theta_j}}(\boldsymbol{x},y_t)-\epsilon'B-\beta\epsilon'^2}+\frac{\xi+\epsilon'B\sqrt{\frac{1+\mathbb{E}[S(\mathcal{D}_s,\bar{V})]}{2}}}{\ell_{\bar{V}}(\boldsymbol{x},y_t)-\epsilon'B-\beta\epsilon'^2},$$

where $\mathbb{E}[S(\mathcal{D}_s,\bar{V})]$ is upper bounded by using Lemma 6 as follows:

$$\mathbb{E}[S(\mathcal{D}_s,\bar{V})]\le\frac{B^2+\|\nabla_{\boldsymbol{x}}\ell_{\bar{V}}(\boldsymbol{x},y)\|_2^2-\frac{\lambda^2}{|\mathcal{D}_s|}}{2C\|\nabla_{\boldsymbol{x}}\ell_{\bar{V}}(\boldsymbol{x},y)\|_2}.$$

$\square$

## H    LIMITATIONS AND BROADER IMPACTS

**Limitations.**    While BAT demonstrates strong targeted transferability under single-surrogate constraints, it has several limitations. First, the computational cost increases approximately linearly with the number of discriminators, as shown in Fig. 6a, which may raise concern in resource-constrained environments. Second, although Tab. 16 shows that BAT is generally stable across different random pruning seeds, certain seeds or surrogate architectures may lead to higher variability, potentially affecting reliability. Third, as illustrated in Tab. 15, BAT's transferability declines under smaller perturbation budgets, indicating the sensitivity to the strength of the threat model. Finally, BAT is currently evaluated only under the $\ell_\infty$ perturbation constraint; its applicability to other settings (e.g., physical-world attacks) remains an open question. Addressing these limitations presents important opportunities for future research.

**Broader Impacts.** This work proposes BAT, a generative framework aimed at improving the targeted transferability of adversarial examples under single-surrogate constraints. The primary intent is to advance our understanding of adversarial robustness and transfer behavior, which can aid in designing more secure and generalizable machine learning systems. In particular, BAT highlights how small structural modifications (e.g., pruning) and confidence-aware training can lead to stronger transferable attacks, offering valuable insights for future defenses.

However, as with many works on adversarial attacks, there is potential for misuse. Techniques developed in BAT could be repurposed to generate stronger targeted attacks against real-world systems in domains such as biometric authentication or autonomous driving. To mitigate this risk, we limit our experiments to standard datasets (e.g., ImageNet) and do not release pretrained generators or plug-and-play attack pipelines. Any shared code will include disclaimers and be intended solely for research and defense-oriented applications.

We believe that responsibly studying the targeted transferability is necessary to anticipate and counter future adversarial threats, and we encourage the broader community to approach this space with similar care.

# I    VISUALIZATION OF ADVERSARIAL EXAMPLES

In this section, we present multiple adversarial examples generated by the three variants of the proposed BAT along with their corresponding perturbations for different target classes, as illustrated in Fig. 8 to Fig. 11.

# J    MORE ATTENTION HEATMAPS

We present additional attention heatmaps for four different input images and their corresponding adversarial examples, for target class#100, generated using the I-FGSM (Kurakin et al., 2018) method, as illustrated in Fig. 12 and Fig. 13. These adversarial examples are crafted on pretrained models on ImageNet-1K (Russakovsky et al., 2015), including ResNet18, ResNet50, VGG16, and VGG19, along with five pruned versions of each. From Fig. 12 and Fig. 13, it is clear that the attention heatmaps differ across the pruned models derived from the pretrained models, reflecting diverse decision boundaries resulting from the pruning process.

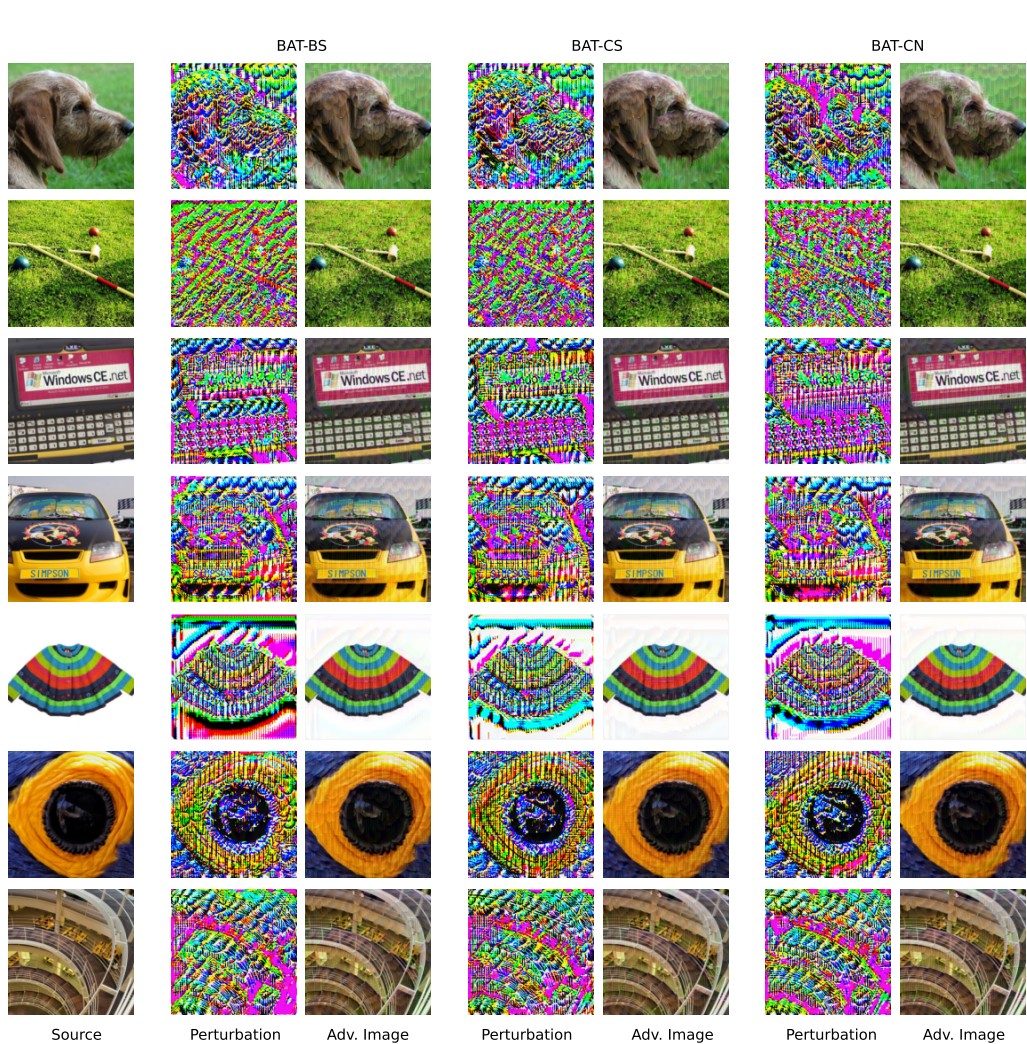

Figure 8: Visualization of adversarial examples and their corresponding perturbations for the target class "**Vulture**" on the ImageNet-1K dataset, generated by the proposed BAT methods using ResNet50 as the surrogate under no domain shift ($\mathcal{P}=\mathcal{Q}$).

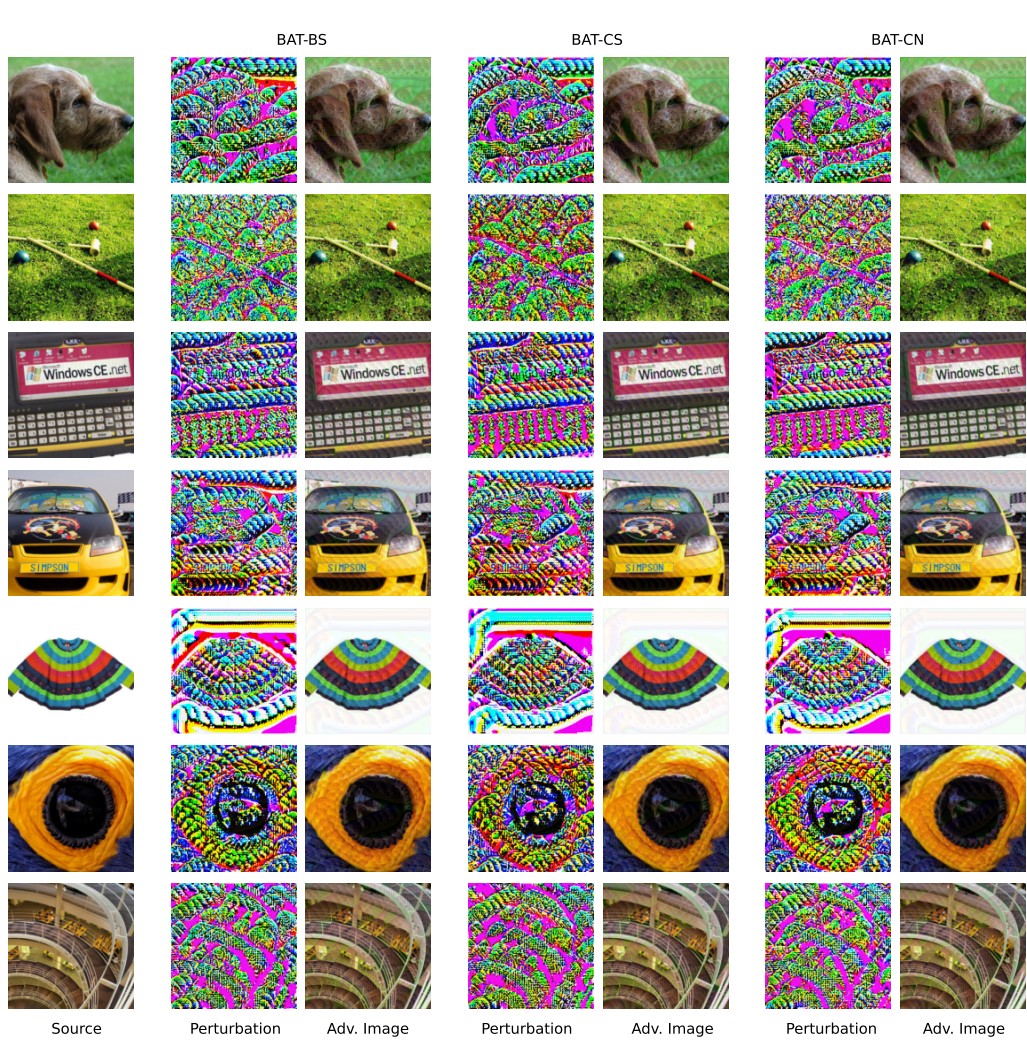

Figure 9: Visualization of adversarial examples and their corresponding perturbations for the target class "**Night snake**" on the ImageNet-1K dataset, generated by the proposed BAT methods using ResNet50 as the surrogate under no domain shift ($\mathcal{P}=\mathcal{Q}$).

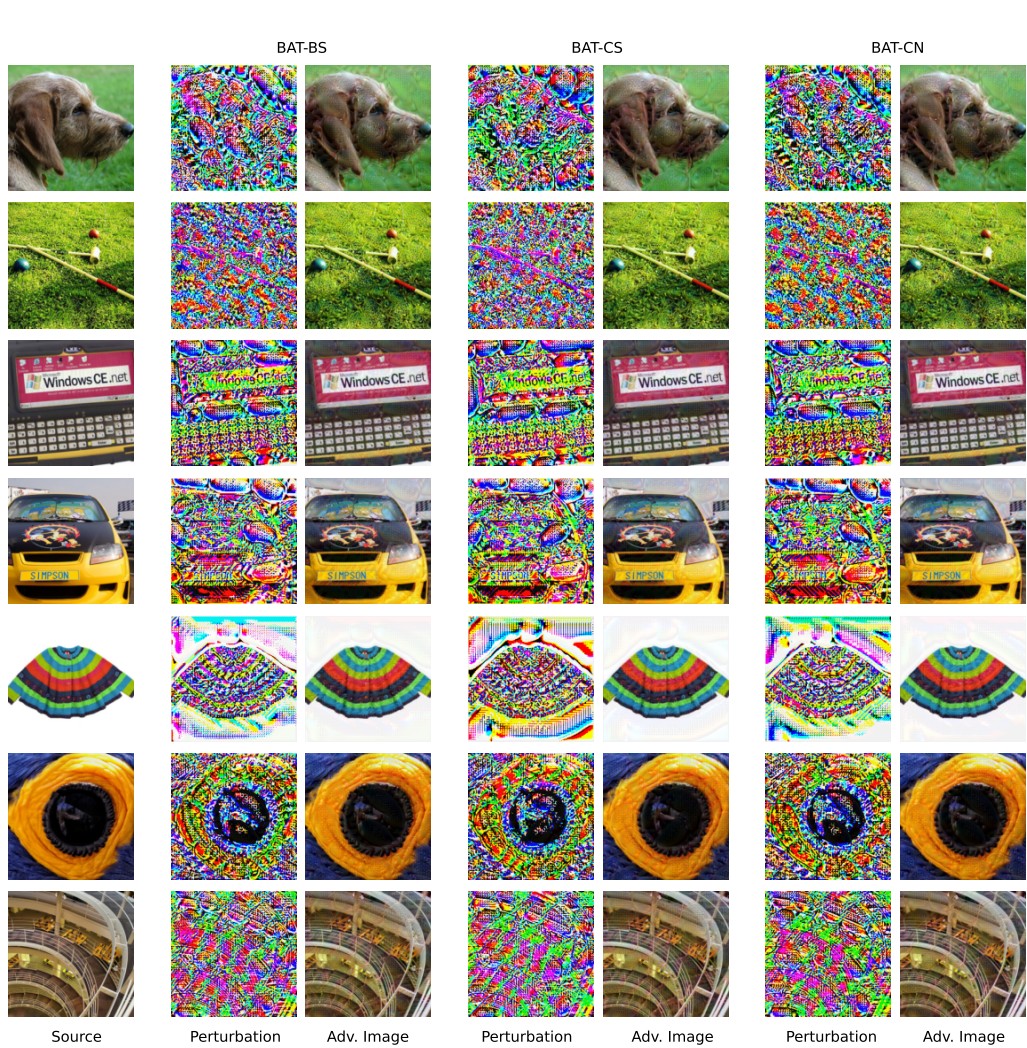

Figure 10: Visualization of adversarial examples and their corresponding perturbations for the target class "**Crayfish**" on the ImageNet-1K dataset, generated by the proposed BAT methods using ResNet50 as the surrogate under no domain shift ($\mathcal{P}=\mathcal{Q}$).

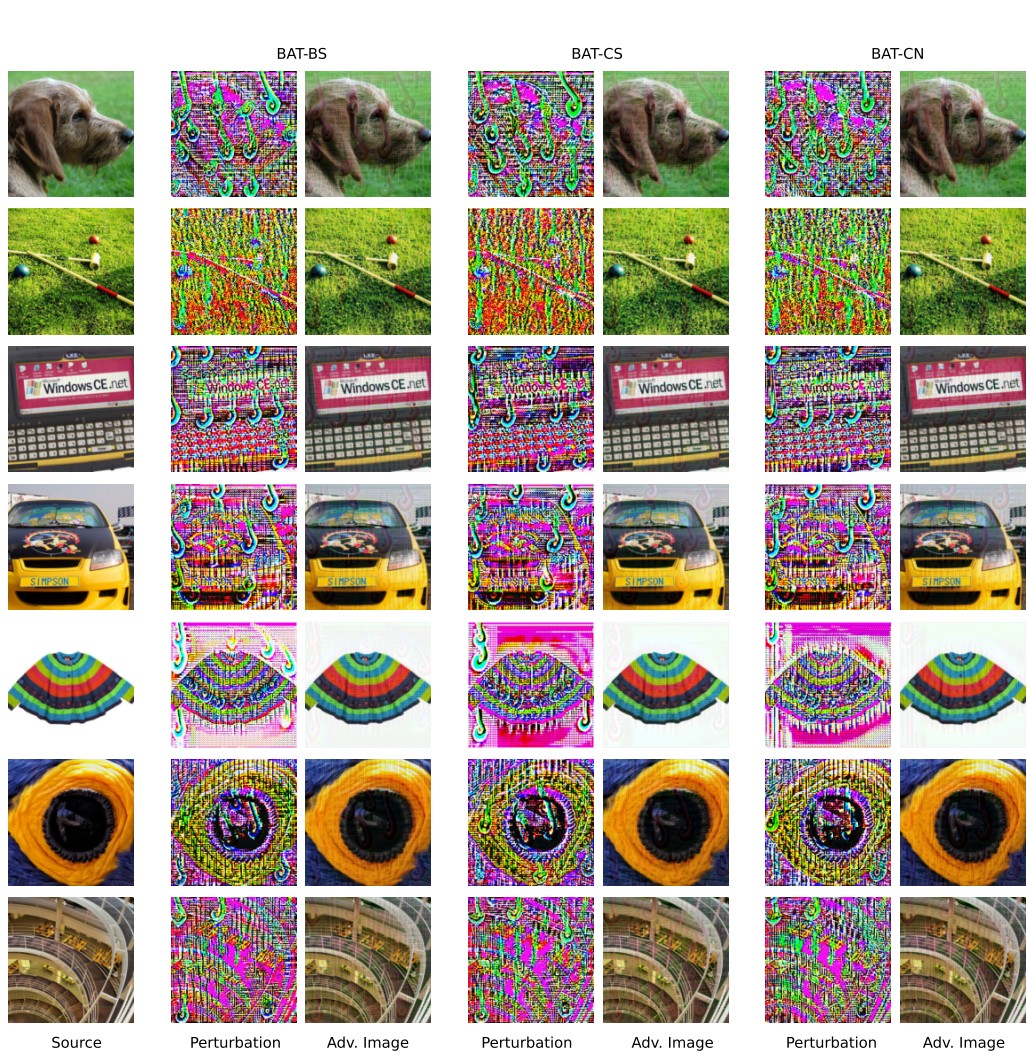

Figure 11: Visualization of adversarial examples and their corresponding perturbations for the target class "**Hook**" on the ImageNet-1K dataset, generated by the proposed BAT methods using ResNet50 as the surrogate under no domain shift ($\mathcal{P}=\mathcal{Q}$).

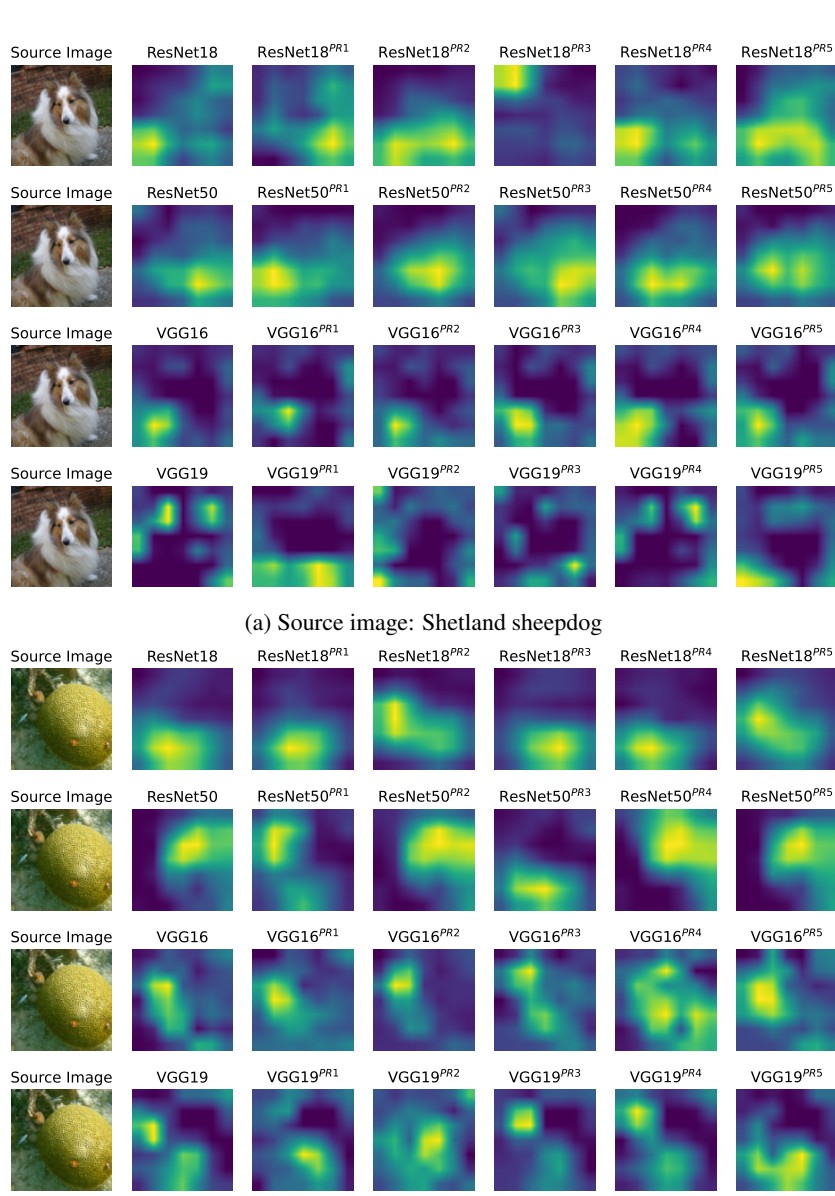

(a) Source image: Shetland sheepdog

(b) Source image: Brain coral

Figure 12: Attention heatmaps, obtained using Grad-CAM (Selvaraju et al., 2017), are shown for adversarial images of input classes Shetland sheepdog and Brain coral. These adversarial examples are crafted with target class #100 of ImageNet-1K on different classifiers and their corresponding pruned versions.

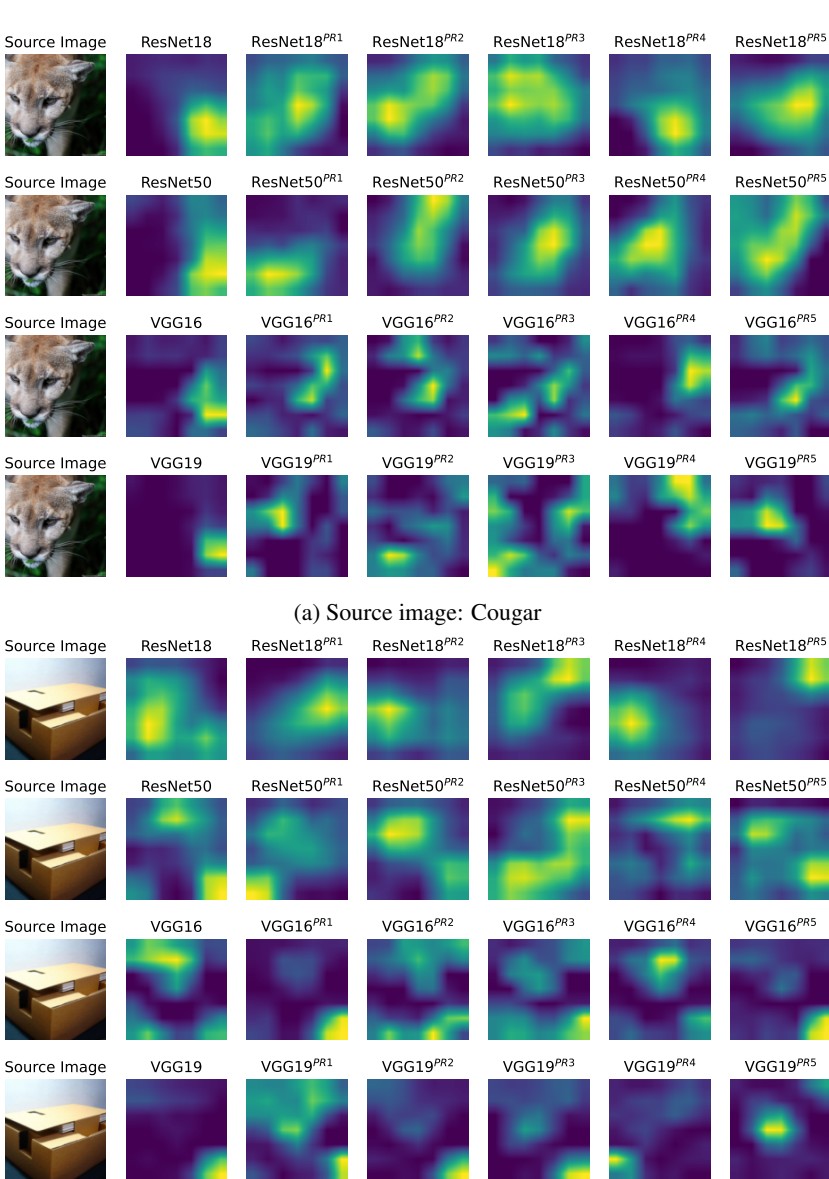

Figure 13: Attention heatmaps, obtained using Grad-CAM (Selvaraju et al., 2017), are shown for adversarial images of input classes Cougar and Carton. These adversarial examples are crafted with target class #100 of ImageNet-1K on different classifiers and their corresponding pruned versions.

