# OpenReview forum: "Boosting Targeted Adversarial Transferability: A Generative Approach Guided by Core Target Samples"
_ICLR.cc/2026/Conference — Submitted to ICLR 2026_

### Official Review · Reviewer_Q6HX · 2025-10-20

**Soundness:** 3
**Presentation:** 3
**Contribution:** 3
**Rating:** 6
**Confidence:** 4

**Summary:**

he paper introduces BAT, a generative targeted-transfer attack that aligns adversarial examples to a small “core” set of high-confidence target samples in both output and intermediate feature spaces. To overcome single-surrogate overfitting, BAT builds a self-ensemble of frozen discriminators by pruning a single pretrained model, avoiding extra training while injecting boundary diversity. Three variants handle data availability: BAT-BS (best real target images), BAT-CS (confidence-crafted target references), and BAT-CN (references synthesized from noise), covering both P=Q and P≠Q regimes. On ImageNet-1K and a Painting domain shift, BAT yields notably higher targeted TSR than strong iterative and generative baselines, with a simple theory linking transferability to ensemble size and its diminishing returns.

**Strengths:**

- Clear, modular method: dual-space alignment to high-confidence core targets plus pruned-ensemble discriminators to mitigate single-surrogate overfitting.
- Broad applicability: supports domain match and shift and target-data-guided vs. target-data-free training (BAT-CN works without real target images).
- Strong empirical gains: consistent TSR improvements over state of the art across many victim architectures, including evaluations against robustly trained models.
- Ablations isolate contributions of ensemble size and core-set selection and show monotonic improvements from baseline to full BAT variants.
- Theory offers lower/upper bounds explaining the role of ensemble size and why gains saturate, matching empirical trends.

**Weaknesses:**

- Statistical reporting is light: no multi-seed confidence intervals or seed-paired comparisons; robustness claims could be sensitive to randomness
- Threat-model breadth could be expanded: emphasis on $\ell_\infty$ at a fixed $\epsilon$; limited coverage of ℓ2 or structure-preserving perturbations in the main text
- The pruning-diversity assumption is plausible but under-analyzed: lack of explicit diversity metrics (e.g., gradient/decision disagreement) vs. multi-surrogate ensembles
- While theoretical support for a tighter $\ell_2$ constraint than $\ell_\infty$ holds, empirical testing would more strengthen the claim
- Limited discussion of compute/latency for training and inference, especially as |Ds| and core-set size scale

**Questions:**

- How stable are the reported gains across multiple seeds, and can you provide confidence interval range comparisons vs. the strongest baseline?
- Can you quantify ensemble diversity induced by pruning (e.g., gradient cosine, boundary disagreement) and relate it to TSR?
- Do the main trends persist for a second norm (ℓ2) and a simple spatial/color threat without heavy retuning?
- What are the compute/latency costs for training and inference as |Ds| and core-set size vary, and where is the best accuracy/efficiency frontier?

---

> ### Author Response · Authors · 2025-11-23
> **Responses to Reviewer Q6HX (1/3)**
>
> We thank the reviewer for the detailed and insightful feedback. Below we address each weakness and question.
> Numbers in parentheses in the tables indicate the absolute TSR gain of BAT variant (in percentage points) over the best baseline in each column.
>
> ---
>
> ### W1 / Q1 — Statistical stability and multi-seed comparison
>
> BAT-BS remains consistently better than M3D across seeds; its variance is comparable to M3D while achieving higher mean TSR on most victims.
>
> In the original draft, Table 12 already reported 5-seed results for BAT-BS. Following the reviewer’s suggestion, we now also run M3D under the same 5 seeds and report mean $\pm$ standard deviation of TSR (%) using ResNet-50 as the surrogate and the no-domain-shift setting on ImageNet-1K:
>
> | Attack  | RN18             | RN50*            | RN101            | DN121            | DN161            | VGG16-BN         | VGG19-BN         | MN-V2            | ViT-B/16         |
> |---------|------------------|------------------|------------------|------------------|------------------|------------------|------------------|------------------|------------------|
> | M3D     | 86.52 $\pm$ 1.75 | 95.79 $\pm$ 0.22 | 88.63 $\pm$ 0.92 | 88.67 $\pm$ 1.43 | 89.03 $\pm$ 0.99 | 85.32 $\pm$ 1.22 | 84.15 $\pm$ 0.95 | 81.54 $\pm$ 1.59 | 49.69 $\pm$ 1.38 |
> | BAT-BS  | 89.49 $\pm$ 2.18 | 98.50 $\pm$ 0.25 | 94.61 $\pm$ 1.15 | 93.35 $\pm$ 0.71 | 91.81 $\pm$ 1.42 | 91.39 $\pm$ 1.86 | 89.09 $\pm$ 1.03 | 88.10 $\pm$ 3.64 | 40.61 $\pm$ 1.60 |
>
> We have included the above multi-seed table in the appendix (Table 16 in the revision) and explicitly referenced it in the main text.
>
> ---
>
> ### W3 / Q2 — Diversity of pruned ensembles and its relation to TSR
>
> We now **quantify diversity** via decision disagreement between the base ResNet-50 and its pruned variants, and show that **moderate pruning increases both diversity and TSR**, whereas heavy pruning yields excessive disagreement and **reduces TSR**. We further compare these disagreement levels to those obtained from using *different architectures* as surrogates.
>
> We measure **decision disagreement** between the unpruned ResNet-50 and its randomly pruned variants for different pruning ratios $p_r$. For each $p_r \in \{0.01, 0.02, 0.05, 0.10\}$, we create 10 randomly pruned models and, on 5000 ImageNet validation images, compute the fraction of predictions that differ from the base model (averaged over the 10 models). We then report BAT’s TSR at each $p_r$. The discriminator ensemble always includes:
>
> - the **unpruned ResNet-50**,
> - a **60% L1-pruned ResNet-50** (disagreement 24.72%), and
> - **three additional randomly pruned models** at the specified $p_r$.
>
> The results are:
>
> | $p_r$                          | 0.0 | 0.01  | 0.02  | 0.05  | 0.10  |
> |--------------------------------|-----|-------|-------|-------|-------|
> | Decision disagreement (%)      | 0.0 | 15.06 | 23.16 | 48.12 | 74.16 |
> | TSR (%)                        | 78.35 | 83.06 | 85.46 | 78.59 | 72.62 |
>
> To contextualize these numbers, we also compute decision disagreement between ResNet-50 and ten standard architectures (`resnet18`, `resnet101`, `vgg16`, `vgg19`, `vgg16_bn`, `vgg19_bn`, `densenet121`, `densenet169`, `mobilenet_v2`, `vit_b_16`) on the same 1000 images. The average inter-architecture disagreement is **22.35%**, which is very close to the disagreement at $p_r = 0.02$ (23.16%) and to the 60% L1-pruned model (24.72%). This suggests that *moderate* pruning induces a level of diversity comparable to that obtained by using heterogeneous architectures as surrogates.
>
> Two clear patterns emerge:
>
> 1. **Moderate pruning improves both diversity and TSR.**
>    Moving from $p_r = 0$ (no random pruning) to $p_r = 0.01$ and $0.02$ increases disagreement from 0% to 15.06% and 23.16%, and TSR from 78.35% to 83.06% and 85.46%. These disagreement levels are close to the 22.35% inter-architecture disagreement, indicating that moderate additional diversity—similar in magnitude to using different architectures—is beneficial for targeted transfer.
>
> 2. **Over-pruning degrades TSR despite high disagreement.**
>    Increasing $p_r$ further to 0.05 and 0.10 pushes disagreement up to 48.12% and 74.16%, but **TSR drops** to 78.59% and 72.62%. At these levels, pruned models lose too much discriminative power and provide noisy gradients, so transfer deteriorates even though “diversity” (disagreement) is high.
>
> In the revision, we replace the previous Table 13 from the original submission with this diversity–TSR table (now Table 7), together with the accompanying discussion, making the pruning–diversity assumption empirically grounded.
>
> ---

---

> > ### Author Response · Authors · 2025-11-23
> > **Responses to Reviewer Q6HX (2/3)**
> >
> > ### W2, W4, Q3 — Threat-model breadth and empirical support
> >
> > Without any retraining, BAT-CS **retains its advantage over M3D under an $\ell_2$ constraint** by changing only the projection step, and **also maintains higher TSR** under a simple spatial+color threat (ColorJitter). This shows that our main trends are **not specific to $\ell_\infty$ at a fixed $\epsilon$**.
> >
> > **$\ell_2$ evaluation without retraining**
> >
> > Our framework is norm-agnostic; only the projection step depends on the norm. To test this, we reuse the **same generators trained under the $\ell_\infty$ threat model ($\epsilon = 16/255$)** for both M3D and BAT-CS and change only the projection at test time. For each source image $x_s$ with generated perturbation $\delta$, we construct an $\ell_2$-constrained adversarial example:
> > $$
> > x_{\text{adv}} = x_s + \epsilon_2 \frac{\delta}{\lVert \delta \rVert_2}, \quad \epsilon_2 = 16,
> > $$
> > and evaluate targeted transfer to 9 victim models on 1000 ImageNet images. The resulting TSR (%) is:
> >
> > | Attack  | RN18 | RN50 | RN101 | DN121 | DN161 | VGG16-BN | VGG19-BN | MN-V2 | ViT-B/16 | Average |
> > |---------|------|------|-------|-------|-------|----------|----------|-------|----------|---------|
> > | M3D     | 82.19 | 95.48 | 85.28 | 82.85 | 80.67 | 79.05 | 80.10 | 78.72 | 50.12 | 79.38 |
> > | BAT-CS  | **91.46** (+9.27) | **97.74** (+2.26) | **92.03** (+6.75) | **90.59** (+7.74) | **91.24** (+10.57) | **93.00** (+13.95) | **91.81** (+11.71) | **86.81** (+8.09) | **48.93** (−1.19) | **87.07** (+7.69) |
> >
> > Even **without retraining**, BAT-CS improves the average TSR from 79.38% to 87.07%, mirroring the $\ell_\infty$ trend and supporting our theoretical discussion that a well-aligned perturbation direction can remain effective under an $\ell_2$ constraint.
> >
> > **Simple spatial/color threat without retuning**
> >
> > The main paper (Table 8) already evaluates robustness under standard spatial transformations (resize & crop, flip, rotation). To further address the reviewer’s question, we additionally apply **ColorJitter** at test time to both M3D and BAT-CS, again using the same $\ell_\infty$-trained generators, with no extra tuning:
> >
> > ```python
> > transforms.ColorJitter(
> >     brightness=0.4,
> >     contrast=0.4,
> >     saturation=0.4,
> >     hue=0.1
> > )
> > ```
> >
> > and measure TSR (%) on the same set of victim models:
> >
> >
> > | Attack | RN18  | RN50  | RN101 | DN121 | DN161 | VGG16-BN | VGG19-BN | MN-V2 | ViT-B/16 | Average |
> > |--------|-------|-------|-------|-------|-------|----------|----------|-------|----------|---------|
> > | M3D    | 83.76 | 95.57 | 87.01 | 86.43 | 87.30 | 83.28    | 83.82    | 82.08 | 59.30    | 83.17   |
> > | BAT-CS | **92.00** (+8.24) | **97.94** (+2.37) | **93.03** (+6.02) | **93.64** (+7.21) | **93.14** (+5.84) | **93.98** (+10.70) | **93.78** (+9.96) | **89.41** (+7.33) | **57.87** (−1.43) | **89.42** (+6.25) |
> >
> > BAT-CS again **improves the average TSR** from 83.17% to 89.42%, demonstrating that the relative gains persist under a simple combined spatial+color threat without any retuning. We have added the $\ell_2$-constraint table to Appendix C (Table 14) and referenced it in the main text.
> >
> > ---

---

> > > ### Author Response · Authors · 2025-11-23
> > > **Responses to Reviewer Q6HX (3/3)**
> > >
> > > ### W5 / Q4 — Compute/latency vs. $|\mathcal{D}_s|$ and core-set size; accuracy/efficiency frontier
> > >
> > >
> > >
> > > Training cost scales **approximately linearly** with ensemble size $|\mathcal{D}_s|$, core-set size has negligible impact on total compute in our regime, and **inference uses only a single generator forward pass**, independent of $|\mathcal{D}_s|$ and core-set size. Our experiments show that moderate $|\mathcal{D}_s|$ gives a good accuracy/efficiency trade-off before TSR gains saturate.
> > >
> > > **Training cost vs. ensemble size**
> > >
> > > - Figure 6(a) (appendix) reports training time per target class as $|\mathcal{D}_s|$ varies for BAT-BS.
> > > - The trend is approximately linear: training time increases from $\approx 4.3$ hours (1 discriminator) to $\approx 7.9$ hours (5 discriminators).
> > > - This confirms that $|\mathcal{D}_s|$ is the dominant cost driver.
> > >
> > > **Effect of core target set size**
> > >
> > > - Core target samples form a pool from which we draw mini-batches; increasing the pool size does not change the per-iteration cost.
> > > - For BAT-CS and BAT-CN, the one-time construction of 300 core samples per target class takes $\approx 2$ minutes, which is negligible compared to multi-hour generator training.
> > > - Within the considered range, core-set size does not materially affect the overall compute.
> > >
> > > **Comparison with other generative methods**
> > >
> > > - Figure 6(b) compares training time per target class across generative attacks.
> > > - BAT-BS/BAT-CS/BAT-CN are faster than CGNC and M3D and only slightly slower than TTP, despite using an ensemble of discriminators.
> > > - Thus, our robustness gains do not come at a prohibitive training cost.
> > >
> > > **Inference latency**
> > >
> > > - At attack time, we use only the trained generator: one forward pass through $G_\Phi$ plus a projection step.
> > > - The discriminator ensemble and core-set selection are not used at inference.
> > > - Consequently, inference latency is independent of $|\mathcal{D}_s|$ and core-set size and is substantially smaller than iterative targeted attacks that require many forward–backward steps per image.
> > >
> > > **Accuracy/efficiency frontier**
> > >
> > > - Combining Fig. 4(a) and Fig. 6(a), we observe:
> > >   - Moving from $|\mathcal{D}_s| = 1$ to a moderate ensemble size yields large TSR gains with a roughly linear increase in training time.
> > >   - Beyond the ensemble size used in our main experiments, TSR gains saturate quickly, while compute continues to grow.
> > >
> > > We have revised the discussion provided in Appendix D to include the aforementioned points.

---

### Official Review · Reviewer_Np53 · 2025-10-30

**Soundness:** 3
**Presentation:** 3
**Contribution:** 2
**Rating:** 4
**Confidence:** 4

**Summary:**

This paper presents BAT,  a generative attack framework for targeted adversarial transfer under the constraint of a single surrogate. BAT is based on aligning both output and intermediate feature spaces of generated adversarial examples to those of a curated set of high-confidence core target samples. The key contribution is the creation of a diverse, frozen discriminator ensemble by pruning a single pretrained surrogate, thereby avoiding the need for multiple trained surrogates. The approach is evaluated in both domain-matched and domain-shift scenarios and supports both target-data-guided and target-data-free settings. Through extensive experiments and theoretical analyses, BAT is shown to improve targeted transferability over prior methods.

**Strengths:**

1. This work has a comprehensive evaluation  under both no-shift and domain-shift settings  across diverse architectures.

2.  Detailed ablations (Figure 5, Table 6) help EXPLAIN where performance gains come from,  diversified discriminators, core target selection.

3.  The section 3 and algorithm 1 and 2 in the appendix and clear and easy to understand.

4. The self-ensemble is simple and effective.

**Weaknesses:**

1. The description of the generator architecture is under-specified.  Since it is an important component of the proposed work,  the authors should provide a detailed description.

2. It is easy to convert some untargeted transferable attacks (e.g., admix-DT ) to the targeted ones. The authors should compare with them


[1] Xiaosen Wang, Xuanran He, Jingdong Wang, and Kun He. Admix: Enhancing the transferability of adversarial attacks. In Proceedings of the IEEE/CVF International Conference on Computer Vision, pp. 16158–16167, 2021.

**Questions:**

1. Augmentation strategies can also be viewed as a form of self-ensemble. Could data-level self-ensemble via various augmentations achieve better performance compared to the model-level self-ensemble achieved by pruning? if data-level self-ensemble and model-level self-ensemble are combined, can the TSR be further improved?

2. While the paper presents an interesting framework for boosting targeted transferability, I remain concerned about the level of novelty. The idea of ensembling surrogates to improve transferability is well established (e.g.,[2]); the use of feature-space alignment is also explored (e.g., [3]);  The notion of target samples as anchors is also explored (e.g., TTAA). So is this work a recombination of existing techniques into the generative targeted-transfer setting? I am not sure whether I understand correctly.


[2] Hung-Jui Wang, Yu-Yu Wu, and Shang-Tse Chen. Enhancing targeted attack transferability via diversified weight pruning. In Proceedings of the IEEE/CVF Conference on Computer Vision and Pattern Recognition, pp. 2904–2914, 2024a.

[3] Zhipeng Wei, Jingjing Chen, Zuxuan Wu, and Yu-Gang Jiang. Enhancing the self-universality for transferable targeted attacks. In Proceedings of the IEEE/CVF conference on computer vision and pattern recognition, pp. 12281–12290, 2023.

---

> ### Author Response · Authors · 2025-11-23
> **Responses to Reviewer Np53 (1/3)**
>
> We thank the reviewer for the constructive comments and for recognizing the strengths of our evaluation, ablations, and self-ensemble design. Below we address the weaknesses and questions in detail.
>
> ---
>
> ### W1. “The description of the generator architecture is under-specified.”
>
> Our generator uses **exactly the same ResNet-style architecture as prior generative targeted-transfer attacks TTP and M3D**, with all architectural hyperparameters unchanged. We have now provided this architecture in the revision in Appendix F to make the design fully transparent.
>
> For clarity, our generator $G_\Phi$ is a lightweight encoder–residual–decoder network:
>
> - **Input / output:**
>   Takes a $3 \times H \times W$ source image and outputs a $3 \times H \times W$ adversarial image.
>
> - **Encoder (down-sampling):**
>   - block1: `ReflectionPad2d(3)` → 7×7 conv (3→64, stride 1) → BN → ReLU.
>   - block2: 3×3 conv (64→128, stride 2) → BN → ReLU.
>   - block3: 3×3 conv (128→256, stride 2) → BN → ReLU.
>
> - **Residual bottleneck:**
>   6 residual blocks at 256 channels. Each block:
>   `ReflectionPad2d(1)` → 3×3 conv → BN → ReLU → Dropout(0.5) → `ReflectionPad2d(1)` → 3×3 conv → BN,
>   with a skip connection $x \mapsto x + \mathrm{block}(x)$.
>
> - **Decoder (up-sampling):**
>   - upsampl1: ConvTranspose2d (256→128, kernel 3, stride 2, padding 1, `output_padding=1`) → BN → ReLU.
>   - upsampl2: ConvTranspose2d (128→64, kernel 3, stride 2, padding 1, `output_padding=1`) → BN → ReLU.
>
> - **Output head:**
>   `ReflectionPad2d(3)` → 7×7 conv (64→3), followed by $(\tanh(\cdot) + 1)/2$ to map pixels to $[0, 1]$. We then project onto the $ \ell_\infty $-ball around the source image as described in Sec. 3.
>
> This generator is significantly lighter than the CGNC generator (which uses cross-attention, conditioning, and spectral normalization).
>
> ---
>
> ### W2. “It is easy to convert some untargeted transferable attacks (e.g., Admix-DT) to the targeted ones. The authors should compare with them.”
>
> We implemented **targeted variants of Admix and BSR** by changing only the loss to targeted cross-entropy, and observed that—even with near-perfect white-box success—they still exhibit **weak black-box targeted transfer**, consistent with our claim that iterative targeted transfer attacks do **not** yield strong targeted transfer.
>
> Following the original implementations, we convert **Admix** [1] and **BSR** [a] from untargeted to targeted attacks by simply replacing the loss with a targeted cross-entropy objective, while keeping all other hyperparameters (step size, number of iterations, admixing / block-shuffle / rotation operations) unchanged. We use **ResNet-50 as the surrogate** and evaluate TSR considering 500 ImageNet-1K test samples under the same $ \ell_\infty = 16/255 $ budget.
>
> | Attack | RN18  | RN50*  | RN101 | DN121 | DN161 | VGG16-BN | VGG19-BN | MobileNet-V2 | ViT-B/16 | Avg (all) |
> |--------|-------|--------|-------|-------|-------|----------|----------|--------------|----------|-----------|
> | Admix  | 22.14 | 99.86  | 33.14 | 29.62 | 26.82 | 16.22    | 16.10    | 17.22        | 2.58     | 29.30     |
> | BSR    | 49.78 | 98.29  | 59.02 | 64.47 | 62.22 | 48.56    | 43.09    | 36.82        | 4.31     | 51.84     |
> | BAT-CS | **93.78** (+44.00) | **98.78*** (+0.49) | **95.22** (+36.20) | **94.16** (+29.69) | **93.31** (+31.09) | **94.45** (+45.89) | **94.04** (+50.95) | **86.60** (+49.78) | **50.45** (+46.14) | **88.98** (+37.14) |
>
> Numbers in parentheses in the tables indicate the absolute TSR gain of BAT-CS (in percentage points) over the best baseline in each column.
>
> ---
>
> >[1] Xiaosen Wang, Xuanran He, Jingdong Wang, and Kun He. “Admix: Enhancing the transferability of adversarial attacks.” ICCV, 2021.
> [a] Wang, Kunyu, et al. “Boosting adversarial transferability by block shuffle and rotation.” CVPR, 2024.
>
> ---

---

> > ### Author Response · Authors · 2025-11-23
> > **Responses to Reviewer Np53 (2/3)**
> >
> > ### Q1. Does the data-level self-ensemble improve the TSR further? Does this work consider both data-level self-ensemble and model-level self-ensemble?
> >
> > BAT already **combines data-level and model-level self-ensemble**: we use strong augmentations (as in prior generative attacks) together with a pruned discriminator ensemble.
> >
> > **Data-level self-ensemble is already used in our framework.**
> > Following prior generative attacks such as TTP, M3D, and CGNC, we employ **data-level self-ensemble** during generator training. For each source image $x_s$, we construct augmented views using:
> >
> > - A standard augmentation pipeline (identical to the baselines), e.g.:
> >   $$
> >   \text{RandomResizedCrop} \rightarrow \text{RandomHorizontalFlip} \rightarrow \text{RandomApply(ColorJitter)} \rightarrow \text{RandomGrayscale} \rightarrow \text{ToTensor},
> >   $$
> >
> > - An additional **random rotation** augmentation (0°, 90°, 180°, 270°) applied per image in the batch, as done in the baselines.
> >
> > We feed both the original $x_s$ and its augmented views through the generator and **compute the loss on all of them**, matching dual-space feature similarity under the discriminator ensemble. This is effectively data-level self-ensemble.
> >
> > We did not explicitly emphasize this in the draft because using both the source image and its augmented versions to train the generator has become standard practice in generative targeted-transfer attacks, and we followed the baselines’ protocols for a fair comparison. In the revision, we have explicitly mentioned this in Sec. 3 (lines 162-164).
> >
> > ---

---

> > > ### Author Response · Authors · 2025-11-23
> > > **Responses to Reviewer Np53 (3/3)**
> > >
> > > ### Q2. “Is this work mainly a recombination of existing techniques (ensembles [2], feature alignment [3], target sample anchors/TTAA), or is there a genuinely new idea?”
> > >
> > > BAT is *not* a simple recombination of existing techniques.
> > > The single-surrogate pruning-based self-ensemble, the dual-space alignment (output and intermediate feature spaces) together with the core-target design (with/without access to target samples), and the accompanying theory form a coherent and, to our knowledge, novel framework for **boosting generative targeted transferability** in both no-domain-shift and domain-shift regimes, rather than a direct recombination of [2], [3], and TTAA.
> > >
> > > While BAT builds on known ideas (ensembles, feature-space considerations, target samples), our contribution lies in **how we integrate them** into a **class-aware generative targeted-transfer framework guided by core target samples and a pruned discriminator self-ensemble**. Concretely, BAT:
> > >
> > > 1. uses a **self-ensemble from a single surrogate via pruning** to obtain diverse yet competent discriminators;
> > > 2. performs **dual-space alignment** (KL over logits + cosine similarity in feature space) within a single **generator–discriminator training loop**;
> > > 3. crafts **core target samples** (BAT-BS/BAT-CS/BAT-CN) with or without access to real target images, steering the generator toward confident target regions; and
> > > 4. provides **theoretical insights** linking ensemble size to targeted transferability via explicit upper and lower bounds.
> > >
> > > Our ablations (Fig. 4, Fig. 5, Table 6) confirm that each of these choices yields tangible TSR gains; removing any of them degrades performance. To our knowledge, this combination has not been explored in the generative targeted-transfer setting.
> > >
> > > **Naive combination vs. full BAT.**
> > > Table 6 makes the progression from naive combinations to full BAT explicit:
> > >
> > > - **TTP** (single surrogate, logit-only KL) achieves 70.73% TSR (Table 1).
> > > - **Single-discriminator and dual-space alignment** yields 71.12% TSR (1st row, Table 6).
> > > - **Naive self-ensemble combination** ($|\mathcal{D}_s| = 5$, all targets, dual-space alignment, no cores) reaches 75.85% TSR.
> > > - **BAT-BS** (cores selected by ensemble confidence) with $|\mathcal{D}_s| = 5$ boosts TSR to 85.46%.
> > > - **BAT-CS/CN** (strengthened cores / core noise) further increase TSR to 88.98% and 87.84%, respectively.
> > >
> > > Thus, BAT’s gains come from the *moderately pruned* self-ensembles (Table 7), confidence-based cores, and dual-space alignment working together, rather than from simply stacking existing ingredients.
> > >
> > > **Relation to prior work.**
> > > Pruning-based targeted attacks such as [2] operate in a PGD-like, instance-specific *iterative* setting and do not train a reusable generator; TTAA does not exploit ensemble-level confidence or support the target-data–free regime; and feature-alignment methods like [3] study cosine alignment in a PGD-like iterative setting, where global and local views of the *same* added perturbation are aligned.
> > >
> > > In contrast, BAT combines
> > >  (i) a pruning-based self-ensemble derived from a *single* surrogate, (ii) ensemble-confidence–driven cores that target high-confidence adversarial regions and naturally extend to the target-data–free regime (BAT-CN), and  (iii) dual-space alignment between generated adversarial examples and core samples inside a unified generator–discriminator loop, supported by a theoretical analysis that explains the empirical $1/\sqrt{|\mathcal{D}_s|}$ saturation in Fig. 4(a).
> > >
> > > We therefore view BAT as a genuinely new framework for generative targeted transferability, rather than a direct recombination of [2], [3], and TTAA.
> > >
> > > ---
> > >
> > > >[2] Hung-Jui Wang, Yu-Yu Wu, and Shang-Tse Chen. *Enhancing targeted attack transferability via diversified weight pruning.* In Proceedings of the IEEE/CVF Conference on Computer Vision and Pattern Recognition, pp. 2904–2914, 2024a.
> > > [3] Zhipeng Wei, Jingjing Chen, Zuxuan Wu, and Yu-Gang Jiang. *Enhancing the self-universality for transferable targeted attacks.* In Proceedings of the IEEE/CVF Conference on Computer Vision and Pattern Recognition, pp. 12281–12290, 2023.

---

### Official Review · Reviewer_b9rh · 2025-11-01

**Soundness:** 3
**Presentation:** 3
**Contribution:** 3
**Rating:** 4
**Confidence:** 3

**Summary:**

The paper proposes BAT, a generative method to improve targeted adversarial transferability by training a generator to align perturbations with curated high-confidence core target samples, guided in both output and feature spaces. To reduce overfitting without multiple surrogates, BAT uses an ensemble of frozen discriminators obtained by pruning a single pretrained surrogate, and it remains effective under domain shift and even without real target-class images.

**Strengths:**

1. Multiple BAT variants are developed and evaluated, offering a clear view of design choices and their impact.
2. Against competitive baselines, BAT demonstrates consistent gains, supporting the claimed effectiveness.
3. On ImageNet-1K, BAT surpasses existing ℓ∞-constrained targeted attacks and is supported by theoretical bounds linking ensemble size to transferability, matching empirical trends.

**Weaknesses:**

1. The rationale for why randomly pruning a frozen surrogate yields effective discriminator ensembles is under-explained; a deeper analysis or ablation would clarify mechanism and sensitivity.
2. The preliminaries section is cluttered and difficult to follow; tighter structure and clearer notation would improve readability.
3. Criteria for identifying “high-confidence” target samples are insufficiently specified.

**Questions:**

1. What mechanism makes randomly pruned, frozen discriminator ensembles effective for targeted transfer?
2. How well do the theoretical bounds track empirical gains with ensemble size, does BAT generalize across model families beyond the surrogate, and what are the main failure modes?

---

> ### Author Response · Authors · 2025-11-23
> **Responses to Reviewer b9rh (1/3)**
>
> We thank the reviewer for their thoughtful and constructive feedback and for recognizing the strengths of our method and theory. Below, we address each concern and question in turn.
>
> ### Q1 / W1. What mechanism makes randomly pruned, frozen discriminator ensembles effective for targeted transfer?
>
> Random pruning of a single surrogate yields a **diverse yet competent ensemble**, and training the generator against this ensemble plus core target samples drives perturbations toward **shared, high-confidence target regions**, which improves targeted transfer until over-pruning destroys discriminative power.
>
> **Mechanism and supporting evidence:**
>
> - **Pruning induces diverse decision boundaries.**
>   Randomly pruning layers of a pretrained surrogate at moderate ratios yields subnetworks with **meaningfully different decision boundaries and attention maps** for adversarial examples (Fig. 1(b)), analogous to the diversity obtained from using different architectures (Fig. 1(a)). This shows that pruning is a way to approximate architectural diversity from one pretrained model.
>
> - **Ensembles enforce agreement on robust target features.**
>   Using an ensemble of discriminators prevents it from overfitting to a single discriminator. Instead, the generator is forced to find perturbations that are simultaneously high-confidence for **all** (frozen) pruned discriminators. This pushes the generator toward more robust, target-consistent features that transfer better to unseen victims. Empirically, TSR increases with ensemble size (Fig. 4(a)).
>
> - **Role of core target samples.**
>   We exploit the ensemble to extract **core target samples** that lie in high-confidence regions for the target class across discriminators. Mapping source images to these regions (instead of merely fooling one model) further increases the confidence and transferability of the generated examples.
>
> - **Diversity–accuracy trade-off and over-pruning.**
>   Our pruning strategy is designed to *retain enough discriminative power* while *injecting diversity* into the ensemble. A low pruning ratio preserves high accuracy but offers limited diversity; an aggressive pruning ratio yields highly diverse but weak discriminators that no longer provide reliable gradients. As shown in Table 7, there is a clear “sweet spot” where transferability improves as ensemble diversity increases, and then degrades once pruning becomes too aggressive.
>
> Finally, if multiple truly diverse architectures (e.g., CNNs and ViTs) are available, BAT can directly use them as the discriminator ensemble and benefits from even stronger diversity (Tab. 5), consistent with the above mechanism.
>
>
> ---

---

> ### Author Response · Authors · 2025-11-23
> **Responses to Reviewer b9rh (2/3)**
>
> ### Q2. How well do the theoretical bounds track empirical gains with ensemble size, does BAT generalize across model families beyond the surrogate, and what are the main failure modes?
>
> Our theoretical bounds predict **diminishing gains with increasing ensemble size**, which matches the empirical saturation trend in Fig. 4(a). BAT **generalizes across model families** (CNN $\leftrightarrow$ ViT) as demonstrated by additional ViT-surrogate experiments, and the main failure mode is **over-pruning**, where discriminators lose their discriminative power.
>
> **(a) How do the bounds relate to the empirical ensemble-size trend?**
>
> The lower bound on targeted transferability derived in the paper can be written as
> $$
> \Pr\big(T_r(\mathcal{D}_s, \bar{V}, \mathbf{x}^{\text{adv}}, y_t) = 1\big)
> \ge \xi - \zeta \sqrt{\kappa + \frac{\sigma}{B\sqrt{|\mathcal{D}_s|}}},
> $$
> where $\zeta,\kappa > 0$ and $|\mathcal{D}_s|$ is the ensemble size.
>
> The key term capturing the impact of ensemble size is $\sigma / \sqrt{|\mathcal{D}_s|}$, which yields:
>
> - When $|\mathcal{D}_s|$ is small, increasing the ensemble size **significantly improves** the lower bound.
> - As $|\mathcal{D}_s|$ grows, the benefit **quickly saturates**, since the term decays as $1 / \sqrt{|\mathcal{D}_s|}$.
>
> This is exactly the trend observed empirically in Fig. 4(a): targeted transferability improves as we increase the number of discriminators, then plateaus beyond a certain ensemble size. The upper bound from Theorem 2 also increases with $|\mathcal{D}_s|$, reinforcing the qualitative conclusion that larger, diverse ensembles are beneficial up to a point.
>
> **(b) Does BAT generalize across model families beyond the surrogate?**
>
> Yes. In the main text, we already use CNN surrogates (e.g., ResNet-50, DenseNet-121) and evaluate transfer to both CNN and ViT victims, showing cross-family transferability.
>
> To more directly test generalization, we additionally consider **ViT-B/16 as the surrogate** and evaluate transfer to both CNNs and other transformers on ImageNet-1K. The TSR (%) is:
>
> | Attack | VGG19-BN | ResNet50 | DenseNet121 | MobileNetV2 | ViT-B/16* | Swin-B | DeiT-B |
> |--------|----------|----------|-------------|-------------|-----------|--------|--------|
> | M3D    | 46.62    | 63.52    | 65.55       | 52.67       | 97.43*    | 57.15  | 76.42  |
> | BAT-CS | **50.12** (+3.50) | **66.91** (+3.39) | **68.20** (+2.65) | **56.76** (+4.09) | **97.83*** (+0.40) | **59.77** (+2.62) | **78.39** (+1.97) |
>
> Numbers in parentheses in the tables indicate the absolute TSR gain of BAT-CS (in percentage points) over the best baseline in each column. Even when using a transformer (ViT-B/16) as the surrogate, BAT consistently outperforms M3D across both CNN and transformer victims. This indicates that BAT’s mechanism is not tied to a single architecture family and generalizes across CNN and ViT models.
>
> **(c) Main failure modes**
>
> Our experiments reveal the following primary failure mode related to the ensemble construction:
>
> - **Over-pruning the discriminator ensemble.**
>   As shown in Table 7, once the pruning ratio becomes too high, the pruned subnetworks lose discriminative power (even on clean data), leading to **noisy or uninformative gradients**. In this regime, the generator cannot reliably target the intended adversarial region, and targeted transferability degrades.
>
> In the revision, we have (i) included the ViT-surrogate experiments in Appendix C (Table 13) and (ii) clearly described the over-pruning failure mode in the discussion of Table 7.
>
> ---

---

> > ### Author Response · Authors · 2025-11-23
> > **Responses to Reviewer b9rh (3/3)**
> >
> > ### W2. The preliminaries section is cluttered and difficult to follow; tighter structure and clearer notation would improve readability.
> >
> > We have revised the preliminary section to improve readability.
> >
> > ---
> >
> > ### W3. Criteria for identifying “high-confidence” target samples are insufficiently specified.
> >
> > We have made the **core sample selection rule explicit**: we define an ensemble confidence $\bar{p}(y_t \mid x)$ over the discriminators and choose high-confidence samples by ranking or optimizing this quantity, with variant-specific procedures for BAT-BS, BAT-CS, and BAT-CN.
> >
> > Let $\mathcal{D}_{\theta_k}$ denote the $k$-th discriminator and $p_k(y_t \mid x)$ its softmax probability for target class $y_t$. We define the **ensemble confidence**:
> > $$
> > \bar{p}(y_t \mid x) = \frac{1}{|\mathcal{D}_s|} \sum_k p_k(y_t \mid x).
> > $$
> >
> > We then use $\bar{p}(y_t \mid x)$ as follows (Sec. 3.2 will state this explicitly):
> >
> > - **BAT-BS:**
> >   For each target class, among available training images of that class, we **rank by $\bar{p}(y_t \mid x)$** and select the top 300 as *core target samples*.
> >
> > - **BAT-CS (Core Samples with strengthened confidence):**
> >   Starting from these 300 images, we run a **PGD-like optimization** under the discriminator ensemble (with target label $y_t$) to further **increase $\bar{p}(y_t \mid x)$**, yielding slightly perturbed but more confident core target samples.
> >
> > - **BAT-CN (Core Noise):**
> >   When no target-class images are available, we initialize random noise vectors and **optimize them to maximize $\bar{p}(y_t \mid x)$**. The top-performing noise vectors (by ensemble confidence) are kept as *core targets*.
> >
> > Thus, “high-confidence” is concretely defined via **ensemble-level probability**, not a single model’s score, and the procedures for BAT-BS, BAT-CS, and BAT-CN are fully specified in the paper.

---

### Official Review · Reviewer_EqLf · 2025-11-01

**Soundness:** 3
**Presentation:** 3
**Contribution:** 3
**Rating:** 6
**Confidence:** 4

**Summary:**

This paper proposes BAT (Boosting Adversarial Transferability), a generative framework that enhances targeted adversarial transferability by training generators to align outputs with curated high-confidence core target samples. The method employs an ensemble of frozen discriminators derived via pruning from a single pretrained surrogate model, eliminating the need for multiple surrogates. Theoretical bounds are provided to explain how ensemble size influences transferability. Experiments demonstrate consistent improvements in targeted success rates across various settings, including no domain shift, domain shift, and against robust models.​

**Strengths:**

Innovative Single-Surrogate Ensemble Approach: The paper introduces a novel method to create diverse discriminator ensembles through pruning of a single pretrained model, eliminating the need for multiple distinct surrogate models while still achieving strong transferability. This addresses a key limitation of existing methods that require access to multiple models.​
Theoretical Foundations with Practical Insights: The work provides rigorous theoretical analysis establishing lower and upper bounds on targeted transferability, revealing how ensemble size trades off with performance. This theoretical framework aligns well with empirical observations and offers valuable guidance for practical implementation.​

**Weaknesses:**

1. Your experimental evaluation primarily focuses on transfer scenarios using ResNet and DenseNet architectures. However, the increasing adoption of vision transformers (ViT) in computer vision applications necessitates a more comprehensive analysis of cross-architecture transferability. Could you please conduct additional experiments to address: (1) ResNet-to-ViT transferability across multiple ViT variants (e.g., ViT-Base, ViT-Large, DeiT); (2) ViT-as-surrogate transferability to traditional convolutional networks.

2. Universality under domain shift: Beyond the single Painting↔ImageNet experiment, could you provide additional results on other cross-domain pairs (e.g., Sketch, Photo, synthetic datasets) to substantiate that the “core-sample + self-ensemble” strategy remains robust across a wider spectrum of distributional discrepancies?

**Questions:**

See weaknesses for details.

---

> ### Author Response · Authors · 2025-11-23
> **Response to Reviewer EqLf (1/2)**
>
> We sincerely thank the reviewer for their thoughtful and constructive feedback. The reviewer recognized the novelty of our method in achieving strong transferability with access to a single surrogate model along with the theoretical contribution. Below, we address each of the raised concerns in detail.
> Numbers in parentheses in the tables indicate the absolute TSR gain of BAT-CS (in percentage points) over the best baseline in each column.
>
>
> ---
>
> ### W1. Cross-architecture transferability with ViTs
>
> We conduct additional experiments in two cross-family settings:
> (i) using a ViT surrogate and attacking both CNN and transformer victims, and
> (ii) using a CNN surrogate (as in the main paper) and attacking CNN and additional transformer victims.
>
> Overall, BAT-CS maintains **strong cross-architecture transferability**, with  **consistent gains over M3D on CNN victims** and **competitive performance on ViT-family victims**.
>
> **(a) ViT surrogate → CNNs and ViTs**
>
> We first use **ViT-B/16 as the surrogate** and evaluate targeted transfer to both CNN and transformer victims on ImageNet-1K. The TSR (%) comparison between the strong baseline M3D and BAT-CS is:
>
> | Attack | VGG19-BN | RN50 | DN121 | MN-V2 | ViT-B/16* | Swin-B | DeiT-B |
> |--------|----------|------|-------|-------|-----------|--------|--------|
> | M3D    | 46.62    | 63.52 | 65.55 | 52.67 | 97.43*    | 57.15  | 76.42  |
> | BAT-CS | **50.12** (+3.50) | **66.91** (+3.39) | **68.20** (+2.65) | **56.76** (+4.09) | **97.83*** (+0.40) | **59.77** (+2.62) | **78.39** (+1.97) |
>
> **Key observations.**
>
> - BAT-CS **improves TSR on all CNN victims and both ViT-family black-box victims** (Swin-B, DeiT-B), while preserving strong white-box performance on the ViT-B/16 surrogate.
> - The overall average TSR increases from 65.62% (M3D) to 68.28%, and the black-box-only average (excluding the surrogate) increases from 60.32% to 63.36%.
>
> **(b) CNN surrogate → CNNs and ViTs**
>
> We next consider **ResNet-50 as the surrogate** and evaluate transfer to three representative CNN victims and three transformer victims (ViT-B/16, Swin-B, DeiT-B). The TSR (%) on ImageNet-1K for M3D and BAT-CS is:
>
> | Attack | VGG19-BN | RN50* | DN121 | MN-V2 | ViT-B/16 | Swin-B | DeiT-B |
> |--------|----------|-------|-------|-------|----------|--------|--------|
> | M3D    | 82.57    | 95.77 | 88.32 | 81.54 | **51.73** | **45.45** | **41.86** |
> | BAT-CS | **94.04** (+11.47) | **98.78** (+3.01) | **94.16** (+5.84) | **86.60** (+5.06) | 50.45 (−1.28) | 42.37 (−3.08) | 37.71 (−4.15) |
>
> **Key observations.**
>
> - BAT-CS **consistently improves TSR on all CNN victims**, yielding substantial gains over M3D when using a CNN surrogate.
> - On the ViT-family victims (ViT-B/16, Swin-B, DeiT-B), BAT-CS remains **competitive** with M3D, while still providing markedly higher transferability on conventional CNNs.
>
> In summary, these experiments demonstrate that BAT remains effective against both CNN and ViT victims when ViT-B/16 is used as the surrogate, and that when ResNet-50 is used as the surrogate, BAT-CS offers strong improvements on CNN victims while maintaining competitive transferability to ViT-family models. We have added a paragraph titled **“ViT surrogate: cross-family transferability”** in Appendix C, which reports the ViT-surrogate experiments and is referenced in the main text.
>
> ---

---

> > ### Author Response · Authors · 2025-11-23
> > **Response to Reviewer EqLf (2/2)**
> >
> > ### W2. Universality under domain shift
> >
> > In addition to the ImageNet–Painting pair used in the paper for the domain-shift setting, we now evaluate BAT under a **second, substantially different domain shift** using the AnimeFace [1] dataset. BAT-CS again significantly outperforms the strong generative baseline CGNC-FT, providing further evidence that **BAT remains robust under diverse distribution shifts**.
> >
> > Following the same protocol as in the Painting experiments, and using **ResNet-50 as the surrogate**, we compare BAT-CS with CGNC-FT on the ImageNet evaluation set. The TSR (%) is:
> >
> > | Attack  | RN18  | RN50* | RN101 | DN121 | DN161 | VGG16-BN | VGG19-BN | MN-V2 | ViT-B/16 |
> > |---------|-------|-------|-------|-------|-------|----------|----------|-------|----------|
> > | CGNC-FT | 88.06 | 97.97* | 92.51 | 90.93 | 90.68 | 89.01 | 85.07 | 81.30 | 54.73 |
> > | BAT-CS  | **92.21** (+4.15) | **98.92*** (+0.95) | **94.77** (+2.26) | **94.17** (+3.24) | **94.57** (+3.89) | **93.11** (+4.10) | **92.17** (+7.10) | **88.91** (+7.61) | **64.09** (+9.36) |
> >
> > **Key observations.**
> >
> > - BAT-CS **improves TSR on every victim model**, often by a substantial margin (e.g., +7.1% on VGG19-BN, +7.6% on MN-V2, +9.4% on ViT-B/16).
> > - The gains hold across **both CNN and transformer victims**, indicating that the benefits of the core-sample + self-ensemble design persist under different domain shifts.
> >
> > These ImageNet–AnimeFace results, together with the existing ImageNet–Painting experiments, demonstrate that BAT remains effective across **distinct cross-domain settings**.
> >
> > We have added the AnimeFace–ImageNet experiments to Appendix C (Table 12) in the revised version and reference them in the main text when discussing performance under domain shift.
> >
> > >[1] Gwern Branwen. “Danbooru2019 portraits: A large-scale anime head illustration dataset.” 2019.

---

### Author Response · Authors · 2025-11-23
**General Response to All the Reviewers**

We sincerely thank all reviewers for their time, careful evaluation, and constructive feedback. Beyond the per-reviewer replies, we (1) acknowledge key strengths noted by the reviews, (2) summarize the main concerns, and (3) highlight the corresponding updates in the revision.

## Key Contributions Acknowledged by Reviewers

- **Single-Surrogate Self-Ensemble.**
  BAT builds a diverse discriminator ensemble from a *single* pretrained surrogate via pruning, avoiding multiple surrogates while preserving strong targeted transferability. \[EqLf, b9rh, Q6HX\]
- **Core Targets + Dual-Space Alignment.**
  The modular design—core target samples plus dual-space feature/logit alignment under a frozen ensemble—was found clear and well motivated. \[Q6HX, Np53\]
- **Strong and Broad Empirical Gains.**
  BAT consistently improves TSR over state-of-the-art baselines across many CNN and ViT victims, including domain shift and robust models, with detailed ablations. \[EqLf, Np53, Q6HX\]
- **Theory on Ensemble Size.**
  Theoretical upper/lower bounds linking ensemble size to targeted transferability, and explaining saturation, were viewed as practically useful and consistent with empirical trends. \[EqLf, b9rh, Q6HX\]

---
## Major Concerns Raised by Reviewers

- **Cross-Architecture / ViT Transferability:** Need for more systematic evaluation of CNN→ViT and ViT-as-surrogate transfer. \[EqLf, b9rh\]
- **Domain-Shift Universality:** Desire for additional cross-domain pairs beyond Painting↔ImageNet. \[EqLf\]
- **Pruning Mechanism and Diversity:** Request to quantify pruning-induced diversity, relate it to TSR, and discuss failure modes (e.g., over-pruning). \[b9rh, Q6HX\]
- **Statistical Stability & Threat-Model Breadth:** Ask for multi-seed statistics and tests beyond a single $\ell_{\infty}$ setting ($\ell_2$, simple spatial/color threats). \[Q6HX\]
- **Generator Architecture & Baselines:** Generator description considered under-specified; request to compare with targeted versions of strong untargeted attacks (e.g., Admix). \[Np53\]
- **Data- vs. Model-Level Self-Ensemble and Novelty:** Clarify how augmentations interact with pruning-based self-ensemble and articulate the conceptual novelty vs. prior works on ensembles, feature alignment, and target anchors. \[Np53\]
- **Core Sample Definition & Preliminaries:** Need for a precise definition of “high-confidence” core target samples and clearer preliminaries / notation. \[b9rh\]
- **Compute and Latency:** Request for explicit discussion of training cost vs. ensemble size and inference latency. \[Q6HX\]

All the major concerns are addressed in the revision.

---
## Major Updates in the Revised Version

In the rebuttal and revised paper, we made the following key updates:

1. **Expanded CNN↔ViT Experiments**
   We add ViT-B/16-as-surrogate → {CNNs, ViTs} experiments (Tab. 13, App. C). BAT-CS consistently improves over M3D on both CNN and transformer victims, demonstrating cross-family generalization.
2. **Additional Domain-Shift Setting**
   Besides Painting↔ImageNet, we now include AnimeFace↔ImageNet experiments (Tab. 12, App. C). BAT-CS substantially outperforms CGNC-FT on CNN and ViT victims, reinforcing robustness of BAT under diverse distribution shifts.
3. **Diversity–TSR Analysis and Failure Mode**
   We introduce a decision-disagreement metric between the base ResNet-50 and pruned variants and report TSR vs. pruning ratio (Tab. 7). Moderate pruning induces diversity comparable to multi-architecture ensembles and maximizes TSR, while heavy pruning causes excessive disagreement and degraded TSR, now identified as the main failure mode.
4. **Multi-Seed and Broader Threat Models**
   We report 5-seed mean ± std TSR for M3D and BAT-BS (Table 16, App. C), showing higher means with comparable variance. Using the same $\ell_\infty$-trained generators, we also evaluate $\ell_2$ projection threat without retraining (Tab 14, App. C).
5. **Detailed Generator Architecture**
   We provide the generator architecture in Appendix F.
6. **Core Target Definition, Self-Ensemble, and Clarity**
   We formalize ensemble confidence and state explicit selection rules for core targets in BATs (Sec. 3.2). We also clarify that BAT already couples strong data-level augmentations (data-level self-ensemble), and we streamline preliminaries and notation.
7. **Compute and Latency**
   We report training time per target class as $|\mathcal{D}\_s|$ varies, showing approximately linear scaling with negligible overhead from core-set construction, and emphasize that inference requires only a single forward pass through $G_\Phi$ plus projection, independent of $|\mathcal{D}_s|$ and substantially faster than iterative attacks (App. D, Fig. 6).
---

**Note.**
All major revisions are highlighted in the revised manuscript. Equation, figure, and table numbers in the responses refer to the revised version; when needed, we also indicate the original numbering for clarity.

---

---

### Author Response · Authors · 2025-12-03
**Response to AC**

**Dear Area Chair,**

Thank you for taking over the evaluation of our submission during this unusual situation. We sincerely appreciate your time and effort. Our paper proposes **BAT**, a generative targeted-transfer attack that constructs a diverse self-ensemble from a *single* surrogate via pruning and aligns source images to high-confidence **core target samples** using dual-space (logit + feature) alignment. BAT achieves strong targeted transferability under both domain-matched and domain-shift settings.

### Strengths
Reviewers consistently highlighted:
- the effectiveness of constructing self-ensemble discriminators from a *single* pretrained model in generative adversarial attacks;
- the clear modular design for aligning generated adversarial examples with core targets in dual space across discriminators;
- strong empirical gains over state-of-the-art targeted-transfer attacks, e.g., BAT-CS improves the average TSR from ≈83% (M3D) to ≈89% on ImageNet-1K with ResNet50 as surrogate under the domain-matched scenario; and
- theoretical bounds that explicitly capture a $1/\sqrt{|\mathcal{D}_s|}$ dependence on ensemble size, explaining why gains saturate as $|\mathcal{D}_s|$ increases and matching empirical trends.


They also noted the thorough ablations and robustness of BAT-CN when target images are unavailable.

### Reviewer Concerns and Our Responses
- **Cross-architecture / ViT transferability (EqLf, b9rh):**
  We added CNN→ViT and ViT-surrogate→CNN/ViT experiments (App. C, Tab. 13), showing consistent improvements over M3D.
- **Domain-shift universality (EqLf):**
  We added a second domain-shift setting (AnimeFace↔ImageNet, Tab. 12) where BAT-CS improves TSR on every victim, with gains of about 5\% over the strong baseline CGNC-FT.
- **Pruning mechanism & ensemble diversity (b9rh, Q6HX):**
  We quantified disagreement vs. pruning ratio (Table 7), showed that moderate pruning yields diversity comparable to multi-architecture ensembles and enhances TSR, and identified over-pruning as the main failure mode.
- **Statistical stability & broader threats (Q6HX):**
  We added 5-seed TSR comparisons for M3D and BAT-BS (Tab. 16). Additionally, we evaluated $\ell_2$ and spatial/color (ColorJitter) threats **without retraining**, where BAT-CS maintains high TSR (e.g., ≈87% vs 79% for M3D under $\ell_2$, and ≈89% vs 83% with ColorJitter).
- **Generator architecture & targeted baselines (Np53):**
  We added a full layer-by-layer generator description (App. F) and compared with targeted Admix/BSR variants, where BAT-CS significantly outperforms them on all victims.
- **Core target definition & preliminaries (b9rh):**
  We provided explicit ensemble-confidence $\bar{p}(y_t \mid x)$ rules and selection/optimization procedures for BAT-BS/CS/CN, and streamlined the preliminaries for clarity.
- **Compute & latency (Q6HX):**
  We further clarified runtime analysis (App. D, Fig. 6), showing training time scales approximately linearly with ensemble size and that inference requires only a single generator forward pass plus projection, independent of $|\mathcal{D}_s|$.

All tables and figures referenced above are included in the revised submission.

We believe the revised submission satisfactorily addresses all concerns and maintains a principled and clear presentation of the method. Thank you again for your time and consideration.

---

### Meta-Review · Area_Chair_oH7F · 2026-01-06

**Summary:**

The submission received mixed evaluations, with two positive ratings and two negative ratings.

Reviewer EqLf requests additional evaluations on ViT-based models and in cross-domain settings. Reviewer b9r’s primary concern is that the underlying rationale of the method is insufficiently explained. Reviewer Np53 asks for clearer descriptions of the model architecture, more comprehensive comparisons with related work, and additional evaluations on other augmentation strategies; concerns regarding the novelty of the approach are also raised. Reviewer Q6HX points out weaknesses in statistical reporting, insufficient analysis of assumptions, and recommends further evaluations on latency as well as quantitative analysis of ensemble diversity.

**Reviewer Concerns:**

The authors have provided a point-by-point response to the reviewers’ comments. The AC believes that the majority of concerns have been adequately addressed; however, two issues remain outstanding.

First, the AC concurs with Reviewer Np53 that the novelty of the proposed method is not clearly articulated, as ensembling surrogate models to improve transferability is a well-established idea in this research area.

Second, the reviewers requested extensive additional experiments to more thoroughly evaluate the method. While the AC acknowledges the authors’ efforts during the rebuttal period, incorporating all of these evaluations would likely require more than a light revision.

**Reviewer Scores:**

All review scores will remain unchanged.

---

### Decision · Program_Chairs · 2026-01-26

Reject